# To Distill or Decide? The Algorithmic Trade-off in Partially Observable Reinforcement Learning

**Yuda Song**
CMU
yudas@cs.cmu.edu

**Dhruv Rohatgi**
MIT
drohatgi@mit.edu

**Aarti Singh**
CMU
aarti@cs.cmu.edu

**J. Andrew Bagnell**
Aurora Innovation, CMU
dbagnell@aurora.tech

## Abstract

Partial observability is a notorious challenge in reinforcement learning (RL), due to the need to learn complex, history-dependent policies. Recent empirical successes have used *privileged expert distillation* — which leverages availability of latent state information during training (e.g., from a simulator) to learn and imitate the optimal latent, Markovian policy — to disentangle the task of "learning to see" from "learning to act" [56, 12, 9]. While expert distillation is more computationally efficient than RL without latent state information, it also has well-documented failure modes. In this paper — through a simple but instructive theoretical model called the *perturbed Block MDP*, and controlled experiments on challenging simulated locomotion tasks — we investigate the algorithmic trade-off between privileged expert distillation and standard RL without privileged information. Our main findings are: **(1)** The trade-off empirically hinges on the *stochasticity* of the latent dynamics, as theoretically predicted by contrasting *approximate decodability* with *belief contraction* in the perturbed Block MDP; and **(2)** The optimal latent policy is not always the best latent policy to distill. Our results suggest new guidelines for effectively exploiting privileged information, potentially advancing the efficiency of policy learning across many practical partially observable domains.

## 1 Introduction

Partial observability is a common challenge in applied reinforcement learning: the decision-making agent may not see the true state of the environment at all time-steps, whose information might only be probabilistically inferred from the history of observations. An illustrative task is robot learning for robots with *image-based perception* [58, 68]. A single image of the robot (or, in first-person perspective, of the environment) will not capture important elements of the state such as the robot's velocity, and may miss other features due to e.g. occlusion or limited view.

The canonical theoretical model for such tasks is *Partially Observable Markov Decision Process (POMDP)*. Unfortunately, there are well-documented computational [57] and statistical [28] barriers to planning and learning in POMDPs, which have motivated many theoretical works that seek to bypass these barriers by making additional structural assumptions [28, 33, 19, 24, 23, 42]. On the empirical side, the standard technique for mitigating partial observability is *frame-stacking*, which enabled notable successes for learning to play Atari games [50, 51]. The idea is to treat the "state" of the environment as the concatenation of a short window of $L$ recent observations, and apply a standard algorithm for fully-observed reinforcement learning (RL). This technique inspired theoretical developments such as $L$-step decodability [19], and has some theoretical underpinnings for $\gamma$-*observable* POMDPs [24]. Yet frame-stacking is not a silver bullet for partially observable decision-making: sometimes effective planning requires long memory [18]. Also, high-dimensional observations (such as stacks of images) can confound learning complex behaviors [58, 78].

39th Conference on Neural Information Processing Systems (NeurIPS 2025).

**Learning from latent state information.** A common heuristic for planning in known POMDPs is to use the optimal *latent* policy (also known as the state-based policy or privileged policy) — i.e., the optimal policy that is allowed to "cheat" and see the underlying state of the environment — as a starting point for computing an *executable* policy — i.e. a policy that only depends on the observable history [40, 65, 12]. More recent works have brought this ansatz to the learning task, where the description of the POMDP is a priori unknown. In the standard theoretical formalization of this task [31], the latent states of the POMDP are never observed (nor even identifiable); however, for applications such as robotics, it is often practically reasonable to construct a *simulator* of the environment [13], from which the learning agent may draw trajectories that include both the observations as well as the latent states — "privileged" information that is only available at training time, not at test time.

The most prominent paradigm for exploiting this additional information is called *privileged expert distillation*,[1] which applies methods from imitation learning and structured prediction [15, 62, 64, 8] to learning in POMDPs. Expert distillation has two steps: (1) learn an optimal latent policy, using a standard RL algorithm with the latent state information provided by the simulator; and (2) distill the latent policy to an executable policy, using an imitation learning algorithm such as `DAgger` [64]. This paradigm has achieved impressive success in applications such as autonomous driving [9], robotics [37, 48, 85] and LLMs [11].

These successes suggest a fundamental question: *when does expert distillation help in realistic decision-making tasks*? On the one hand, in controlled experiments, expert distillation uniformly converges faster and more stably than RL without latent state information [53], likely because it disentangles representation learning from decision-making [9]. Moreover, expert distillation enjoys a provable computational advantage in decodable POMDPs[2] [7]. On the other hand, there are well-documented failure modes of expert distillation — most notably, due to its inability to encourage purely information-gathering actions [2, 79] — where more expensive hybrid methods such as Asymmetric Actor-Critic [58] are fundamentally required.

In this paper, motivated by image-based locomotion tasks, we focus on the middle ground where (perfect) decodability may fail, yet the observations are still highly informative of the latent state. In this regime, we ask: (i) when and how is expert distillation as performant as standard RL with frame-stacking, and (ii) are there lightweight *improvements* to expert distillation? We use simple theoretical models in tandem with controlled experiments to address the preceding questions.

**Our contributions.**

1. The prior theoretical model for understanding the benefits of latent state information was a *perfectly decodable* POMDP [7]. We begin by empirically demonstrating that this model is too restrictive for image-based locomotion tasks.

2. We then introduce *approximate decodability*, and connect it to the success of expert distillation — in analogy with the connection between *belief contraction* and the success of standard reinforcement learning with frame-stacking. But when are these conditions satisfied? As a theoretical testbed, we introduce the *perturbed Block MDP*.

3. We show both theoretically (by analyzing the perturbed Block MDP model) and experimentally that the performance of expert distillation compared to standard reinforcement learning depends crucially on the *stochasticity* of the model dynamics: for deterministic dynamics, distillation is competitive with RL, but as the stochasticity increases, its performance comparatively degrades.

4. Finally, we show that distillation of the optimal latent policy is often a sub-optimal use of latent state information: the simple modification of *adding stochasticity to the latent MDP* before computing the optimal policy yields robust performance benefits via improved *smoothness*.

## 2 Preliminaries

A (finite-horizon, layered) Partially Observable Markov Decision Process (POMDP) is a tuple $\mathcal{P} = (H, \mathcal{X}, \mathcal{S}, \mathcal{A}, \mathbb{P}, \mathbb{O}, R)$, where $H \in \mathbb{N}$ is the horizon, $\mathcal{X} = \{\mathcal{X}_h\}_{h=1}^{H}$ is the observa-

---

[1]The same paradigm is sometimes called *Learning by Cheating* or *Teacher to Student Learning*.
[2]Decodable POMDPs without any prefix refer to $H$-step decodable POMDPs, where $H$ is the horizon of the POMDPs.

tion space, $\mathcal{S} = \{\mathcal{S}_h\}_{h=1}^H$ is the latent state space, $\mathcal{A} = \{\mathcal{A}_h\}_{h=1}^H$ is the action space, $\mathbb{P} = \{\mathbb{P}_h : \mathcal{S}_{h-1} \times \mathcal{A}_{h-1} \to \Delta(\mathcal{S}_h)\}_{h=1}^H$ describes the latent transitions, $\mathbb{O} = \{\mathbb{O}_h : \mathcal{S}_h \to \Delta(\mathcal{X}_h)\}_{h=1}^H$ describes the emission distributions, and $R = \{R_h : \mathcal{S}_h \times \mathcal{A}_h \to [0,1]\}$ describes the rewards. We write $A := \max_h |\mathcal{A}_h|$, $S := \max_h |\mathcal{S}_h|$, and $X := \max_h |\mathcal{X}_h|$. Given any timestep $h$ and $L \in [H]$, we denote $\mathcal{X}^{h-L:h} := \mathcal{X}_{h-L} \times \mathcal{X}_{h-L+1} \times \cdots \times \mathcal{X}_h$, and similarly for $\mathcal{A}^{h-L:h}$, with the shorthand $h - L := \max\{1, h - L\}$. Then an $L$-step *executable* policy is a collection $\pi = \{\pi_h : \mathcal{X}^{h-L+1:h} \times \mathcal{A}^{h-L:h-1} \to \Delta(\mathcal{A}_h)\}$; we let $\Pi^L$ denote the class of such policies. Given any executable policy $\pi \in \Pi := \Pi^H$, a trajectory $\tau = (s_1, x_1, a_1, r_1, \ldots, s_H, x_H, a_H, r_H)$ is generated by $s_h \sim \mathbb{P}_h(s_{h-1}, a_{h-1})$, $x_h \sim \mathbb{O}_h(s_h)$, $a_h \sim \pi(x_{1:h}, a_{1:h-1})$, $r_h = R_h(s_h, a_h)$. We use $\mathbb{P}^\pi$ and $\mathbb{E}^\pi$ to denote the law and expectation under this process. Following convention, we assume $\sum_{h=1}^H r_h \le 1$ almost surely under all policies. The *value* of a policy $\pi$ is $J(\pi) := \mathbb{E}^\pi \left[ \sum_{h=1}^H r_h \right]$.

Note that the POMDP $\mathcal{P}$ also defines an underlying Markov Decision Process (MDP) $\mathcal{M} = \{\mathcal{S}, \mathcal{A}, \mathbb{P}, R, H\}$ (which we call the *latent MDP*) where the state is fully observable. A latent (Markovian) policy is a collection $\pi^{\text{latent}} = \{\pi_h^{\text{latent}} : \mathcal{S}_h \to \Delta(\mathcal{A}_h)\}$, and we let $\Pi^{\text{latent}}$ denote the class of latent policies. A latent trajectory $\tau^{\text{latent}} = (s_1, a_1, \ldots, s_H, a_H)$ is generated by $s_h \sim \mathbb{P}_h(s_{h-1}, a_{h-1})$, $a_h \sim \pi_h^{\text{latent}}(s_h)$, and we define $\mathbb{P}^{\pi^{\text{latent}}}$ and $\mathbb{E}^{\pi^{\text{latent}}}$ accordingly.

**Learning with/without latent state information.** In the standard theoretical RL access model (i.e. without latent state information) [30, 27], at training time, the learning agent can repeatedly interact with the POMDP $\mathcal{P}$ by playing an executable policy $\pi$ and observing the partial trajectory $(x_{1:H}, a_{1:H}, r_{1:H})$. In contrast, in the *learning with latent state information* model [7], at training time, the learning agent can play *any* policy, and observes the full trajectory $(s_{1:H}, x_{1:H}, a_{1:H}, r_{1:H})$. In both settings, the goal is to eventually produce an *executable* policy $\widehat{\pi}$ that minimizes $J(\pi^\star) - J(\widehat{\pi})$ (where $\pi^\star$ is the optimal executable policy).

**Belief states.** A *belief state* is a distribution over latent states. For a prior $b$ on the latent state at step $h - 1$, let $\mathbb{U}_h(b; a_{h-1}, x_h)$ be the posterior on the latent state at step $h$ after taking action $a_{h-1}$ and then observing $x_h$ (see Definition B.1 for the formal algebraic definition).

**Definition 2.1.** *For any observation/action sequence* $(x_{1:h}, a_{1:h-1})$, *the* true belief state $\mathbf{b}_h(x_{1:h}, a_{1:h-1})$ *is defined as follows. For* $h = 1$ *with observation* $x_1$, *let* $\mathbf{b}_1(x_1) := \mathbb{B}_1(\mathbb{P}_1; x_1)$. *For any* $2 \le h \le H$, *let*

$$\mathbf{b}_h(x_{1:h}, a_{1:h-1}) := \mathbb{U}_h(\mathbf{b}_{h-1}(x_{1:h-1}, a_{1:h-2}); a_{h-1}, x_h). \tag{1}$$

For any executable policy $\pi$, step $h$, and history $(x_{1:h}, a_{1:h-1})$, $\mathbf{b}_h(x_{1:h}, a_{1:h-1})$ is the distribution of the latent state $s_h$ under $\mathbb{P}^\pi$, conditioned on $(x_{1:h}, a_{1:h-1})$ (Lemma C.2).

Many methods for efficient planning in POMDPs are based on *approximate belief states* that only depend on a short window of recent actions and observations [29, 24]. Informally, the approximate belief state $\mathbf{b}_h^{\text{apx}}(x_{h-L+1:h}, a_{h-L:h-1}; \mathcal{D})$ is the posterior on state $s_h$ after observing $(x_{h-L+1:h}, a_{h-L:h-1})$ with prior $\mathcal{D}$ on state $s_{h-L}$. See Definition B.2 for the formal definition (analogous to Definition 2.1).

**Additional notation.** For distributions $b, b' \in \Delta(\mathcal{S}_h)$, the density ratio is $\|b/b'\|_\infty = \sup_{s \in \mathcal{S}_h} b(s)/b'(s) \in [1, \infty]$, with the convention that $0/0 = 1$. For a belief state $b \in \Delta(\mathcal{S}_h)$ and conditional distribution $\pi_h : \mathcal{S}_h \to \Delta(\mathcal{A}_h)$, we let $\pi_h \circ b$ denote the distribution over $\mathcal{A}_h$ obtained as

$$(\pi_h \circ b)(a_h) := \sum_{s_h \in \mathcal{S}_h} b(s_h)\pi_h(a_h \mid s_h). \tag{2}$$

**Experimental Setup.** We use three tasks in the Deepmind control suite [73]: walker-run, dog-walk and the challenging humanoid-walk. To implement online (resp., offline) expert distillation, we (1) train an expert on the latent state information using MrQ [22], and (2) imitate the expert via DAgger [64] (resp., Behavior Cloning (BC)) on $L$-step executable policies. Unless otherwise specified, we use the standard choice of $L = 3$, and we use mean squared error (MSE) as the loss function: give input $X = \{x^i\}_{i=1}^N$ and target $Y = \{y^i \in \mathbb{R}^d\}_{i=1}^N$, the loss of a function $f$ is $\ell(f, X, Y) = \frac{1}{Nd} \sum_{i=1}^N \sum_{j=1}^d (f(x^i)_j - y_j^i)^2$. To implement reinforcement learning (RL), we use MrQ [22] on $L$-step executable policies. In experiments, we follow the common empirical practice of only stacking observations (rather than both observations and actions).

**Appendices.** See Appendix A for additional related work, and Appendix H for experimental details.

# 3  Approximate Decodability and Belief Contraction

Even with access to latent state information during training, the problem of learning a near-optimal policy in a POMDP is as hard as the *planning* task (where a description of the POMDP is already known), which is well-known to be computationally intractable in the worst case [57]. However, POMDPs encountered in practice will often satisfy additional structural properties that may mitigate this hardness. Some of the most widely-studied properties are *decodability* [19, 7] and *belief contraction* (also known as *filter stability*) [29, 24].

Privileged information is known to yield a provable computational benefit in decodable POMDPs [7]. However, as we empirically demonstrate in Section 3.1, perfect decodability is an unrealistic assumption in our motivating tasks. For this reason, in Section 3.2 we introduce the notion of *approximate decodability*. Heuristically, this property governs the success of expert distillation with $L$-step framestacking, whereas belief contraction governs the success of standard RL (also with $L$-step framestacking). But when are these properties satisfied? As a clean theoretical testbed for studying this question, in Section 3.3 we introduce the $\delta$-*perturbed Block MDP*.

## 3.1  Prior Work: Perfectly Decodable POMDPs

In some applications, such as video games, it is plausible that the agent can deduce the latent state from a small number of recent observations. This was empirically substantiated by the success of DQN [50] and its variants, which only use the most recent four observations as policy inputs. Theoretically, this motivated the study of the $L$-step decodable model [19], which posits that the most recent $L$ observations and actions suffice to fully disambiguate the latent state (Definition B.3).

Without latent state information (i.e. in the standard RL access model), learning a near-optimal policy in an $L$-step decodable POMDPs requires $\Omega(A^L)$ samples [19]. However, with latent state information, [7] show that the sample and time complexity of learning a near-optimal policy $\hat{\pi} \in \Pi^M$ such that $J(\hat{\pi}) \geq \arg\max_{\pi \in \Pi^M} J(\pi) - \varepsilon$ with high probability is only $\mathrm{poly}(S, A, X, H, 1/\varepsilon)$. Thus, for large $L$, there is a clear theoretical benefit of latent state information (both statistically and computationally). However, unfortunately, $L$-step decodability is not always a realistic assumption:

**Empirical test: does perfect decodability hold?** Through controlled experiments on our three chosen locomotion tasks (Section 2), we observe that latent states are not perfectly decodable in practice, especially in early timesteps. We defer details of this experiment to Appendix H.1.

## 3.2  Errors in POMDPs

The above empirical result motivates the following theoretical definition of decodability error:

**Definition 3.1** (Decodability Error). *Fix a POMDP $\mathcal{P}$. The* decodability error *for an executable policy $\pi$ and timestep $h \in [H]$ is*

$$\varepsilon_h^{\mathsf{decode}}(\pi) := \mathbb{E}^\pi[1 - \|\mathbf{b}_h(x_{1:h}, a_{1:h-1})\|_\infty].$$

Intuitively, decodability error quantifies stochasticity of the true belief. Below, we show that it upper bounds the *misspecification* of any latent policy $\pi^{\mathsf{latent}}$ with respect to the class of executable policies.

**Lemma 3.1** (See Lemma E.3). *Let $\pi^{\mathsf{latent}} \in \Pi^{\mathsf{latent}}$ be a latent policy and let $\widetilde{\mathbf{b}}_{1:H}$ be a collection of functions $\widetilde{\mathbf{b}}_h : \mathcal{X}^h \times \mathcal{A}^{h-1} \to \Delta(\mathcal{S}_h)$. Define executable policies $\widetilde{\pi}, \pi$ by $\widetilde{\pi}(x_{1:h}, a_{1:h-1}) := \pi^{\mathsf{latent}} \circ \widetilde{\mathbf{b}}_h(x_{1:h}, a_{1:h-1})$ and $\pi(x_{1:h}, a_{1:h-1}) := \pi^{\mathsf{latent}} \circ \mathbf{b}_h(x_{1:h}, a_{1:h-1})$ (see Eq. (2)). Then*

$$\mathsf{TV}(\mathbb{P}^{\pi^{\mathsf{latent}}}, \mathbb{P}^{\widetilde{\pi}}) \leq \sum_{h=1}^{H} 2\varepsilon_h^{\mathsf{decode}}(\pi) + \mathbb{E}^{\widetilde{\pi}}\left[\left\|\mathbf{b}_h(x_{1:h}, a_{1:h-1}) - \widetilde{\mathbf{b}}_h(x_{1:h}, a_{1:h-1})\right\|_1\right]. \quad (3)$$

Low decodability error is not strictly required for low misspecification (see Section 6), but some such assumption is needed to rule out models requiring active information-gathering [79]. As a special case, Lemma 3.1 implies that $\pi^{\mathsf{latent}}$ is $2\sum_{h=1}^{H} \varepsilon_h^{\mathsf{decode}}(\pi)$-close to the executable policy $\pi$, which evaluates $\pi^{\mathsf{latent}}$ at a random state $s'_h$ sampled from the true belief $\mathbf{b}_h(x_{1:h}, a_{1:h-1})$; this is because if $\mathbf{b}_h(x_{1:h}, a_{1:h-1})$ is highly concentrated, then $s'_h$ likely matches the true latent state. The second error term in Eq. (3) quantifies error in learning the true belief — e.g., due to using only $L$-step histories.

Next, it is instructive to contrast decodability error with *belief contraction error*, the discrepancy between the true belief and the approximate belief induced by the $L$ most recent observations/actions:

**Definition 3.2** (Belief Contraction Error [24]). *Fix a POMDP $\mathcal{P}$. For an executable policy $\pi$, and timestep $h \in [H]$, the $L$-step belief contraction error ($L \in [h-1]$) is*

$$\varepsilon_h^{\mathsf{contract}}(\pi; L) := \mathbb{E}^\pi\left[\left\|\mathbf{b}_h(x_{1:h}, a_{1:h-1}) - \mathbf{b}_h^{\mathsf{apx}}(x_{h-L+1:h}, a_{h-L:h-1}; \mathsf{unif}(\mathcal{S}_{h-L}))\right\|_1\right].$$

In the absence of latent state information, bounding the belief contraction error is the standard method of analyzing provably efficient algorithms for RL in POMDPs [29, 75, 24]. Indeed, belief contraction implies that the POMDP with $L$-step frame-stacking is approximately Markovian, which heuristically suggests that a standard RL algorithm [30, 5] with frame-stacking should achieve low error in time $\approx (AX)^{\mathcal{O}(L)}$. Due to technical issues with error compounding, this is not formally true, but under an additional *observability* condition, there *is* an algorithm that provably achieves that guarantee:

**Theorem 3.1** (Informal; see Theorem B.1; due to [23]). *Suppose the POMDP is $\gamma$-observable (Definition B.4), and satisfies $L$-step belief contraction with error $\varepsilon$.[3] There exists a reinforcement learning algorithm that achieves the sub-optimality bound*

$$J(\pi^\star) - J(\pi^{\mathsf{rl}}) \leq \varepsilon \cdot \mathrm{poly}(S, X, H, \gamma^{-1}),$$

*in time $(XA)^{\mathcal{O}(L)} \cdot \mathrm{poly}(H, S, \gamma^{-1}, \varepsilon^{-1})$.*

Technically, the explicit result in [23] fixes $L \sim \log^4(SH/\varepsilon)/\gamma$ (in which case the desired belief contraction bound is implied by $\gamma$-observability, but the algorithm requires quasi-polynomial time), but we observe that the proof extends to the result above — see Theorem B.1. Notably, Theorem 3.1 gives a polynomial-time algorithm if belief contraction holds for $L = \mathcal{O}(1)$.

### 3.3 The Perturbed Block MDP

Approximate decodability and belief contraction are conditions under which expert distillation and standard RL with frame-stacking, respectively, may be reasonably expected to succeed. But when are these conditions satisfied, and how do they compare? As a theoretical testbed, we introduce the *perturbed Block MDP* model. Block MDPs [16] are a well-studied abstraction of environments with rich observations yet simple latent dynamics. However, they assume that the latent state is fully determined by the current observation. Below, we generalize Block MDPs by allowing for $\delta$ probability that the observation is sampled from an arbitrary conditional distribution.[4]

**Definition 3.3.** *Fix a parameter $\delta > 0$. A POMDP $\mathcal{P}$ is a $\delta$-perturbed Block MDP if, for each $h \in [H]$, there are $\widetilde{\mathbb{O}}_h, E_h : \mathcal{S}_h \to \Delta(\mathcal{X}_h)$ such that $\widetilde{\mathbb{O}}_h : \mathcal{S}_h \to \Delta(\mathcal{X}_h)$ satisfies the block property [17], i.e. $\widetilde{\mathbb{O}}_h(\cdot \mid s_h), \widetilde{\mathbb{O}}_h(\cdot \mid s_h')$ have disjoint supports for all $s_h \neq s_h'$, and moreover the emission distribution $\mathbb{O}_h$ at step $h$ can be decomposed as follows: $\mathbb{O}_h(x_h \mid s_h) = (1 - \delta)\widetilde{\mathbb{O}}_h(x_h \mid s_h) + \delta E_h(x_h \mid s_h)$.*

A simple example is the *noisy sensor* model where $\mathcal{S} = \mathcal{X}$ and the true state is observed with probability at least $1 - \delta$. Later, we will examine the empirical validity of this model; for now we study its theoretical implications. Below, we prove that for any $\delta$-perturbed Block MDP, the belief contraction error decays exponentially as the frame-stack increases, by a factor of $O(\delta)$ per frame.

**Theorem 3.2** (See Theorem D.1). *Suppose that the POMDP $\mathcal{P}$ is a $\delta$-perturbed Block MDP. There is a universal constant $C_{D.1} > 1$ with the following property. Fix an executable policy $\pi$, indices $1 \leq h - L < h \leq H$, and a distribution $\mathcal{D} \in \Delta(\mathcal{S}_{h-L})$. Then for any partial history $(x_{1:h-L}, a_{1:h-L-1})$,*

$$\mathbb{E}^\pi\left[\|\mathbf{b}_h(x_{1:h}, a_{1:h-1}) - \mathbf{b}_h^{\mathsf{apx}}(x_{h-L+1:h}, a_{h-L:h-1}; \mathcal{D})\|_1\right] \leq (C_{D.1}\delta)^{L/9}\left\|\frac{\mathbf{b}_h(x_{1:h-L}, a_{1:h-L-1})}{\mathcal{D}}\right\|_\infty$$

*where the expectation is over trajectories drawn from policy $\pi$ conditioned on the partial history $(x_{1:h-L}, a_{1:h-L-1})$. Thus, in particular, $\varepsilon_h^{\mathsf{contract}}(\pi; L) \leq (C_{D.1}\delta)^{L/9}S$.*

---

[3]Technically, the result requires slightly generalizing Definition 3.2; see Theorem B.1 for the formal statement.

[4]To be clear, our theoretical focus is on issues arising from partial observability, not on representation learning. The size of the observation space is conceptually tangential, so we omit introducing technical complications such as function approximation, which are central to theory for fully-observed Block MDPs [82, 47, 61].

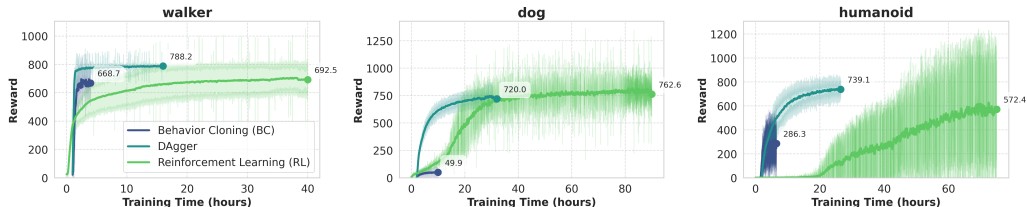

Figure 1: The performance of (offline/online) expert distillation and RL with respect to wall-clock time. We repeat each experiment 5 times and plot the mean and standard deviation. For the time complexity of BC, we include the data collection time, and amortize it over the training steps. For both BC and DAgger, we include the time to train the latent expert (also amortized).

While prior belief contraction results [24] apply to this model, they only yield contraction by $1 - (1 - 2\delta)/C$ per frame, for a large constant $C > 1$ (Remark D.1), and so are vacuous for $L = o(\log S)$, even in the regime $\delta \ll 1$ (i.e. low observation noise). Theorem 3.2 remedies this limitation; e.g. for $\delta = 1/S$ it yields $\varepsilon_h^{\text{contract}}(\pi; L) \leq \mathcal{O}(1/S)$ with only $L = \mathcal{O}(1)$. To prove Theorem 3.2, one might hope that each new observation contracts the TV-distance by $\text{poly}(\delta)$ in expectation. This is false (Example D.1), but in such cases, it turns out that the density ratio decays, yielding a win-win argument.

Heuristically, Theorem 3.2 suggests that standard RL with $L$-step frame-stacking should progressively improve as $L$ increases. Formally, Theorem 3.2 and Theorem 3.1 imply the following end-to-end learning guarantee for the RL algorithm of [23] (which does not use latent state information):

**Corollary 3.1** (Informal; see Corollary F.1). *There is a method that, for any $\delta$-perturbed Block MDP, learns a policy $\widehat{\pi}$ with $J(\pi^\star) - J(\widehat{\pi}) \leq (C_{3.2}\delta)^{L/9}(SXH)^{\mathcal{O}(1)}$ in time $(XA/\delta)^{\mathcal{O}(L)}(HS)^{\mathcal{O}(1)}$.*

From a theoretical view, it remains to understand the decodability error for the perturbed Block MDP. As we will show, this qualitatively depends on the stochasticity of the transition dynamics.

## 4 Distillation is Competitive for Deterministic Dynamics

In some environments, it is reasonable to assume that the latent transition dynamics are *deterministic* (e.g., if the dynamics are governed by simple Newtonian mechanics). Simulation benchmarks with this property include some Atari games as well as MuJoCo tasks. In this section, we theoretically and empirically study the performance of expert distillation, versus standard RL with frame-stacking, in such environments (with deterministic latent transitions, but stochastic initial state and observations).

### 4.1 Theoretical Analysis under Deterministic Dynamics

Below, we show that for perturbed Block MDPs with deterministic dynamics, the decodability error decays exponentially as the step $h \in [H]$ increases. Intuitively, each observation concentrates the true belief state further, and the deterministic transitions cannot "spread out" the belief state. While this intuition is not quite rigorous, it can be proven that *most* observations concentrate the belief state; the result follows from an appropriate martingale analysis (Lemma C.5).

**Proposition 4.1** (See Proposition D.1). *There is a universal constant $C_{4.1} > 1$ so that the following holds. Suppose that $\mathcal{P}$ is a $\delta$-perturbed Block MDP with deterministic transitions. For any executable policy $\pi$ and index $h \in [H]$, it holds that $\varepsilon_h^{\text{decode}}(\pi) \leq \min(\delta, (C_{4.1}\delta)^{(h-1)/9})$.*

From Lemma 3.1, the "ideal" distillation of a latent expert $\pi^{\text{latent}}$ is $\pi^{\text{imitation}} := \pi^{\text{latent}} \circ \mathbf{b}$, i.e., given any history, query the latent expert based on the true belief. Combining Lemma 3.1 and Proposition 4.1 immediately yields a strong, horizon-independent guarantee for this policy: if $\pi^{\text{latent}}$ is the optimal latent policy, then

$$J(\pi^\star) - J(\pi^{\text{imitation}}) \leq J(\pi^{\text{latent}}) - J(\pi^{\text{imitation}}) \leq 2\sum_{h=1}^{H} \min(\delta, (C_{4.1}\delta)^{(h-1)/9}) \leq \mathcal{O}(\delta),$$

where the first inequality is by Lemma C.4. Of course, exactly learning the true belief state may be unrealistic, since this would require conditioning on the entire history. However, we can prove that (a

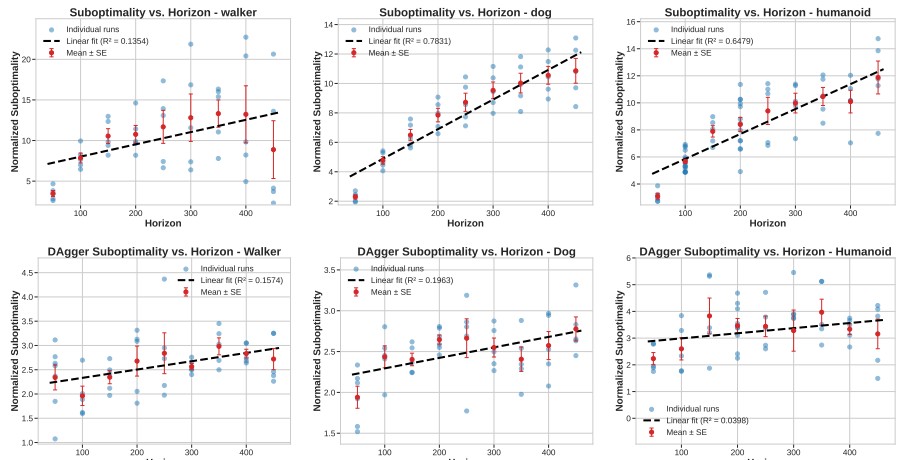

Figure 2: The normalized suboptimality of the expert distillation algorithms (top: behavior cloning; bottom: `DAgger`) with respect to the horizon. We repeat 5 runs for each horizon and task, and perform linear regression on the results from each task. Note that the trajectory rewards for this plot have been normalized by horizon (and by action-prediction error), so linear scaling indicates compounding errors.

slight modification of) the `Forward` algorithm [62] (the non-stationary version of `DAgger`) on $L$-step executable policies learns the following approximation of $\pi^{\text{imitation}}$,[5] in the infinite-sample limit:

$$\pi_h^{\text{Forward}}(\cdot \mid x_{h-L+1:h}, a_{h-L:h-1}) = \begin{cases} \pi_h^{\text{latent}} \circ \mathbf{b}_h^{\text{apx}}(x_{h-L+1:h}, a_{h-L:h-1}; d_{h-L}^{\pi^{\text{Forward}}}) & \text{if } h > L \\ \pi_h^{\text{latent}} \circ \mathbf{b}_h(x_{1:h}, a_{1:h-1}) & \text{otherwise} \end{cases}$$

See Appendix E.2 for the algorithm and proof. Applying this derivation to Lemma 3.1, then using Proposition 4.1 to bound the decodability error and Theorem 3.2 to bound the error in approximate beliefs, gives the following guarantee for expert distillation under deterministic latent dynamics:

**Theorem 4.1** (See Theorem E.1). *Suppose that the POMDP $\mathcal{P}$ is a $\delta$-perturbed Block MDP with deterministic transitions, and fix $L \in \mathbb{N}$. Let $\pi^{\text{latent}} \in \Pi^{\text{latent}}$ be the optimal latent policy, and let $\pi^{\text{Forward}}$ be the policy computed by `Forward` with policy class $\Pi^L$ (i.e. all $L$-step executable policies) and expert $\pi^{\text{latent}}$, in the infinite-sample limit. Then*

$$J(\pi^\star) - J(\pi^{\text{Forward}}) \leq J(\pi^{\text{latent}}) - J(\pi^{\text{Forward}}) \leq \mathsf{TV}(\mathbb{P}^{\pi^{\text{latent}}}, \mathbb{P}^{\widetilde{\pi}}) \leq \mathcal{O}(\delta) + (C_{D.1}\delta)^{L/9} SH.$$

**Comparison with RL.** While Theorem 4.1 is presented in the infinite-sample limit, the effective sample complexity is only $\approx (XA)^{\mathcal{O}(L)}$, since the optimization is over $L$-step executable policies. More concretely, up to additional error $\varepsilon_{\text{opt}}$, the above guarantee can be achieved by the same algorithm with only $\text{poly}((AX)^L, H, \varepsilon_{\text{opt}}^{-1})$ time and samples (Theorem E.2). Thus, the guarantee for `Forward` qualitatively matches the guarantee for RL (Corollary 3.1), aside from the additional horizon-independent term of $\mathcal{O}(\delta)$ incurred above (due to poor decodability in initial steps).

### 4.2 Empirical Analysis under Deterministic Dynamics

Theorem 4.1 gives a strong performance guarantee for expert distillation under deterministic latent dynamics, nearly matching that of RL. This suggests that expert distillation may be preferred over standard RL due to its (practical) efficiency. Also, Theorem 4.1 suggests that error may compound with the horizon $H$. However, the result is only an upper bound, and only for a stylized setting. We now investigate whether these two theoretical implications hold up empirically.

**Expert distillation outperforms RL under deterministic dynamics.** In this experiment, we compare the (a) asymptotic performance and (b) computational efficiency of expert distillation and

---

[5]Note that behavior cloning will not learn the same policy, due to the latching effect [72] (i.e. conditioning on past actions of the latent expert). It may nevertheless achieve the same regret bound as `Forward` in our setting: theoretically separating these algorithms likely requires assuming e.g. recoverability [20].

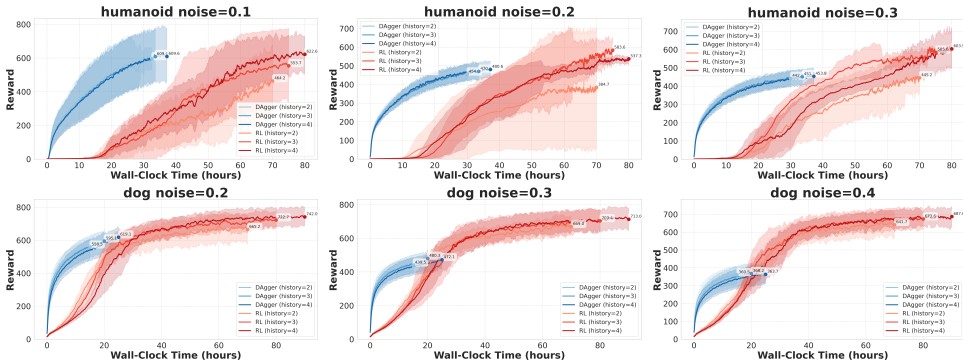

Figure 3: Performance of DAgger and RL with different frame-stacks on humanoid-walk and dog-walk with motor noise. We repeat each experiment 5 times and plot the mean and standard deviation. Note that in general, the improvement of RL over DAgger increases with the motor noise.

standard RL. We train each method until convergence, and we plot the episodic return with respect to the wall clock time in Figure 1. We see that offline expert distillation (i.e., behavior cloning) is competitive in easier tasks such as walker, but is suboptimal in harder tasks such as humanoid and dog. However, online imitation learning (i.e., DAgger) is able to achieve the best performance in all tasks, and with better computational efficiency (i.e., faster convergence) than RL. This supports our theory that under deterministic dynamics, expert distillation can be close to optimal.

**Empirical vignette: the source of error compounding?** The horizon dependence of the error in imitation learning has received intensive empirical [62, 35, 3] and theoretical [59, 20, 60] study, both from the perspective of sample complexity [59, 20] and misspecification [60]. It is widely believed that behavior cloning suffers error compounding over the horizon, which is avoided by online methods such as DAgger that are able to *recover* from mistakes [63, 59]. Does this compounding manifest in expert distillation for POMDPs, and is the cause sampling error or misspecification? In Figure 2, we vary the horizon $H \in [50, 450]$, and measure the sub-optimality of offline and online expert distillation. We normalize rewards so that trajectory reward lies in $[0, 1]$. We further normalize by mean action-prediction MSE (averaged over choice of $H$). We see strong horizon dependence for behavior cloning (and weaker for DAgger, likely due to recoverability). This contrasts with empirical results of [20]: they perform *well-specified* behavior cloning in similar tasks, and find little horizon dependence. Together, our results therefore suggest that misspecification, rather than sampling error, may be the more fundamental source of horizon dependence for behavior cloning.

## 5 RL Outperforms Distillation for Stochastic Dynamics

While deterministic dynamics are plausible in some applications, there are also many potential sources of stochasticity; in real-world robotics, stochasticity may be required to model e.g. internal motor noise or unknowable features of the external environment. Some robotics simulators [44] also have stochasticity arising from a PDE solver. How does the stochasticity of the environment affect the performance of expert distillation and RL?

### 5.1 Theoretical Analysis under Stochastic Dynamics

We show a negative result in the perturbed Block MDP model: for general dynamics, the misspecification of the optimal latent policy with respect to the class of $L$-step executable policies *does not* necessarily decay as $L$ increases, in contrast with the case of deterministic dynamics (Lemma 3.1).

**Proposition 5.1** (See Proposition D.2). *Let $\delta > 0$ and $H \in \mathbb{N}$. There is a $\delta$-perturbed Block MDP $\mathcal{P}$ with horizon $H$ such that for all $L \in [H]$, the optimal latent policy $\pi^{\text{latent}}$ satisfies the following bound, where $\Pi^L$ is the class of $L$-step executable policies:*

$$\min_{\pi \in \Pi^L} \mathsf{TV}(\mathbb{P}^{\pi^{\text{latent}}}, \mathbb{P}^\pi) \geq \Omega(\min(1, \delta H)).$$

This result also highlights the difference between decodability error and belief contraction error, which does decay as $L$ increases, regardless of the transition dynamics (Theorem 3.2). The intuition for Proposition 5.1 is simple: in the extreme case where the dynamics are *uniformly mixing* at every step, prior observations yield no information about the current state, so the $\delta$ error incurred by trying to decode the current observation is irreducible. This decodability error compounds over timesteps, and means that executable policies are unable to simulate the latent policy that plays an action uniquely indexed by the latent state. In contrast, POMDPs with uniform mixing are *easy* for standard RL, precisely because they reduce to $H$ independent horizon-1 subproblems.

**Comparison with RL.** The above result, compared with Corollary 3.1, suggests a potential empirical benefit of standard RL over expert distillation: the former may generically be able to trade increased computation (by increasing $L$) for improved performance (by mitigating observation noise), whereas the latter — at least in the worst case — incurs irreducible error due to stochasticity in the dynamics. To be sure, the uniformly-mixing construction from Proposition 5.1 is practically unrealistic; nevertheless, below we verify that this benefit occurs in more realistic environments.

## 5.2 Experimental Analysis under Stochastic Dynamics

**RL with more computation eventually outperforms distillation.** To simulate a POMDP with stochastic latent dynamics, we apply motor noise in the humanoid-walk task. We add 0-mean isotropic Gaussian noise with std-dev $\in \{0.1, 0.2, 0.3\}$ to each action. We compare DAgger and RL with frame-stack $L \in \{2, 3, 4\}$. We run each method until convergence (with the same number of episodes for all runs with fixed algorithm/noise level) and plot episodic return against wall-clock time (Figure 3). We observe that expert distillation does not benefit from larger $L$, whereas the performance of RL sometimes benefits (at the cost of longer wall-clock time). This improvement is not as dramatic as the theory predicts, perhaps suggesting that there is theoretically unaccounted-for *dependence* between observation errors. Nevertheless, the results do corroborate the main prediction: RL robustly outperforms expert distillation for higher noise levels.

**Empirical vignette: does belief contraction error track RL sub-optimality?** We empirically estimate belief contraction error for each task with no motor noise, and for humanoid-walk with std-dev $= 0.2$. We approximate the (unknown) ground truth belief by training a model $\widehat{\mathbf{b}}^{L^\star}$ that takes $L^\star = 10$ input frames. We compare against models $\widehat{\mathbf{b}}^L$ with $L \in [2, 5]$ input frames. Each model's output belief is parametrized as a multivariate Gaussian distribution with diagonal covariance. All models are trained on the same 2000 trajectories collected by the latent expert policy. For each $L$ we compute the KL-divergence (a tractable proxy for TV-distance) between outputs of $\widehat{\mathbf{b}}^L$ and $\widehat{\mathbf{b}}^{L^\star}$, and average across 100 episodes of validation data, also collected by the same latent expert policy. We find that the empirical error decreases slightly as $L$ increases (Figure 7), though not as fast as the theory predicts.[6] Adding motor noise has little noticeable effect (Figure 8). Interestingly, the error is *not* predictive across tasks: dog-walk has highest empirical error among the three tasks, yet RL achieves the lowest sub-optimality on it (Figure 1), indicating a theoretically-unexplained confounder.

# 6 Towards Better Distillation: Imitating a Smoother Expert

In this section, we discuss how the bounds via approximate decodability (e.g., Lemma 3.1) are loose since they fail to capture the *smoothness* of the latent expert. A tighter bound with smoothness suggests potential benefits of artificially smoothing the latent expert before distillation. We then propose a broadly-applicable method for improving the smoothness, and show that it yields empirical benefits. We view these results as largely a proof-of-concept and leave more detailed investigation to future work.

**Smoothness of the latent policy.** Suppose that the true belief state at some step is always uniform over two particular states $\{s, s'\}$. Then decodability error is large, and a worst-case latent policy $\pi^{\text{latent}}$ — namely, one that plays different actions on these states — is unavoidably misspecified with respect to the class of executable policies. However, ambiguity between $s$ and $s'$ is most likely to occur if these states are somehow *similar* (e.g., close w.r.t. a metric). If $\pi^{\text{latent}}$ is *smooth* in the sense that it plays similar action distributions for nearby states, then the misspecification should be mitigated.

---

[6]Note that for $\gamma$-observable POMDPs, KL-divergence is also predicted to decay as $L$ increases [24].

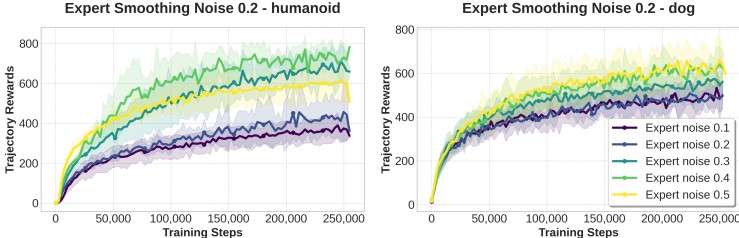

Figure 4: Performance of `DAgger` on the validation dataset for the `humanoid-walk` and `dog-walk` environments with motor noise $\sigma = 0.2$, as the noise level for the training environment (i.e. the environment in which the latent expert was trained) varies over $\{0.1, 0.2, 0.3, 0.4, 0.5\}$.

This phenomenon can be captured more generally by the following variant of Definition 3.1, which measures decodability error of the *actions* (and hence is adaptive to the latent expert):

**Definition 6.1** (Action-prediction error). *Fix a latent policy $\pi^{\mathsf{latent}}$. For a fixed executable policy $\pi$, and timestep $h$, the* action-prediction error *is defined as*

$$\varepsilon_h^{\mathsf{act};\pi^{\mathsf{latent}}}(\pi) = \mathbb{E}^\pi\big[1 - \big\|\pi^{\mathsf{latent}} \circ \mathbf{b}_h(x_{1:h}, a_{1:h-1})\big\|_\infty\big].$$

In Lemma 3.1, the decodability error can indeed be replaced by the action-prediction error — see Lemma E.4. Note that so long as $\pi^{\mathsf{latent}}$ is deterministic (which is without loss of generality for the optimal latent policy), it generically holds that $\varepsilon_h^{\mathsf{act};\pi^{\mathsf{latent}}}(\pi) \leq \varepsilon_h^{\mathsf{decode}}(\pi)$.

**Algorithmic intervention: smoothing experts with motor noise.** One way to construct a smoother expert policy is to pre- or post-compose the optimal latent policy at each step with e.g. a Gaussian convolution kernel (on the state or action space, respectively). However, such approaches ignore the sequential nature of decision-making: smoothing the policy at later steps means that earlier actions may no longer be optimal. We propose instead computing the optimal policy for a *modified* latent MDP with *additional motor noise*. This encourages robustness to motor noise, as a tractable proxy for robustness to observation noise—see Appendix G for an example of one potential mechanism by which the former may lead to the latter.

**Experimental results.** For both `humanoid-walk` and `dog-walk`, for each $\sigma \in \{0.1, 0.2, 0.3, 0.4, 0.5\}$, we train an expert latent policy $\pi^\sigma$ in the environment with mean-0, std. dev.-$\sigma$, Gaussian motor noise on each action. We distill each expert to an executable policy via `DAgger` in an environment with $\sigma = 0.2$. We observe that $\pi^{0.2}$ incurs worse estimated action-prediction error than some higher-noise experts (Figure 9). Moreover, despite being the optimal latent policy for this environment, it is *not* the best expert to distill (Figure 4): the distillations of policies with lower action-prediction error achieve higher reward (substantially for `humanoid-walk` and modestly for `dog-walk`). We also observe that the effect disappears when the true environment has deterministic dynamics (Appendix H.2), likely since it is near-decodable.

**Related methods.** We view this method as a lightweight version of asymmetric RL methods that iteratively refine the expert [78]. It is also closely related to the principle of noise injection in imitation learning [35, 4], which has been shown to robustify Behavior Cloning—to *match* the performance of `DAgger`—by mitigating out-of-distribution effects.[7] Figure 4 demonstrates that even though `DAgger` uses online data collection to mitigate out-of-distribution effects, noise injection can still *improve* its performance in challenging image-based tasks—thus suggesting that there may be a qualitatively different phenomenon at play in highly misspecified settings.

**Limitations and future work.** Our theoretical results are for discrete tabular models with independent observation noise; weakening these assumptions could yield more precise understanding of the fundamental challenges that arise in applications with rich partial observations. Our experiments use synthetic injected motor noise; extending to more natural sources of stochasticity could be valuable. Also, there is a vast design space of algorithmic interventions for smoothing, of which we have only touched the surface. Finally, our work is motivated by applications like robot learning where near-decodability is plausible, but an important problem — which we did not explore — is to understand the algorithmic trade-offs in applications that require active information-gathering.

---

[7]We remark that `DAgger` saturates most of the benchmarks used by [35].

## Acknowledgements

The authors are grateful to Noah Golowich, Audrey Huang, Nan Jiang, Akshay Krishnamurthy, Ankur Moitra, Wen Sun, Gokul Swamy, and Kaiqing Zhang for their insightful discussion. AS and YS acknowledge and thank the support of ONR grant N000142212363 and NSF AI Institute for Societal Decision Making AI-SDM grant IIS2229881. DR is supported by NSF awards CCF-2430381 and DMS-2022448, and ONR grant N00014-22-1-2339.

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

# Contents of Appendix

## A  Additional Related Work

### A.1  Theoretical literature

**Planning and learning in POMDPs.** It is well-known that the *planning* problem in POMDPs (i.e. finding a near-optimal policy given the description of the POMDP) is computationally intractable [57, 39, 6, 54], and the harder *learning* problem (i.e. finding a near-optimal policy given interactive sample access to the POMDP) is also statistically intractable [31], without additional assumptions. In light of these results, there has been recent interest in uncovering natural assumptions that allow statistically or computationally efficient algorithms. On the computational side, [19] introduced the $L$-step decodability assumption, and under this assumption derived a learning algorithm with time complexity $\mathrm{poly}(X^L, A^L, H)$ via frame-stacking. Additionally, [24, 23] derived a quasi-polynomial time algorithm for learning in $\gamma$-observable POMDPs. Computationally efficient learning algorithms are also known for certain classes of POMDPs with deterministic dynamics [28, 76] and certain latent MDPs [33, 34], which are a special case of POMDPs with fixed latent information.

On the statistical side, [28] derived a statistically efficient algorithm for POMDPs satisfying a weak observability condition. Recently, [41, 75, 42, 81] proposed statistically efficient algorithms for POMDPs or Predictive State Representations (PSR) satisfying certain low-rank conditions. More tangential to our work, there has also been increasing interest in off-policy evaluation in POMDPs [74, 83].

**Learning with privileged information in POMDPs.** The most relevant theoretical works to ours are recent works that study the problem of learning POMDPs with latent state information (also called *hindsight observability*) [33, 84, 36, 7]. Of these, [33, 84] are focused on a narrow yet interesting special case of POMDPs called *latent MDPs*, where the unobserved data is fixed and low-dimensional. [36] show that learning in general POMDPs with latent state information is statistically tractable, in contrast with the situation without latent state information. [7] show that with latent

state information the sample complexity of the algorithm for learning $\gamma$-observable POMDPs [23] can be improved from quasi-polynomial to polynomial, though it is an open question whether this is possible without latent state information. Finally, as mentioned earlier, [7] showed that in perfectly decodable POMDPs (Definition B.3), expert distillation yields a fully polynomial time algorithm for learning (for arbitrarily large window size $L$). Since there is a statistical lower bound of $\Omega(A^L)$ in the absence of latent state information [19], this yields a provable computational benefit of latent state information (and, in particular, for expert distillation), but only for perfectly decodable POMDPs.

Compared to the preceding theoretical works, our work seeks both theoretically *and* empirically grounded understanding of the relative merits of expert distillation versus standard reinforcement learning. Among works with similar motivations or results, [12] derive expressions for the output of expert distillation (analogous to Lemma 3.1, except they use a slightly different value-based distillation procedure rather than policy-based) and establish sub-optimality bounds for several imitation learning algorithms. However, they do not instantiate these bounds for concrete models, or theoretically contrast with reinforcement learning. [71] establish provable benefits of imitating the optimal policy in a *fully-observed* MDP (versus learning it via reinforcement learning), but they do not consider partial observability nor the ensuing error due to misspecification. [72] discuss a failure mode of expert distillation in POMDPs when using *offline* imitation learning to distill the latent expert. The "latching" effect that they discuss is due to conditioning on previous actions (see also [66]) — a technical issue that corresponds to why we analyze `Forward` with $L$ random actions — though it is not clear whether this effect is related to the performance gaps between behavior cloning and `DAgger` in our locomotion experiments. Finally, several works [2, 79] give examples of a more fundamental failure mode of expert distillation: in general POMDPs, the optimal policy may need to take *information-gathering actions*. The classical "Tiger Door" exemplifies this failure mode [40].

**Learning with rich observations.** There has been extensive recent interest in reinforcement learning with *rich observations* [31], i.e. where the observation space is too large to enumerate. This line of work has developed largely in parallel with the literature on partial observability, but it is motivated by similar applications as our work (e.g. robotics with image-based perception), and these works formalize the fundamental empirical challenge of representation learning, i.e. "learning to see" [9]. The most well-studied model is the Block MDP [14, 17], which corresponds to perfect decodability with $L = 1$, but is studied in function approximation settings where the observation space is extremely large or infinite, since the problem is computationally easy if the observation space has polynomially-bounded cardinality. While the task of learning in Block MDPs is typically computationally intractable as it inherits the intractable of PAC learning [25], there is by now precise understanding of the computational complexity relative to supervised learning oracles [49, 82, 46, 70, 61].

As observed by [7], there is also a provable computational benefit of latent state information in Block MDPs. Recent works [23, 61] showed that in the absence of latent state information, learning in $\Phi$-decodable Block MDPs (where $\Phi$ is the function approximation class) is strictly harder than the supervised learning task of $\Phi$-decodable *one-context regression*. In contrast, it is straightforward to see that with latent state information and a one-context regression oracle, the true decoding function $\phi^\star \in \Phi$ can be learned up to inverse-polynomial error (on average over any exploratory policy). This function, composed with the optimal latent policy, yields the optimal executable policy. [7] formally proved this result with a slightly different (multi-class classification rather than regression) supervised learning oracle.

## A.2 Empirical literature

**Applied methods that learn with privileged information.** Privileged information has been widely used in training policies for real-world POMDPs, such as in robotics and autonomous driving. The most prominent and successful method is expert distillation [56, 9, 37, 48, 32, 85, 80, 69, 21, 26, 10]. First, one trains an expert policy with access to privileged information — either the latent state in a simulator [9], or observation data from more expensive sensors that will not be available at deployment [56]. Second, one trains an executable policy by performing offline or online imitation learning with respect to the latent policy. [56] also observe empirical benefits of online imitation learning compared to offline, which is corroborated by our results. While there are also notable successes of using RL without privileged information [1, 43], some of the previously-mentioned works observed that RL without privileged information failed to learn locomotion in their environment [37].

Motivated by the theoretical failure modes of expert distillation in the prequel, there is also a line of work in the middle ground between expert distillation and RL without privileged information, that seeks to avoid these failure modes while also improving the convergence of standard RL. These hybrid methods include Asymmetric Actor-Critic [58], and more broadly are described as *asymmetric learning* [58, 79, 78, 55, 77, 67, 45, 52, 38]. While there is some evidence that an algorithm inspired by Asymmetric Actor-Critic may enjoy an improved statistical/computational trade-off for $\gamma$-observable POMDPs [7] (compared to the best-known algorithm that does not use privileged information [23]), the theoretical foundations for these methods remain otherwise largely unexplored.

**Learning with and without privileged information.** In addition to the previously-mentioned ablation experiments [37], recent work of [53] conducted controlled comparisons between expert distillation and standard RL on simulated locomotion and manipulation tasks, with the goal of providing heuristic guidance on when to prefer expert distillation over RL without privileged information. They found that expert distillation converges faster. They also classified tasks as "easy" or "hard" based on the convergence speed of standard RL, and suggested that expert distillation performed better on the "hard" tasks.

**Improvements to expert distillation.** Recall that a key benefit of expert distillation for POMDPs with rich observations (e.g. as found in robotics with image-based perception) was that it avoids performing reinforcement learning on the high-dimensional and complex observation space. In contrast, most of the previously-mentioned works on asymmetric learning use exactly such an algorithm (e.g., as the "Actor" component in Asymmetric Actor-Critic). An exception is the method of [78], which is a variant of expert distillation that iteratively refines the expert with the goal of decreasing misspecification. Our smoothed distillation method (Section 6) can be thought of as a more lightweight approach that refines the expert in one shot. As mentioned in Section 6, it is also similar (though not identical) to several noise injection methods in imitation learning [35, 4].

# B  Additional Preliminaries

## B.1  Belief states

The following operators describe how a belief state evolves as more information is revealed.

**Definition B.1** (Belief state update [24])**.** *For each $h \in \{1, \ldots, H\}$, the* Bayes operator *is* $\mathbb{B}_h : \Delta(\mathcal{S}_h) \times \mathcal{X}_h \to \Delta(\mathcal{S}_h)$ *defined by*

$$\mathbb{B}_h(b; x_h)(s_h) := \frac{\mathbb{O}_h(x_h \mid s_h) b(s_h)}{\sum_{z_h \in \mathcal{S}_h} \mathbb{O}_h(x_h \mid z_h) b(z_h)}.$$

*For each $h \in \{2, \ldots, H\}$, the* belief update operator $\mathbb{U}_h : \Delta(\mathcal{S}_{h-1}) \times \mathcal{A}_{h-1} \times \mathcal{X}_h \to \Delta(\mathcal{S}_h)$*, is defined by* $\mathbb{U}_h(b; a_{h-1}, x_h) := \mathbb{B}_h(\mathbb{P}_h(a_{h-1}) \cdot b; x_h)$ *where* $\mathbb{P}_h(a)$ *denotes the real-valued* $|\mathcal{S}_h| \times |\mathcal{S}_{h-1}|$ *matrix of latent transition probabilities from step $h - 1$ to step $h$ under action $a_{h-1}$.*

The following definition of an approximate belief state is analogous to the inductive definition of a true belief state (Definition 2.1); the only difference is that it updates based on a window of the $L$ most recent observations and actions $(x_{h-L+1:h}, a_{h-L:h-1})$ rather than the entire history, and is additionally parametrized by a distribution $\mathcal{D}$ (which represents the prior on the latent state at step $h - L$).

**Definition B.2.** *For a window length $L > 0$, any $h > L$, and prior $\mathcal{D} \in \Delta(\mathcal{S}_{h-L})$, the* approximate belief state *is inductively defined as*

$$\mathbf{b}_h^{\mathsf{apx}}(x_{h-L+1:h}, a_{h-L:h-1}; \mathcal{D}) := \mathbb{U}_h(\mathbf{b}_{h-1}^{\mathsf{apx}}(x_{h-L+1:h-1}, a_{h-L:h-2}; \mathcal{D}), a_{h-1}, x_h)$$

*where for $L = 0$, $\mathbf{b}_h^{\mathsf{apx}}(\emptyset; \mathcal{D}) := \mathcal{D}$. For $h \leq L$, the approximate belief state is defined to coincide with the true belief state.*

## B.2  Decodability and $\gamma$-Observability

For completeness, we include the definition of (perfect) $L$-step decodability from [19].

**Definition B.3** ($L$-step decodable model [19])**.** *A POMDP is said to be $L$-step decodable if, for each timestep $h \in [H]$, there exists a deterministic mapping $\phi_h : \mathcal{X}^{h-L:h} \times \mathcal{A}^{h-L:h-1} \to \mathcal{S}_h$ such that for any admissible trajectory $\tau = (s, x, a)_{1:h}$ (i.e., a trajectory that occurs with positive probability under the uniformly random policy), we have $s_h = \phi_h(x_{h-L:h}, a_{h-L:h-1})$.*

Next, we introduce relevant definitions and results relating to $\gamma$-observable POMDPs from [24, 23].

**Definition B.4** ($\gamma$-observability [24]). *Let $\gamma \in (0, 1)$. A POMDP is $\gamma$-observable if for any $h \in [H]$ and distributions $b, b' \in \Delta(\mathcal{S}_h)$, it holds that*

$$\left\| \mathbb{O}_h^\top b - \mathbb{O}_h^\top b' \right\|_1 \geq \gamma \|b - b'\|_1,$$

*where $\mathbb{O}_h \in \mathbb{R}^{\mathcal{S}_h \times \mathcal{X}_h}$ is the observation matrix defined by $(\mathbb{O}_h)_{sx} := \mathbb{O}_h(x \mid s)$.*

The algorithm of [23] for learning $\gamma$-observable POMDPs in quasi-polynomial time requires bounding a slightly generalized version of the belief contraction error defined in Definition 3.2. We state this version below.

**Definition B.5** (Generalized belief contraction [23]). *Let $\varepsilon, \phi \in (0, 1)$ and $L \in \mathbb{N}$. We say that a POMDP $\mathcal{P}$ satisfies $(\varepsilon; \phi, L)$-belief contraction if the following property holds. Let $\pi$ be an executable policy, let $h \in \{L + 1, \dots, H\}$, and let $\mathcal{D}, \mathcal{D}' \in \Delta(\mathcal{S}_{h-L})$. If $\left\| \frac{\mathcal{D}'}{\mathcal{D}} \right\|_\infty \leq 1/\phi$, then for any fixed history $(x_{1:h-L}, a_{1:h-L-1})$ it holds that*

$$\mathbb{E}_{s_{h-L} \sim \mathcal{D}'} \mathbb{E}^\pi \left[ \| \mathbf{b}_h^{\mathsf{apx}}(x_{h-L+1:h}, a_{h-L:h-1}; \mathcal{D}') - \mathbf{b}^{\mathsf{apx}}(x_{h-L+1:h}, a_{h-L:h-1}; \mathcal{D}) \|_1 \right] \leq \varepsilon$$

*where the inner expectation is over partial trajectories $(x_{h-L+1:h}, a_{h-L:h-1})$ drawn from $\mathcal{P}$ by initializing to latent state $s_{h-L}$ at step $h - L$, and sampling action $a_k \sim \pi(x_{1:k}, a_{1:k-1})$ at each $h - L \leq k < h$.*

For context, see [23, Theorem 6.2] for the formal statement that $\gamma$-observability implies $(\varepsilon; \phi, L)$-belief contraction with $L \sim \gamma^{-4} \log(1/(\varepsilon\phi))$. Definition 3.2 is a special case of the above definition, with $\mathcal{D}' := \mathbf{b}_{h-L}(x_{1:h-L}, a_{1:h-L-1})$ and $\mathcal{D} := \mathsf{unif}(\mathcal{S}_{h-L})$.

**Theorem B.1** ([23]). *There is a constant $C^\star$ with the following property. Given $\varepsilon, \beta, \gamma > 0$, $L \in \mathbb{N}$, and a $\gamma$-observable POMDP $\mathcal{P}$, set $\phi := \frac{\gamma}{C^\star \cdot H^5 S^{7/2} X^2} \varepsilon$. If $\mathcal{P}$ satisfies $(\varepsilon; \phi, L)$-belief contraction, the algorithm* BaSeCAMP *[23] produces an executable policy $\widehat{\pi}$ that satisfies*

$$J(\pi^\star) - J(\widehat{\pi}) \leq \varepsilon \cdot \mathsf{poly}(S, X, H, \gamma^{-1})$$

*with probability at least $1 - \beta$. Moreover, the time complexity is $\mathsf{poly}((XA)^L, H, S, \varepsilon^{-1}, \gamma^{-1}, \log(\beta^{-1}))$.*

**Proof.** Immediate from inspecting the analysis of BaSeCAMP [23]: while their analysis sets $L := \gamma^{-4} \log(1/(\varepsilon\phi))$, the only place this is used in the proof is to invoke [23, Theorem 6.2] (which is the claim that any $\gamma$-observable POMDP satisfies $(\varepsilon; \phi, L)$-belief contraction with that choice of $L$). Thus, it is sufficient to choose any $L$ for which $(\varepsilon; \phi, L)$-belief contraction holds. $\qquad\square$

## C Technical Lemmas

**Lemma C.1** (Data processing inequality). *Let $\mathcal{S}, \mathcal{T}$ be sets, let $p, q \in \Delta(\mathcal{S})$ be distributions, and let $K : \mathcal{S} \to \Delta(\mathcal{T})$ be a conditional distribution function. Then*

$$\mathsf{TV}(K \circ p, K \circ q) \leq \mathsf{TV}(p, q).$$

*Similarly, if $p \ll q$, then*

$$\left\| \frac{K \circ p}{K \circ q} \right\|_\infty \leq \left\| \frac{p}{q} \right\|_\infty.$$

**Proof.** The first inequality follows from the fact that total variation distance is an $f$-divergence. The second inequality can be directly checked: for all $y \in \mathcal{T}$,

$$(K \circ p)(y) = \sum_{x \in \mathcal{S}} K(y \mid x) p(x) \leq \left\| \frac{p}{q} \right\|_\infty \sum_{x \in \mathcal{S}} K(y \mid x) q(x) = \left\| \frac{p}{q} \right\|_\infty (K \circ q)(y)$$

as needed. $\qquad\square$

Recall that a policy is *executable* if the action distribution at any step is determined by the action/observation history (note that a latent policy therefore may not be executable). The following lemma, which was implicitly used in prior work [24], verifies under any executable policy, the conditional distribution of the latent state given the history is the true belief state.[8]

**Lemma C.2.** *Fix any step $h \in [H]$ and executable policy $\pi$. Then*

$$\mathbb{P}^\pi[s_h \mid x_{1:h}, a_{1:h-1}] = \mathbf{b}_h(x_{1:h}, a_{1:h-1})(s_h)$$

*and, if $h > 1$,*

$$\mathbb{P}^\pi[x_h \mid x_{1:h-1}, a_{1:h-1}] = (\mathbb{O}_h^\top \cdot \mathbb{P}_h(a_{h-1}) \cdot \mathbf{b}_{h-1}(x_{1:h-1}, a_{1:h-2}))(x_h)$$

*for any action/observation history $(x_{1:h}, a_{1:h-1})$ and latent state $s_h$.*

**Proof.** We prove the first claim by induction on $h$. It is clear that $\mathbb{P}^\pi[s_1 \mid x_1] \propto \mathbb{O}_1(x_1 \mid s_1)\mathbb{P}[s_1] \propto \mathbb{B}_1(\mathbb{P}_1; x_1) = \mathbf{b}_1(x_1)(s_1)$, where proportionality is up to factors independent of $s_1$. Since $\mathbb{P}^\pi[\cdot \mid x_1]$ and $\mathbf{b}_1(x_1)$ are distributions, it follows from the proportionality that they are equal. Now fix any $h \in \{2, \ldots, H\}$ and assume the claim holds for $h-1$. Let $(s_{1:h}, x_{1:h}, a_{1:h-1})$ be a random trajectory drawn from $\mathbb{P}^\pi$, i.e. generated via $s_h \sim \mathbb{P}_h(s_{h-1}, a_{h-1})$, $x_h \sim \mathbb{O}_h(s_h)$, and $a_h \sim \pi(x_{1:h}, a_{1:h-1})$ (since we assumed that $\pi$ is executable, the action distribution does not directly depend on $s_{1:h}$). Then,

$$\mathbb{P}^\pi[s_h \mid x_{1:h}, a_{1:h-1}] \propto \sum_{s_{1:h-1}} \mathbb{P}^\pi[s_{1:h}, x_{1:h}, a_{1:h-1}]$$

$$= \sum_{s_{1:h-1}} \mathbb{P}^\pi[s_{1:h-1}, x_{1:h-1}, a_{1:h-2}]\pi(x_{1:h-1}, a_{1:h-2})\mathbb{P}[s_h \mid s_{h-1}, a_{h-1}]\mathbb{O}_h(x_h \mid s_h)$$

$$\propto \sum_{s_{1:h-1}} \mathbb{P}^\pi[s_{1:h-1}, x_{1:h-1}, a_{1:h-2}]\mathbb{P}[s_h \mid s_{h-1}, a_{h-1}]\mathbb{O}_h(x_h \mid s_h)$$

$$= \mathbb{O}_h(x_h \mid s_h) \sum_{s_{h-1}} \mathbb{P}[s_h \mid s_{h-1}, a_{h-1}] \sum_{s_{1:h-2}} \mathbb{P}^\pi[s_{1:h-1}, x_{1:h-1}, a_{1:h-2}]$$

$$\propto \mathbb{O}_h(x_h \mid s_h) \sum_{s_{h-1}} \mathbb{P}[s_h \mid s_{h-1}, a_{h-1}]\mathbb{P}^\pi[s_{h-1} \mid x_{1:h-1}, a_{1:h-2}]$$

$$= \mathbb{O}_h(x_h \mid s_h) \sum_{s_{h-1}} \mathbb{P}[s_h \mid s_{h-1}, a_{h-1}]\mathbf{b}_{h-1}(x_{1:h-1}, a_{1:h-2})(s_{h-1})$$

$$\propto \mathbf{b}_h(x_{1:h}, a_{1:h-1})(s_h)$$

where the penultimate equality uses the induction hypothesis and the final equality uses Eq. (1). This proves the first claim. To prove the second claim, observe that by a similar argument to above, for any $h > 1$,

$$\mathbb{P}^\pi[s_h \mid x_{1:h-1}, a_{1:h-1}] \propto \sum_{s_{h-1}} \mathbb{P}[s_h \mid s_{h-1}, a_{h-1}]\mathbf{b}_{h-1}(x_{1:h-1}, a_{1:h-2})(s_{h-1})$$

so that $\mathbb{P}^\pi[s_h \mid x_{1:h-1}, a_{1:h-1}] = (\mathbb{P}_h(a_{h-1}) \cdot \mathbf{b}_{h-1}(x_{1:h-1}, a_{1:h-2}))(s_h)$. But then

$$\mathbb{P}^\pi[x_h \mid x_{1:h-1}, a_{1:h-1}] \propto \sum_{s_{1:h}} \mathbb{P}^\pi[s_{1:h}, x_{1:h}, a_{1:h-1}]$$

$$= \sum_{s_h} \mathbb{O}_h(x_h \mid s_h) \sum_{s_{1:h-1}} \mathbb{P}^\pi[s_{1:h}, x_{1:h-1}, a_{1:h-1}]$$

$$\propto \sum_{s_h} \mathbb{O}_h(x_h \mid s_h)\mathbb{P}^\pi[s_h \mid x_{1:h-1}, a_{1:h-1}].$$

Therefore $\mathbb{P}^\pi[x_h \mid x_{1:h-1}, a_{1:h-1}] = (\mathbb{O}_h^\top \cdot \mathbb{P}_h(a_{h-1}) \cdot \mathbf{b}_{h-1}(x_{1:h-1}, a_{1:h-2}))(x_h)$ as claimed. □

We will also need the following variant of Lemma C.2, which shows how approximate belief states arise as conditional probability distributions:

---

[8]Note that this is not true for all policies, since a latent policy could reveal the latent state (or, more generally, bias the conditional distribution) by its choice of action. This issue underpins the "latching" effect observed in behavior cloning of privileged experts [72].

**Lemma C.3.** *Fix any $h \in [H]$, $L \in \{0, \dots, h-1\}$, and executable policy $\pi$ where $\pi_{h-t}(\cdot \mid x_{1:h-t}, a_{1:h-t-1})$ is determined by $(x_{h-L+1:h-t}, a_{h-L:h-t-1})$ for all $t \in [L]$. Then*

$$\mathbb{P}^\pi[s_h \mid x_{h-L+1:h}, a_{h-L:h-1}] = \mathbf{b}_h^{\mathsf{apx}}(x_{h-L+1:h}, a_{h-L:h-1}; d_{h-L}^\pi).$$

**Proof.** We induct on $L$. If $L = 0$ then, for any $h \in [H]$, by definition $\mathbb{P}^\pi[s_h] = d_h^\pi(s_h) = \mathbf{b}_h^{\mathsf{apx}}(\emptyset; d_h^\pi)(s_h)$. Fix any $L > 0$ and suppose the claim holds for $L - 1$ (for all $h > L - 1$). Then for any $h > L$,

$$\mathbb{P}^\pi[s_h \mid x_{h-L+1:h}, a_{h-L:h-1}]$$

$$\propto \sum_{s_{h-L:h-1}} \mathbb{P}^\pi[s_{h-L:h}, x_{h-L+1:h}, a_{h-L:h-1}]$$

$$= \sum_{s_{h-L:h-1}} \mathbb{P}^\pi[s_{h-L:h-1}, x_{h-L+1:h-1}, a_{h-L:h-2}] \pi(a_{h-1} \mid x_{h-L+1:h-1}, a_{h-L:h-2}) \mathbb{P}_h[s_h \mid s_{h-1}, a_{h-1}] \mathbb{O}_h(x_h \mid s_h)$$

$$\propto \sum_{s_{h-L:h-1}} \mathbb{P}^\pi[s_{h-L:h-1}, x_{h-L+1:h-1}, a_{h-L:h-2}] \mathbb{P}_h[s_h \mid s_{h-1}, a_{h-1}] \mathbb{O}_h(x_h \mid s_h)$$

$$= \mathbb{O}_h(x_h \mid s_h) \sum_{s_{h-1}} \mathbb{P}_h[s_h \mid s_{h-1}, a_{h-1}] \mathbb{P}^\pi[s_{h-1}, x_{h-L+1:h-1}, a_{h-L:h-2}]$$

$$\propto \mathbb{O}_h(x_h \mid s_h) \sum_{s_{h-1}} \mathbb{P}_h[s_h \mid s_{h-1}, a_{h-1}] \mathbb{P}^\pi[s_{h-1} \mid x_{h-L+1:h-1}, a_{h-L:h-2}]$$

$$= \mathbb{O}_h(x_h \mid s_h) \sum_{s_{h-1}} \mathbb{P}_h[s_h \mid s_{h-1}, a_{h-1}] \mathbf{b}_{h-1}^{\mathsf{apx}}(x_{h-L+1:h-1}, a_{h-L:h-2}; d_{h-L}^\pi)(s_{h-1})$$

$$\propto \mathbf{b}_h^{\mathsf{apx}}(x_{h-L+1:h}, a_{h-L:h-1}; d_{h-L}^\pi)(s_h)$$

by the induction hypothesis and the definition of $\mathbf{b}_h^{\mathsf{apx}}(x_{h-L+1:h}, a_{h-L:h-1}; d_{h-L}^\pi)$. $\qquad\square$

The following fact is well-known.

**Lemma C.4.** *Let $\pi^\star$ be the optimal policy of the POMDP, and let $\pi^{\mathsf{latent}}$ be the optimal policy of the MDP. Then we have that*

$$J(\pi^\star) \leq J(\pi^{\mathsf{latent}}).$$

**Proof.** We prove by proving a more general result. Consider any POMDP $\mathcal{P}$ and its corresponding MDP $\mathcal{M}$, for any latent policy $\pi^{\mathsf{latent}}$, we use $Q_h^{\mathcal{M};\pi^{\mathsf{latent}}} : \mathcal{S}_h \times \mathcal{A}_h \to [0, 1]$ to denote the Q-value of following $\pi^{\mathsf{latent}}$ in the MDP at timestep $h$. We use $Q_h^{\mathcal{P};\pi} : \mathcal{X}^{1:h} \times \mathcal{A}^{1:h} \to [0, 1]$ to denote the Q-value at timestep $h$ of following executable policy $\pi$. We will use $Q^{\mathcal{P}}$ and $Q^{\mathcal{M}}$ to denote the optimal POMDP Q-value and optimal MDP Q-value functions. Note that $Q^{\mathcal{P}}$ satisfies the following optimality equation, for any $x_{1:h}, a_{1:h}$, we have

$$Q_h^{\mathcal{P}}((x_{1:h}, a_{1:h-1}), a_h) = \sum_{s_h} \mathbf{b}_h(x_{1:h}, a_{1:h-1}) R_h(s_h, a_h) + \sum_{x_{h+1}} P(x_{h+1} \mid x_{1:h}, a_{1:h}) \max_{a_{h+1}} Q_h^{\mathcal{P}}((x_{1:h+1}, a_{1:h}), a_{h+1}),$$

where $P(x_{h+1} \mid x_{1:h}, a_{1:h}) = \sum_{s_h, s_{h+1}} \mathbf{b}_h(s_h \mid x_{1:h}, a_{1:h}) \mathbb{P}_h(s_{h+1} \mid s_h, a_h) \mathbb{O}_{h+1}(x_{h+1} \mid s_{h+1})$. Note that in this case, $x_{1:h}, a_{h-1}$ can be summarize as $\mathbf{b}_h(x_{1:h}, a_{h-1})$; and thus given any belief $b_h \in \Delta(\mathcal{S}_h)$, we will abuse the notation and define

$$Q_h^{\mathcal{P}}(b_h, a_h) = \sum_{s_h} b_h(s_h) R_h(s_h, a_h) + \sum_{x_{h+1}} P(x_{h+1} \mid b_h, a_h) \max_{a_{h+1}} Q_h^{\mathcal{P}}(b_{h+1}, a_{h+1}),$$

where $b_{h+1} := \mathbb{U}_{h+1}(b_h; a_h, x_{h+1})$. Similarly, we can define

$$Q_h^{\mathcal{M}}(b_h, a_h) = \sum_{s_h} b_h(s_h) Q^{\mathcal{M}}(s_h, a_h).$$

Note that in this case, let $\pi^\star$ be the optimal executable policy, and let $\pi^{\mathsf{latent}}$ be the optimal MDP policy, we have that

$$J(\pi^\star) = \mathbb{E}_{x_1}\left[Q_1^{\mathcal{P}}(\mathbf{b}_1(x_1), \pi^\star(x_1))\right],$$

and
$$J(\pi^{\text{latent}}) = \mathbb{E}_{s_1}[Q_1^{\mathcal{M}}(s_1, \pi^{\text{latent}}(s_1))].$$

In the following we will prove that, for any timestep $h \in [H]$, for any admissible belief $b_h \in \Delta(\mathcal{S}_h)$, fix action $a_h$, we have that
$$Q_h^{\mathcal{M}}(b_h, a_h) \geq Q_h^{\mathcal{P}}(b_h, a_h).$$

We proceed with induction. For $h = H$, we have
$$Q_H^{\mathcal{M}}(b_H, a_H) = \sum_{s_H} b_H(s_H) R_H(s_H, a_H) = Q_H^{\mathcal{P}}(b_H, a_H).$$

Then assuming $Q_{h+1}^{\mathcal{M}}(b_{h+1}, a_{h+1}) \geq Q_h^{\mathcal{P}}(b_{h+1}, a_{h+1})$ for any admissible $b_{h+1}$, we have for any admissible $b_h$ and action $a_h$,

$$\begin{aligned}
Q_h^{\mathcal{P}}(b_h, a_h) &= \sum_{s_h} b_h(s_h) R_h(s_h, a_h) + \sum_{x_{h+1}} P(x_{h+1} \mid b_h, a_h) \max_{a_{h+1}} Q_{h+1}^{\mathcal{P}}(b_{h+1}, a_{h+1}) \\
&\leq \sum_{s_h} b_h(s_h) R_h(s_h, a_h) + \sum_{x_{h+1}} P(x_{h+1} \mid b_h, a_h) \max_{a_{h+1}} Q_{h+1}^{\mathcal{M}}(b_{h+1}, a_{h+1}) \\
&\leq \sum_{s_h} b_h(s_h) R_h(s_h, a_h) + \sum_{x_{h+1}} P(x_{h+1} \mid b_h, a_h) \left( \sum_{s_{h+1}} b_{h+1}(s_{h+1}) \max_{a_{h+1}} Q_{h+1}^{\mathcal{M}}(s_{h+1}, a_{h+1}) \right).
\end{aligned}$$

For any function $f$ that only depend on the state, we have

$$\begin{aligned}
& \sum_{x_{h+1}} P(x_{h+1} \mid b_h, a_h) \left( \sum_{s_{h+1}} b_{h+1}(s_{h+1}) f(s_{h+1}) \right) \\
&= \sum_{x_{h+1}} P(x_{h+1} \mid b_h, a_h) \left( \sum_{s_{h+1}} \left( \frac{\mathbb{O}_h(x_{h+1} \mid s_{h+1}) \sum_{s_h} \mathbb{P}_h(s_{h+1} \mid s_h, a_h) b_h(s_h)}{P(x_{h+1} \mid b_h, a_h)} \right) f(s_{h+1}) \right) \\
&= \sum_{x_{h+1}} \sum_{s_{h+1}} \mathbb{O}_{h+1}(x_{h+1} \mid s_{h+1}) \sum_{s_h} \mathbb{P}_h(s_{h+1} \mid s_h, a_h) b_h(s_h) f(s_{h+1}) \\
&= \sum_{s_h, s_{h+1}} \mathbb{P}_h(s_{h+1} \mid s_h, a_h) b_h(s_h) f(s_{h+1}).
\end{aligned}$$

This gives that

$$\begin{aligned}
Q_h^{\mathcal{P}}(b_h, a_h) &\leq \sum_{s_h} b_h(s_h) R_h(s_h, a_h) + \sum_{s_h, s_{h+1}} \mathbb{P}_h(s_{h+1} \mid s_h, a_h) b_h(s_h) \max_{a_{h+1}} Q_{h+1}^{\mathcal{M}}(s_{h+1}, a_{h+1}) \\
&= \sum_{s_h} b_h(s_h) \left( R_h(s_h, a_h) + \sum_{s_{h+1}} \mathbb{P}_h(s_{h+1} \mid s_h, a_h) \max_{a_{h+1}} Q_{h+1}^{\mathcal{M}}(s_{h+1}, a_{h+1}) \right) \\
&= \sum_{s_h} b_h(s_h) Q_h^{\mathcal{M}}(s_h, a_h) \\
&= Q_h^{\mathcal{M}}(b_h, a_h).
\end{aligned}$$

Finally, we conclude the proof by noting that

$$J(\pi^{\text{latent}}) = \mathbb{E}_{s_1}[Q_1^{\mathcal{M}}(s_1, \pi^{\text{latent}}(s_1))] = \mathbb{E}_{s_1}[\max_{a_1} Q_1^{\mathcal{M}}(s_1, a_1)] \geq \max_{a_1} Q_1^{\mathcal{M}}(\mathbf{b}_1, a_1) \geq \max_{a_1} Q_1^{\mathcal{P}}(\mathbf{b}_1, a_1) = J(\pi^\star).$$

$\square$

We will need the following martingale bound to analyze belief contraction error and decodability error in the perturbed Block MDP.

**Lemma C.5.** *Fix $\varepsilon \in (0, 3^{-6})$ and $S > 0$. Let $X_0, \ldots, X_L$ be a non-negative supermartingale with $\mathbb{E}[X_0] \leq S$ and $\Pr[X_{n+1} > \varepsilon X_n | X_n] \leq \varepsilon$ almost surely for all $0 \leq n < L$. Then*

$$\mathbb{E}[\min(X_L, S)] \leq 2 \cdot 3^L \varepsilon^{L/3} S.$$

**Proof.** For any integer $0 \leq n \leq L$ and any integer $k$, define $f(n,k) := \Pr[\varepsilon^{k+1} S < X_n \leq \varepsilon^k S]$. We prove by induction on $n$ that $f(n,k) \leq 3^n \varepsilon^{(n-k)/3}$. If $n = 0$, then the claim is trivially true for $k \geq 0$ since $f(n,k) \leq 1$ always. For any $k < 0$, by Markov's inequality,

$$f(0,k) \leq \Pr[X_0 > \varepsilon^{k+1} S] \leq \frac{\mathbb{E}[X_0]}{\varepsilon^{k+1} S} = \varepsilon^{-k-1} \leq \varepsilon^{-k/3}.$$

For any $0 < n \leq L$ and integer $k$, we have

$$f(n,k) = \sum_{\ell=-\infty}^{\infty} \Pr[\varepsilon^{k+1} S < X_n \leq \varepsilon^k S \mid \varepsilon^{\ell+1} S < X_{n-1} \leq \varepsilon^\ell S] \cdot f(n-1, \ell)$$

$$\leq \sum_{\ell=-\infty}^{k-1} f(n-1, \ell) + \varepsilon f(n-1, k) + \varepsilon f(n-1, k+1) + \sum_{\ell=k+2}^{\infty} \varepsilon^{\ell-k-1} f(n-1, \ell) \quad (4)$$

where the inequality uses the following two facts. First, for any $\ell \geq k$,

$$\Pr[\varepsilon^{k+1} S < X_n \leq \varepsilon^k S \mid \varepsilon^{\ell+1} S < X_{n-1} \leq \varepsilon^\ell S] \leq \Pr[X_n > \varepsilon X_{n-1} \mid \varepsilon^{\ell+1} S < X_{n-1} \leq \varepsilon^\ell S] \leq \varepsilon$$

by lemma assumption. Second, for any $\ell \geq k+2$,

$$\Pr[\varepsilon^{k+1} S < X_n \leq \varepsilon^k S \mid \varepsilon^{\ell+1} S < X_{n-1} \leq \varepsilon^\ell S] \leq \Pr[X_n > \varepsilon^{k+1-\ell} X_{n-1} \mid \varepsilon^{\ell+1} S < X_{n-1} \leq \varepsilon^\ell S]$$

$$\leq \varepsilon^{\ell-k-1}$$

since $X_0, \ldots, X_L$ is a supermartingale. Returning to Eq. (4), we get

$$f(n,k) \leq \sum_{\ell=-\infty}^{k-1} f(n-1, \ell) + \varepsilon f(n-1, k) + \varepsilon f(n-1, k+1) + \sum_{\ell=k+2}^{\infty} \varepsilon^{\ell-k-1} f(n-1, \ell)$$

$$\leq \sum_{\ell=-\infty}^{k-1} 3^{n-1} \varepsilon^{(n-1-\ell)/3} + 3^{n-1} \varepsilon^{1+(n-k-1)/3} + 3^{n-1} \varepsilon^{1+(n-k-2)/3} + \sum_{\ell=k+2}^{\infty} 3^{n-1} \varepsilon^{\ell-k-1+(n-\ell-1)/3}$$

$$\leq 3^{n-1} \varepsilon^{(n-k)/3} \left( \frac{1}{1-\varepsilon^{1/3}} + \varepsilon^{2/3} + \varepsilon^{1/3} + \frac{1}{1-\varepsilon^{2/3}} \right)$$

$$\leq 3^n \varepsilon^{(n-k)/3}$$

where the final inequality holds since $\varepsilon \leq 1/64$. This completes the induction. Next,

$$\mathbb{E}[\min(X_L, S)] \leq \sum_{\ell=-\infty}^{-1} S \cdot f(L, \ell) + \sum_{\ell=0}^{\infty} \varepsilon^\ell S \cdot f(L, \ell)$$

$$\leq 3^L S \cdot \left( \sum_{\ell=-\infty}^{-1} \varepsilon^{(L-\ell)/3} + \sum_{\ell=0}^{\infty} \varepsilon^{\ell+(L-\ell)/3} \right)$$

$$\leq 3^L S \cdot \left( \frac{\varepsilon^{(L+1)/3}}{1-\varepsilon^{1/3}} + \frac{\varepsilon^{L/3}}{1-\varepsilon^{2/3}} \right)$$

$$\leq 3^L S \cdot 2\varepsilon^{L/3}$$

as claimed. $\qquad\square$

# D   Omitted Proofs for Perturbed Block MDP

Below, we restate the definition of a $\delta$-perturbed Block MDP. We then prove our main theoretical results for the perturbed Block MDP. In Appendix D.1, we prove Theorem 3.2 (the belief contraction result, restated as Theorem D.1). In Appendix D.2, we prove Proposition 4.1 (the decodability result for deterministic dynamics, restated as Proposition D.1). Finally, in Appendix D.3, we prove Proposition 5.1 (the misspecification lower bound for stochastic dynamics, restated as Proposition D.2).

**Definition D.1.** *Fix a parameter $\delta > 0$. A POMDP $\mathcal{P}$ is a $\delta$-perturbed Block MDP if, for each $h \in [H]$, there are $\widetilde{\mathbb{O}}_h, E_h : \mathcal{S}_h \to \Delta(\mathcal{X}_h)$ such that $\widetilde{\mathbb{O}}_h : \mathcal{S}_h \to \Delta(\mathcal{X}_h)$ satisfies the* block property [17]*, i.e. $\widetilde{\mathbb{O}}_h(\cdot \mid s_h), \widetilde{\mathbb{O}}_h(\cdot \mid s'_h)$ have disjoint supports for all $s_h \neq s'_h$, and moreover the emission distribution $\mathbb{O}_h$ at step $h$ can be decomposed as follows: $\mathbb{O}_h(x_h \mid s_h) = (1 - \delta)\widetilde{\mathbb{O}}_h(x_h \mid s_h) + \delta E_h(x_h \mid s_h)$.*

*For notational convenience, for each $x \in \mathcal{X}_h$ let $\phi(x) \in \mathcal{S}_h$ be the unique state for which $\widetilde{\mathbb{O}}_h(x \mid \phi(x)) > 0$ (or arbitrary, if no such state exists).*

**Definition D.2.** *For any $h \in [H]$, $b \in \mathcal{S}_h$, and $x_h \in \mathcal{O}_h$, we write*

$$\mathbb{O}_h(x_h \mid b) := \sum_{z_h \in \mathcal{S}_h} \mathbb{O}_h(x_h \mid z_h)b(z_h).$$

*We similarly define $E_h(x_h \mid b)$ and $\widetilde{\mathbb{O}}_h(x_h \mid b)$.*

Notice that $\widetilde{\mathbb{O}}_h(x_h \mid b) = b(\phi(x_h))\widetilde{\mathbb{O}}_h(x_h \mid \phi(x_h))$.

## D.1 Belief Contraction

In this section, we prove Theorem D.1, a slight generalization of Theorem 3.2. The proof broadly follows the proof of belief contraction for $\gamma$-observable POMDPs [24] (of which $\delta$-perturbed Block MDPs are a special case — see Remark D.1), but since we require a stronger bound, we must modify the argument.

The basic idea (and main technical difficulty) in [24] is to identify a monotonic transform of an $f$-divergence that multiplicatively contracts in expectation under the Bayes operator (Definition B.1). In their case, they show that $\sqrt{\mathsf{KL}(\mathbb{B}_h(b; x_h) \,\|\, \mathbb{B}_h(b'; x_h))}$ contracts by roughly a constant factor (relative to $\sqrt{\mathsf{KL}(b \,\|\, b')}$) in expectation over $x_h \sim \mathbb{O}_h(\cdot \mid b)$. That is, updating the true belief and approximate belief by an observation drawn from the true belief tends to decrease the KL-divergence. Updating the two beliefs by applying a transition matrix cannot increase the KL-divergence since it is an $f$-divergence, so an iterative argument (alternating between observation updates and transition updates) proves exponential contraction of the belief error.

However, we would like to prove contraction by $\mathrm{poly}(\delta)$ per step, and the following example seems to present an obstacle to proving such contraction via KL-divergence. It also presents an obstacle to directly analyzing TV-distance.

**Example D.1** (Failure of contraction of TV and KL)**.** *Fix $\delta > 0$. Let $\mathcal{S} = \mathcal{X} = \{0, 1\}$ and let $\mathbb{O} : \mathcal{S} \to \Delta(\mathcal{X})$ be defined by $\mathbb{O}(x \mid x) = 1 - \delta$. Define $b = (1, 0)$ and $b' = (\delta^2, 1 - \delta^2)$. Then it holds almost surely over $x \sim \mathbb{O}(\cdot \mid b)$ that:*

- $\mathsf{TV}(\mathbb{B}(b; x), \mathbb{B}(b'; x)) \geq 1 - \delta$ *even though* $\mathsf{TV}(b, b') \leq 1$.

- $\mathsf{KL}(\mathbb{B}(b; x) \,\|\, \mathbb{B}(b'; x)) \geq \log(1/\delta)$ *even though* $\mathsf{KL}(b \,\|\, b') \leq 2\log(1/\delta)$.

$\triangleleft$

To resolve this, we observe that when the TV-distance fails to decay, the density ratio $\|b/b'\|_\infty$ does decay. To formalize this, we study contraction of the following error metric. While we cannot show that it contracts by $\mathrm{poly}(\delta)$ in expectation, we can show that it contracts with high probability; this is the content of Lemma D.2 below.

**Definition D.3.** *For distributions $b, b' \in \Delta(\mathcal{S})$ with $b \ll b'$, we define*

$$D_\star(b\|b') := \mathsf{TV}(b, b') \cdot \left\| \frac{b}{b'} \right\|_\infty.$$

Note that the above metric upper bounds TV-distance, and is upper bounded by $\|b/b'\|_\infty$. Also, as the product of metrics that satisfy the data processing inequality (Lemma C.1), it also satisfies the same inequality, so it cannot increase under application of (even stochastic) transitions.

**Lemma D.1.** *Let $h \in [H]$. Fix $b, b' \in \Delta(\mathcal{S}_h)$ with $b \ll b'$, and fix $x \in \mathcal{O}_h$. Then*

$$\left\| \frac{\mathbb{B}_h(b; x)}{\mathbb{B}_h(b'; x)} \right\|_\infty = \frac{\mathbb{O}_h(x \mid b')}{\mathbb{O}_h(x \mid b)} \cdot \left\| \frac{b}{b'} \right\|_\infty .$$

**Proof.** We have

$$\left\| \frac{\mathbb{B}_h(b; x)}{\mathbb{B}_h(b'; x)} \right\|_\infty = \max_{s \in \mathcal{S}_h} \frac{\mathbb{O}_h(x \mid s) b(s)}{\mathbb{O}_h(x \mid b)} \cdot \frac{\mathbb{O}_h(x \mid b')}{\mathbb{O}_h(x \mid s) b'(s)}$$

$$= \max_{s \in \mathcal{S}_h} \frac{b(s)}{b'(s)} \cdot \frac{\mathbb{O}_h(x \mid b')}{\mathbb{O}_h(x \mid b)}$$

$$= \frac{\mathbb{O}_h(x \mid b')}{\mathbb{O}_h(x \mid b)} \cdot \left\| \frac{b}{b'} \right\|_\infty$$

as claimed. $\square$

**Lemma D.2.** *Fix $h \in [H]$ and $b, b' \in \Delta(\mathcal{S}_h)$. Draw $x \sim \mathbb{O}_h(\cdot \mid b)$. Define random variable*

$$\xi := D_\star(\mathbb{B}_h(b; x) \| \mathbb{B}_h(b'; x)) = \mathsf{TV}(\mathbb{B}(b; x), \mathbb{B}(b'; x)) \left\| \frac{\mathbb{B}(b; x)}{\mathbb{B}(b'; x)} \right\|_\infty . \tag{5}$$

*Then $\mathbb{E}[\xi] \leq 4 D_\star(b \| b')$ and*

$$\Pr \left[ \xi > 4 \delta^{1/3} D_\star(b \| b') \right] \leq 2 \delta^{1/3}.$$

**Proof.** First, we compute that for any fixed $x \in \mathcal{X}_h$,

$$\mathsf{TV}(\mathbb{B}(b; x), \mathbb{B}(b'; x))$$

$$= \sum_{s \in \mathcal{S}_h} \mathbb{O}_h(x \mid s) \left| \frac{b(s)}{\mathbb{O}_h(x \mid b)} - \frac{b'(s)}{\mathbb{O}_h(x \mid b')} \right|$$

$$= (1 - \delta) \widetilde{\mathbb{O}}_h(x \mid \phi(x)) \left| \frac{b(\phi(x))}{\mathbb{O}_h(x \mid b)} - \frac{b'(\phi(x))}{\mathbb{O}_h(x \mid b')} \right| + \delta \sum_{s \in \mathcal{S}_h} E_h(x \mid s) \left| \frac{b(s)}{\mathbb{O}_h(x \mid b)} - \frac{b'(s)}{\mathbb{O}_h(x \mid b')} \right| \tag{6}$$

by Definition B.1 and Definition D.1. We bound these terms individually. To bound the first term, since $\mathbb{O}_h(x \mid b) = (1 - \delta) b(\phi(x)) \widetilde{\mathbb{O}}_h(x \mid \phi(x)) + \delta E_h(x \mid b)$ and $\mathbb{O}_h(x \mid b') = (1 - \delta) b'(\phi(x)) \widetilde{\mathbb{O}}_h(x \mid \phi(x)) + \delta E_h(x \mid b')$,

$$\widetilde{\mathbb{O}}_h(x \mid \phi(x)) \left| \frac{b(\phi(x))}{\mathbb{O}_h(x \mid b)} - \frac{b'(\phi(x))}{\mathbb{O}_h(x \mid b')} \right|$$

$$= \delta \widetilde{\mathbb{O}}_h(x \mid \phi(x)) \left| \frac{b(\phi(x)) E_h(x \mid b') - b'(\phi(x)) E_h(x \mid b)}{\mathbb{O}_h(x \mid b) \mathbb{O}_h(x \mid b')} \right|$$

$$\leq \delta \widetilde{\mathbb{O}}_h(x \mid \phi(x)) \left( \frac{|b(\phi(x)) - b'(\phi(x))| \cdot E_h(x \mid b)}{\mathbb{O}_h(x \mid b) \mathbb{O}_h(x \mid b')} + \frac{b(\phi(x)) \cdot |E_h(x \mid b) - E_h(x \mid b')|}{\mathbb{O}_h(x \mid b) \mathbb{O}_h(x \mid b')} \right)$$

$$\leq \delta \left( \widetilde{\mathbb{O}}_h(x \mid \phi(x)) \frac{|b(\phi(x)) - b'(\phi(x))| \cdot E_h(x \mid b)}{\mathbb{O}_h(x \mid b) \mathbb{O}_h(x \mid b')} + \frac{|E_h(x \mid b) - E_h(x \mid b')|}{(1 - \delta) \mathbb{O}_h(x \mid b')} \right) . \tag{7}$$

where the second inequality uses the fact that $\mathbb{O}_h(x \mid b) \geq (1 - \delta) b(\phi(x)) \widetilde{\mathbb{O}}_h(x \mid \phi(x))$. To bound the second term,

$$\sum_{s \in \mathcal{S}_h} E_h(x \mid s) \left| \frac{b(s)}{\mathbb{O}_h(x \mid b)} - \frac{b'(s)}{\mathbb{O}_h(x \mid b')} \right| \leq \sum_{s \in \mathcal{S}_h} \frac{E_h(x \mid s)}{\mathbb{O}_h(x \mid b')} |b(s) - b'(s)| + \sum_{s \in \mathcal{S}_h} \frac{E_h(x \mid s) b(s)}{\mathbb{O}_h(x \mid b) \mathbb{O}_h(x \mid b')} |\mathbb{O}_h(x \mid b) - \mathbb{O}_h(x \mid b')|$$

$$= \left( \sum_{s \in \mathcal{S}_h} \frac{E_h(x \mid s)}{\mathbb{O}_h(x \mid b')} |b(s) - b'(s)| \right) + \frac{E_h(x \mid b)}{\mathbb{O}_h(x \mid b) \mathbb{O}_h(x \mid b')} |\mathbb{O}_h(x \mid b) - \mathbb{O}_h(x \mid b')| \tag{8}$$

Let $\mathcal{E}$ be the set of $x \in \mathcal{X}_h$ such that $E_h(x \mid b) \leq \delta^{-1/3}\mathbb{O}_h(x \mid b)$. Then the quantity $\xi$ defined in Eq. (5) satisfies the following bound, where the expectation is over the randomness of $x \sim \mathbb{O}_h(\cdot \mid b)$:

$$\mathbb{E}[\xi \mathbb{1}[x \in \mathcal{E}]] = \sum_{x \in \mathcal{E}} \mathbb{O}_h(x \mid b)\mathsf{TV}(\mathbb{B}_h(b; x), \mathbb{B}_h(b'; x))\left\|\frac{\mathbb{B}_h(b; x)}{\mathbb{B}_h(b'; x)}\right\|_\infty$$

$$= \left\|\frac{b}{b'}\right\|_\infty \sum_{x \in \mathcal{E}} \mathbb{O}_h(x \mid b')\mathsf{TV}(\mathbb{B}_h(b; x), \mathbb{B}_h(b'; x))$$

$$\leq \left\|\frac{b}{b'}\right\|_\infty \left((1-\delta)\delta \sum_{x \in \mathcal{E}} \widetilde{\mathbb{O}}_h(x \mid \phi(x))\frac{E_h(x \mid b)}{\mathbb{O}_h(x \mid b)}|b(\phi(x)) - b'(\phi(x))| + \delta \sum_{x \in \mathcal{E}} |E_h(x \mid b) - E_h(x \mid b')|\right.$$

$$\left. + \delta \sum_{x \in \mathcal{E}} \sum_{s \in \mathcal{S}_h} E_h(x \mid s)|b(s) - b'(s)| + \delta \sum_{x \in \mathcal{E}} \frac{E_h(x \mid b)}{\mathbb{O}_h(x \mid b)}|\mathbb{O}_h(x \mid b) - \mathbb{O}_h(x \mid b')|\right)$$

$$\leq \left\|\frac{b}{b'}\right\|_\infty \left((1-\delta)\delta^{2/3} \sum_{x \in \mathcal{E}} \widetilde{\mathbb{O}}_h(x \mid \phi(x))|b(\phi(x)) - b'(\phi(x))| + \delta \sum_{x \in \mathcal{E}} |E_h(x \mid b) - E_h(x \mid b')|\right.$$

$$\left. + \delta \sum_{x \in \mathcal{E}} \sum_{s \in \mathcal{S}_h} E_h(x \mid s)|b(s) - b'(s)| + \delta^{2/3} \sum_{x \in \mathcal{E}} |\mathbb{O}_h(x \mid b) - \mathbb{O}_h(x \mid b')|\right)$$

$$\leq 4\delta^{2/3}\mathsf{TV}(b, b')\left\|\frac{b}{b'}\right\|_\infty \tag{9}$$

where the second equality is by Lemma D.1; the first inequality bounds each term $\mathsf{TV}(\mathbb{B}_h(b; x), \mathbb{B}_h(b'; x))$ using Eqs. (6) to (8); the second inequality uses the definition of $\mathcal{E}$; and the final inequality uses the data processing inequality for kernels $E_h$ and $\mathbb{O}_h$. Additionally,

$$\Pr[x \notin \mathcal{E}] = \sum_{x \in \mathcal{X}_h \setminus \mathcal{E}} \mathbb{O}_h(x \mid b) = \sum_{x \in \mathcal{X}_h} \mathbb{O}_h(x \mid b)\mathbb{1}\left[\frac{\mathbb{O}_h(x \mid b)}{E_h(x \mid b)} < \delta^{1/3}\right] \leq \delta^{1/3} \sum_{x \in \mathcal{X}_h} E_h(x \mid b) = \delta^{1/3} \tag{10}$$

since $E_h(\cdot \mid b)$ is a distribution. It follows that

$$\Pr\left[\xi > 4\delta^{1/3}\mathsf{TV}(b, b')\left\|\frac{b}{b'}\right\|_\infty\right] \leq \Pr\left[\xi\mathbb{1}[x \in \mathcal{E}] > 4\delta^{1/3}\mathsf{TV}(b, b')\left\|\frac{b}{b'}\right\|_\infty\right] + \Pr[x \notin \mathcal{E}]$$

$$\leq 2\delta^{1/3}$$

where the second inequality applies Markov's inequality to Eq. (9) for the first term, and Eq. (10) for the second term. This proves the second claim of the lemma statement. To prove the first claim, note that $E_h(x \mid b) \leq \delta^{-1}\mathbb{O}_h(x \mid b)$ for all $x \in \mathcal{X}_h$. Thus, modifying the calculation from Eq. (9) (this time summing over all $x \in \mathcal{X}_h$ instead of $x \in \mathcal{E}$) gives

$$\mathbb{E}[\xi] = \sum_{x \in \mathcal{X}_h} \mathbb{O}_h(x \mid b)\mathsf{TV}(\mathbb{B}_h(b; x), \mathbb{B}_h(b'; x))\left\|\frac{\mathbb{B}(b; x)}{\mathbb{B}(b'; x)}\right\|_\infty \leq 4\mathsf{TV}(b, b')\left\|\frac{b}{b'}\right\|_\infty$$

as needed. $\qquad\square$

The following result, which shows that the error metric decays with high probability under a belief update, is straightforward consequence of Lemma D.2 and the data processing inequality.

**Corollary D.1.** Fix $h \in \{2, \dots, H\}$. Let $b, b' \in \Delta(\mathcal{S}_{h-1})$ with $b \ll b'$. For any action $a_{h-1} \in \mathcal{A}_{h-1}$, with expectation over $x_h \sim (\mathbb{O}_h)^\top \mathbb{P}_h(a_{h-1}) \cdot b$,

$$\mathbb{E}[D_\star(\mathbb{U}_h(b; a_{h-1}, x_h)\|\mathbb{U}_h(b'; a_{h-1}, x_h))] \leq 4D_\star(b\|b')$$

and

$$\Pr[D_\star(\mathbb{U}_h(b; a_{h-1}, x_h)\|\mathbb{U}_h(b'; a_{h-1}, x_h)) > 4\delta^{1/3}D_\star(b\|b')] \leq 2\delta^{1/3}.$$

**Proof.** By applying Lemma D.2 with $\mathbb{P}_h(a_{h-1}) \cdot b$ and $\mathbb{P}_h(a_{h-1}) \cdot b'$,

$$\mathbb{E}[D_\star(\mathbb{U}_h(b; a_{h-1}, x_h)\|\mathbb{U}_h(b'; a_{h-1}, x_h))] = \mathbb{E}_{x_h \sim (\mathbb{O}_h)^\top \mathbb{P}_h(a_{h-1}) \cdot b}[D_\star(\mathbb{B}_h(\mathbb{P}_h(a_{h-1}) \cdot b; x_h)\|\mathbb{B}_h(\mathbb{P}_h(a_{h-1}) \cdot b'; x_h))]$$

$$\leq 4D_\star(\mathbb{P}_h(a_{h-1}) \cdot b\|\mathbb{P}_h(a_{h-1}) \cdot b')$$

$$\leq 4D_\star(b\|b')$$

where the final inequality uses the fact that $\mathsf{TV}(\mathbb{T}_h(a) \cdot b, \mathbb{T}_h(a) \cdot b') \leq \mathsf{TV}(b, b')$ and $\left\| \frac{\mathbb{T}_h(a) \cdot b}{\mathbb{T}_h(a) \cdot b'} \right\|_\infty \leq \left\| \frac{b}{b'} \right\|_\infty$ by the data processing inequality (Lemma C.1). Similarly, the second claim of Lemma D.2 gives that

$$\Pr\left[ D_\star(\mathbb{U}_h(b; a_{h-1}, x_h) \| \mathbb{U}_h(b'; a_{h-1}, x_h)) > 4\delta^{1/3} D_\star(\mathbb{P}_h(a_{h-1}) \cdot b \| \mathbb{P}_h(a_{h-1}) \cdot b') \right] \leq 2\delta^{1/3}$$

and therefore

$$\Pr\left[ D_\star(\mathbb{U}_h(b; a_{h-1}, x_h) \| \mathbb{U}_h(b'; a_{h-1}, x_h)) > 4\delta^{1/3} D_\star(b \| b') \right] \leq 2\delta^{1/3}$$

by again applying the data processing inequality as above. $\qquad\square$

We now can prove our main belief contraction result (which includes Theorem 3.2 as a special case) by iteratively applying Corollary D.1. The main technical detail is to verify that the observations are indeed drawn from the true belief states, which relies on Lemma C.2.

**Theorem D.1.** *There is a universal constant $C_{D.1} > 1$ with the following property. Fix an executable policy $\pi$, indices $1 \leq h < h + L \leq H$, and distributions $\mathcal{D}, \mathcal{D}' \in \Delta(\mathcal{S}_h)$. Then for any partial history $(x_{1:h}, a_{1:h-1})$ it holds that*

$$\mathbb{E}_{s_h \sim \mathcal{D}'} \mathbb{E}^\pi [\mathsf{TV}(\mathbf{b}_{h+L}^{\mathsf{apx}}(x_{h+1:h+L}, a_{h:h+L-1}; \mathcal{D}'), \mathbf{b}_{h+L}^{\mathsf{apx}}(x_{h+1:h+L}, a_{h:h+L-1}; \mathcal{D})) \mid s_h] \leq (C_{D.1}\delta)^{L/9} \left\| \frac{\mathcal{D}'}{\mathcal{D}} \right\|_\infty$$

*where the inner expectation is over partial trajectories $(x_{h+1:h+L}, a_{h:h+L-1})$ drawn from policy $\pi$ with the environment initialized in state $s_h$ at step $h$.*

*As a consequence, it holds for any partial history $(x_{1:h}, a_{1:h-1})$ that*

$$\mathbb{E}^\pi[\mathsf{TV}(\mathbf{b}_{h+L}(x_{1:h+L}, a_{1:h+L-1}), \mathbf{b}_{h+L}^{\mathsf{apx}}(x_{h+1:h+L}, a_{h:h+L-1}; \mathcal{D}))] \leq (C_{D.1}\delta)^{L/9} \left\| \frac{\mathbf{b}_h(x_{1:h}, a_{1:h-1})}{\mathcal{D}} \right\|_\infty$$

*where the expectation is over trajectories drawn from $\pi$ conditioned on the partial history $(x_{1:h}, a_{1:h-1})$.*

**Proof.** We first observe that the second claim follows from the first claim by setting $\mathcal{D}' := \mathbf{b}_h(x_{1:h}, a_{1:h-1})$. Indeed, conditioned on $(x_{1:h}, a_{1:h-1})$, the law of $s_h$ is precisely $\mathbf{b}_h(x_{1:h}, a_{1:h-1})$ (Lemma C.2), so drawing $(x_{h+1:h+L}, a_{h:h+L-1})$ conditioned on $(x_{1:h}, a_{1:h-1})$ is equivalent to first drawing $s_h \sim \mathbf{b}_h(x_{1:h}, a_{1:h-1})$ and then drawing $(x_{h+1:h+L}, a_{h:h+L-1})$ from the POMDP initialized at $s_h$. Moreover, by the recursive definitions of $\mathbf{b}$ and $\mathbf{b}^{\mathsf{apx}}$, we have

$$\mathbf{b}_{h+L}(x_{1:h+L}, a_{1:h+L-1}) = \mathbf{b}_{h+L}^{\mathsf{apx}}(x_{h+1:h+L}, a_{h:h+L-1}; \mathbf{b}_h(x_{1:h}, a_{1:h-1})).$$

It remains to prove the first claim. Fix $(x_{1:h}, a_{1:h-1})$. For $0 \leq t \leq L$, define the random variable

$$X_t := 4^{-t} D_\star(\mathbf{b}_{h+t}^{\mathsf{apx}}(x_{h+1:h+t}, a_{h:h+t-1}; \mathcal{D}') \| \mathbf{b}_{h+t}^{\mathsf{apx}}(x_{h+1:h+t}, a_{h:h+t-1}; \mathcal{D}))$$

where $(x_{h+1:h+L}, a_{h:h+L-1})$ is drawn by sampling $s_h \sim \mathcal{D}'$, initializing the POMDP at $s_h$, and then rolling out with policy $\pi$ (to be precise, the action distribution at step $h + t$ is $\pi(x_{1:h+t}, a_{1:h+t-1})$). Note that the roll-out does not resample $x_h$, which is already fixed. Recall that $D_\star(p \| q) := \mathsf{TV}(p, q) \left\| \frac{p}{q} \right\|_\infty$, so that $\mathsf{TV}(p, q) \leq D_\star(p \| q) \leq \left\| \frac{p}{q} \right\|_\infty$ for any distributions $p, q$. Then

$$\begin{aligned}
X_0 &= D_\star(\mathbf{b}_h^{\mathsf{apx}}(\emptyset; \mathcal{D}') \| \mathbf{b}_h^{\mathsf{apx}}(\emptyset; \mathcal{D})) \\
&= D_\star(\mathcal{D}' \| \mathcal{D}) \\
&\leq \left\| \frac{\mathcal{D}'}{\mathcal{D}} \right\|_\infty.
\end{aligned}$$

Moreover,

$$\begin{aligned}
&\mathsf{TV}(\mathbf{b}_{h+L}^{\mathsf{apx}}(x_{h+1:h+L}, a_{h:h+L-1}; \mathcal{D}'), \mathbf{b}_{h+L}^{\mathsf{apx}}(x_{h+1:h+L}, a_{h:h+L-1}; \mathcal{D})) \\
&\leq \min(D_\star(\mathbf{b}_{h+L}^{\mathsf{apx}}(x_{h+1:h+L}, a_{h:h+L-1}; \mathcal{D}') \| \mathbf{b}_{h+L}^{\mathsf{apx}}(x_{h+1:h+L}, a_{h:h+L-1}; \mathcal{D})), 1) \\
&\leq 4^L \min(X_L, 1) \qquad\qquad\qquad\qquad\qquad\qquad\qquad\qquad\qquad\qquad\qquad\qquad\qquad (11)
\end{aligned}$$

since $\mathsf{TV}(p, q) \leq 1$ for any distributions $p, q$. Fix $0 < t \leq L$ and condition on $(x_{h+1:h+t-1}, a_{h:h+t-2})$, which determines $X_{t-1}$. The conditional distribution of $a_{h+t-1}$ is then $\pi(x_{1:h+t-1}, a_{1:h+t-2})$, and for any fixed $a_{h+t-1}$ the conditional distribution of $x_{h+t}$ is $(\mathbb{O}_{h+t})^\top \mathbb{P}_{h+t}(a_{h+t-1}) \cdot \mathbf{b}^{\mathsf{apx}}_{h+t-1}(x_{h+1:h+t-1}, a_{h:h+t-2}; \mathcal{D}')$ by Lemma C.2 (applied to the modified POMDP that is initialized to a latent state $s_h \sim \mathcal{D}'$ immediately before the action $a_h$ is taken; for this POMDP $\mathbf{b}^{\mathsf{apx}}_{h+t-1}(x_{h+1:h+t-1}, a_{h:h+t-2}; \mathcal{D}')$ is the true belief state). Recall that by definition,

$$\mathbf{b}^{\mathsf{apx}}_{h+t}(x_{h+1:h+t}, a_{h:h+t-1}; \mathcal{D}') = \mathbb{U}_{h+t}(\mathbf{b}^{\mathsf{apx}}_{h+t-1}(x_{h+1:h+t-1}, a_{h:h+t-2}; \mathcal{D}'), a_{h+t-1}, x_{h+t})$$

and

$$\mathbf{b}^{\mathsf{apx}}_{h+t}(x_{h+1:h+t}, a_{h:h+t-1}; \mathcal{D}) = \mathbb{U}_{h+t}(\mathbf{b}^{\mathsf{apx}}_{h+t-1}(x_{h+1:h+t-1}, a_{h:h+t-2}; \mathcal{D}), a_{h+t-1}, x_{h+t}).$$

By Corollary D.1, it holds in expectation (resp., in probability) over $x_{h+t}$, conditioned on the prior history, that

$$
\begin{aligned}
\mathbb{E}[X_t] = 4^{-t} \mathbb{E}[D_\star(\mathbf{b}^{\mathsf{apx}}_{h+t}(x_{h+1:h+t}, a_{h:h+t-1}; \mathcal{D}') \| \mathbf{b}^{\mathsf{apx}}_{h+t}(x_{h+1:h+t}, a_{h:h+t-1}; \mathcal{D}))] \\
\leq 4^{1-t} D_\star(\mathbf{b}^{\mathsf{apx}}_{h+t-1}(x_{h+1:h+t-1}, a_{h:h+t-2}; \mathcal{D}') \| \mathbf{b}^{\mathsf{apx}}_{h+t-1}(x_{h+1:h+t-1}, a_{h:h+t-2}; \mathcal{D})) \\
= X_{t-1}
\end{aligned}
$$

and similarly

$$\Pr[X_t > \delta^{1/3} X_{t-1}] \leq 2\delta^{1/3}.$$

Since these bounds hold for any fixed $a_{h+t-1} \in \mathcal{A}_h$, they also hold in expectation (resp., in probability) over the joint draws of $a_{h+t-1}$ and $x_{h+t}$, conditioned on any realization of $(x_{h+1:h+t-1}, a_{h:h+t-2})$. Thus, $\mathbb{E}[X_t \mid X_{t-1}] \leq X_{t-1}$ and $\Pr[X_t > \delta^{1/3} X_{t-1} \mid X_{t-1}] \leq 2\delta^{1/3}$ both hold almost surely. We can now apply Lemma C.5 to the sequence $(X_0, \ldots, X_L)$ with parameters $S := \left\| \frac{\mathcal{D}'}{\mathcal{D}} \right\|_\infty$ and $\varepsilon := 2\delta^{1/3}$; we get that

$$\mathbb{E}[\min(X_L, 1)] \leq \mathbb{E}[\min(X_L, S)] \leq 2 \cdot 3^L 2^{L/3} \delta^{L/9} \left\| \frac{\mathcal{D}'}{\mathcal{D}} \right\|_\infty.$$

Combining this bound with Eq. (11), and setting $C_{D.1}$ to be a sufficiently large constant, completes the proof. $\qquad\square$

**Remark D.1.** Any $\delta$-perturbed Block MDP is $\gamma$-observable with $\gamma = 1 - 2\delta$ (Definition B.4): for any $h \in [H]$, we have $\mathbb{O}_h = (1 - \delta)\widetilde{\mathbb{O}}_h + \delta E_h$, and thus given any $b, b'$, we have

$$
\begin{aligned}
\left\| \mathbb{O}_h^\top b - \mathbb{O}_h^\top b' \right\|_1 &= \left\| (1 - \delta)\widetilde{\mathbb{O}}_h^\top (b - b') + \delta E_h^\top (b - b') \right\|_1 \\
&\geq (1 - \delta) \left\| \widetilde{\mathbb{O}}_h^\top (b - b') \right\|_1 - \delta \left\| E_h^\top (b - b') \right\|_1 \\
&\geq (1 - \delta) \left\| b - b' \right\|_1 - \delta \left\| E_h \right\|_{\mathrm{op}} \left\| b - b' \right\|_1 \\
&\geq (1 - 2\delta) \left\| b - b' \right\|_1
\end{aligned}
$$

because $\widetilde{\mathbb{O}}_h$ satisfies the block property. It was shown in [24, Theorem 4.7] that, for any $\gamma$-observable POMDP $\mathcal{P}$, the belief contraction error can be bounded as $\varepsilon^{\mathsf{contract}}_h(\pi; L) \leq (1 - \gamma^4/2^{40})^L \cdot \mathcal{O}(S)$. However, substituting in $\gamma := 1 - 2\delta$, we see that due to the constant factor of $2^{40}$, this bound does not asymptotically improve as $\delta$ decreases — indeed, it is never better than $(1 - 1/2^{40})^L \cdot \mathcal{O}(S)$ — and moreover is vacuous for $L = o(\log S)$.

### D.2 Approximate Decodability

In this section we prove Proposition 4.1, restated below as Proposition D.1, which states that for $\delta$-perturbed Block MDPs with *deterministic* latent transitions, the decodability error decays exponentially. The proof is somewhat analogous to that of Theorem D.1; the key difference is that the claim that the transitions do not increase decodability error is only true for deterministic transitions (whereas the analogous claim for belief contraction error is unconditionally true).

For notational convenience, we make the following definition of the "$\ell_\infty$ variance" $\mathsf{V}_\infty(b)$ for a given distribution $b$.

**Definition D.4.** *For any set $\mathcal{S}$ and distribution $b \in \Delta(\mathcal{S})$, define $\mathsf{V}_\infty(b) := 1 - \|b\|_\infty$.*

The following lemma shows that the $\ell_\infty$ variance contracts by $\mathrm{poly}(\delta)$ with high probability under the Bayes operator.

**Lemma D.3.** *Let $\delta > 0$, and suppose that $\mathcal{P}$ is a $\delta$-perturbed Block MDP with deterministic latent transitions (but potentially stochastic initial state). Fix $h \in [H]$ and $b \in \Delta(\mathcal{S}_h)$. Draw $x \sim \mathbb{O}_h(\cdot \mid b)$. Then $\mathbb{E}[\mathsf{V}_\infty(\mathbb{B}_h(b; x))] \leq \min(\mathsf{V}_\infty(b), \delta)$ and*

$$\Pr[\mathsf{V}_\infty(\mathbb{B}_h(b; x)) > \delta^{1/3}\mathsf{V}_\infty(b)] \leq 3\delta^{1/3}.$$

**Proof.** Pick any $s^\star \in \mathcal{S}_h$ such that $b_{s^\star} = \|b\|_\infty$, and hence $\sum_{s \in \mathcal{S}_h \setminus \{s^\star\}} b(s) = \mathsf{V}_\infty(b)$. Then

$$
\begin{aligned}
\mathbb{E}[\mathsf{V}_\infty(\mathbb{B}_h(b; x))] &= \sum_{x \in \mathcal{X}_h} \mathbb{O}_h(x \mid b) \left(1 - \max_{s \in \mathcal{S}_h} \mathbb{B}_h(b; x)(s)\right) \\
&= \sum_{x \in \mathcal{X}_h} \min_{s \in \mathcal{S}_h} \left(\mathbb{O}_h(x \mid b) - b(s)\mathbb{O}_h(x \mid s)\right) \\
&= \sum_{x \in \mathcal{X}_h} \min_{s \in \mathcal{S}_h} \sum_{s' \in \mathcal{S}_h \setminus \{s\}} b(s')\mathbb{O}_h(x \mid s') \qquad (12)
\end{aligned}
$$

where the second equality is by the definition $\mathbb{B}_h(b; x)(s) := \frac{b(s)\mathbb{O}_h(x|s)}{\mathbb{O}_h(x|b)}$ (Definition B.1). First, Eq. (12) implies that

$$\mathbb{E}[\mathsf{V}_\infty(\mathbb{B}_h(b; x))] \leq \sum_{x \in \mathcal{X}_h} \sum_{s' \in \mathcal{S}_h \setminus \{s^\star\}} b(s')\mathbb{O}_h(x \mid s') = \sum_{s' \in \mathcal{S}_h \setminus \{s^\star\}} b(s') = \mathsf{V}_\infty(b),$$

where the first equality uses the fact that $\mathbb{O}_h(\cdot \mid s')$ is a distribution for any fixed $s'$. Next, Eq. (12) implies that

$$
\begin{aligned}
\mathbb{E}[\mathsf{V}_\infty(\mathbb{B}_h(b; x))] &\leq \sum_{x \in \mathcal{X}_h} \sum_{s' \in \mathcal{S}_h \setminus \{\phi(x)\}} b(s')\mathbb{O}_h(x \mid s') \\
&= \delta \sum_{x \in \mathcal{X}_h} \sum_{s' \in \mathcal{S}_h \setminus \{\phi(x)\}} b(s')E_h(x \mid s') \\
&= \delta \sum_{s' \in \mathcal{S}_h} b(s') \sum_{x \in \mathcal{X}_h : \phi(x) \neq s'} E_h(x \mid s') \\
&\leq \delta \sum_{s' \in \mathcal{S}_h} b(s') \\
&\leq \delta. \qquad (13)
\end{aligned}
$$

This, together with the preceding bound, proves the first claim of the lemma. Now consider the event that $\phi(x) = s^\star$. Since $\mathbb{O}_h(x \mid b) \geq (1 - \delta)b(\phi(x))\widetilde{\mathbb{O}}_h(x \mid \phi(x))$, we have

$$\Pr[\phi(x) \neq s^\star] = 1 - \sum_{x \in \mathcal{X}_h : \phi(x) = s^\star} \mathbb{O}_h(x \mid b) \leq 1 - (1-\delta)b(s^\star) = 1 - (1-\delta)(1 - \mathsf{V}_\infty(b)) \leq \delta + \mathsf{V}_\infty(b).$$

$$(14)$$

Moreover, by an analogous calculation as Eq. (12),

$$\mathbb{E}[V_\infty(\mathbb{B}_h(b;x))\mathbb{1}[\phi(x)=s^\star]] = \sum_{x\in\mathcal{X}_h:\phi(x)=s^\star} \mathbb{O}_h(x\mid b)\left(1-\max_{s\in\mathcal{S}_h}\mathbb{B}_h(b;x)(s)\right)$$

$$= \sum_{x\in\mathcal{X}_h:\phi(x)=s^\star} \min_{s\in\mathcal{S}_h}\left(\mathbb{O}_h(x\mid b)-b(s)\mathbb{O}_h(x\mid s)\right)$$

$$\leq \sum_{x\in\mathcal{X}_h:\phi(x)=s^\star} \mathbb{O}_h(x\mid b)-b(s^\star)\mathbb{O}_h(x\mid s^\star)$$

$$= \sum_{x\in\mathcal{X}_h:\phi(x)=s^\star}\sum_{s\in\mathcal{S}_h\setminus\{s^\star\}} b(s)\mathbb{O}_h(x\mid s)$$

$$= \delta\sum_{x\in\mathcal{X}_h:\phi(x)=s^\star}\sum_{s\in\mathcal{S}_h\setminus\{s^\star\}} b(s)E_h(x\mid s)$$

$$\leq \delta V_\infty(b).$$

It follows that

$$\Pr[V_\infty(\mathbb{B}_h(b;x))>\delta^{1/3}V_\infty(b)] \leq \Pr[V_\infty(\mathbb{B}_h(b;x))\mathbb{1}[\phi(x)=s^\star]>\delta^{1/3}V_\infty(b)]+\Pr[\phi(x)\neq s^\star]$$

$$\leq \delta^{2/3}+\delta+V_\infty(b)$$

by Markov's inequality and Eq. (14). To conclude, we distinguish two cases. If $V_\infty(b)\leq\delta^{1/3}$, then we get

$$\Pr[V_\infty(\mathbb{B}_h(b;x))>\delta^{1/3}V_\infty(b)]\leq\delta^{2/3}+\delta+\delta^{1/3}\leq 3\delta^{1/3}$$

as needed. Otherwise, $V_\infty(b)>\delta^{1/3}$, so

$$\Pr[V_\infty(\mathbb{B}_h(b;x))>\delta^{1/3}V_\infty(b)]\leq\Pr[V_\infty(\mathbb{B}_h(b;x))>\delta^{2/3}]\leq\delta^{1/3}$$

by Markov's inequality and Eq. (13). This completes the proof. $\qquad\square$

Using Lemma D.3 and the assumption of deterministic latent dynamics, it is straightforward to show that the $\ell_\infty$ variance contracts by $\mathrm{poly}(\delta)$ with high probability under the *belief update* operator:

**Corollary D.2.** Let $\delta>0$, and suppose that $\mathcal{P}$ is a $\delta$-perturbed Block MDP with deterministic latent transitions (but potentially stochastic initial state). Fix $h\in[H]$ and $b\in\Delta(\mathcal{S}_h)$. For any action $a_{h-1}\in\mathcal{A}_{h-1}$, with expectation over $x_h\sim(\mathbb{O}_h)^\top\mathbb{P}_h(a_{h-1})\cdot b$, it holds that $\mathbb{E}[V_\infty(\mathbb{U}_h(b;a_{h-1},x_h))]\leq V_\infty(b)$ and

$$\Pr[V_\infty(\mathbb{U}_h(b;a_{h-1},x_h))>\delta^{1/3}V_\infty(b)]\leq 3\delta^{1/3}.$$

**Proof.** From Definition B.1, we have for any $x_h$ that $\mathbb{U}_h(b;a_{h-1},x_h)=\mathbb{B}_h(\mathbb{P}_h(a_{h-1})\cdot b;x_h)$. Applying Lemma D.3 to the distribution $\mathbb{P}_h(a_{h-1})\cdot b$ (observe that $x_h$ is indeed distributed according to $\mathbb{O}_h(\cdot\mid\mathbb{P}_h(a_{h-1})\cdot b)$), we get $\mathbb{E}[V_\infty(\mathbb{U}_h(b;a_{h-1},x_h))]\leq V_\infty(\mathbb{P}_h(a_{h-1})\cdot b)$ and

$$\Pr[V_\infty(\mathbb{U}_h(b;a_{h-1},x_h))>\delta^{1/3}V_\infty(\mathbb{P}_h(a_{h-1})\cdot b)]\leq 3\delta^{1/3}.$$

To complete the proof, it suffices to show that $V_\infty(\mathbb{P}_h(a_{h-1})\cdot b)\leq V_\infty(b)$. Indeed, since the transitions are deterministic, the matrix $\mathbb{P}_h(a_{h-1})\in\mathbb{R}^{|\mathcal{S}_h|\times|\mathcal{S}_{h-1}|}$ satisfies that every column is a standard basis vector. Identify any $s^\star\in\mathcal{S}_{h-1}$ with $b_{s^\star}=\|b\|_\infty$. Then there is some $s_h\in\mathcal{S}_h$ with $\mathbb{P}_h(a_{h-1})_{s_h,s^\star}=\mathbb{P}_h(s_h\mid s^\star,a_{h-1})=1$. But then the entry of $\mathbb{P}_h(a_{h-1})\cdot b$ indexed by $s_h$ is at least $b_{s^\star}$. So indeed $V_\infty(\mathbb{P}_h(a_{h-1})\cdot b)\leq V_\infty(b)$. $\qquad\square$

We can now prove the following restatement of Proposition 4.1.

**Proposition D.1.** *There is a universal constant $C_{D.1}>1$ so that the following holds. Let $\delta>0$, and suppose that $\mathcal{P}$ is a $\delta$-perturbed Block MDP with deterministic latent transitions (but potentially stochastic initial state). Fix any executable policy $\pi$ and index $h\in[H]$. It holds that*

$$\mathbb{E}^\pi[V_\infty(\mathbf{b}_h(x_{1:h},a_{1:h-1}))]\leq\min(\delta,(C_{D.1}\delta)^{(h-1)/9}).$$

**Proof.** Define a sequence of random variables $X_t := \mathsf{V}_\infty(\mathbf{b}_t(x_{1:t}, a_{1:t-1}))$ for $1 \le t \le h$, where $(x_{1:h}, a_{1:h-1})$ is a random trajectory drawn from policy $\pi$. We have $X_1 \le 1$ almost surely. By the same argument as in Theorem D.1 (except using Corollary D.2 rather than Corollary D.1), we have for all $1 < t \le h$ that $\mathbb{E}[X_t \mid X_{t-1}] \le X_{t-1}$ and $\Pr[X_t > \delta^{1/3}X_{t-1} \mid X_{t-1}] \le 3\delta^{1/3}$ hold almost surely. Thus, Lemma C.5 applied to $(X_1, \ldots, X_h)$ with parameters $S := 1$ and $\varepsilon := 3\delta^{1/3}$ implies that
$$\mathbb{E}[X_h] = \mathbb{E}[\min(X_h, 1)] \le 2 \cdot 3^{h-1} 3^{(h-1)/3} \delta^{(h-1)/9} \le (C_{D.1}\delta)^{(h-1)/9}$$
so long as $C_{D.1}$ is sufficiently large. Additionally, we know that $\mathbf{b}_1(x_1) = \mathbb{B}_1(\mathbb{P}_1; x_1)$ so
$$\mathbb{E}[X_1] = \mathbb{E}^\pi[\mathsf{V}_\infty(\mathbb{B}_1(\mathbb{P}_1; x_1))] \le \delta$$
by Lemma D.3 and the fact that $x_1$ has distribution $\mathbb{O}_1(\cdot \mid \mathbb{P}_1)$. Thus, $\mathbb{E}[X_t] \le \delta$ for all $1 \le t \le h$. $\qquad\square$

### D.3  Misspecification Lower Bound for Stochastic Dynamics

In this section we prove the following restatement of Proposition 5.1, which shows that in a $\delta$-perturbed Block MDP with general (stochastic) latent transitions, the misspecification of the optimal latent policy with respect to the class of executable policies can be as large as $\Omega(\delta H)$ (for $\delta \le 1/H$). This implies an analogous lower bound on decodability error, i.e. it cannot improve exponentially as $h$ increases, unlike the case of deterministic latent transitions. Moreover, it shows a fundamental source of (horizon-dependent) error that is not mitigated by increasing the frame-stack $L$: since the following bound applies to all executable policies, it also applies to the class of $L$-step executable policies for any $L$.

**Proposition D.2.** *Let $\delta > 0$ and $H \in \mathbb{N}$. There is a $\delta$-perturbed Block MDP $\mathcal{P}$ with horizon $H$ such that the optimal latent policy $\pi^{\text{latent}}$ satisfies*
$$\min_{\pi \in \Pi} \mathsf{TV}(\mathbb{P}^{\pi^{\text{latent}}}, \mathbb{P}^\pi) \ge \Omega(\min(1, \delta H))$$
*where $\Pi$ is the class of executable policies.*

**Proof.** We define $\mathcal{P}$ as follows. For all $h \in [H]$, define the latent state space and observation space to be $\mathcal{S}_h := \mathcal{X}_h := \{0, 1\}$; also define $\mathcal{A}_h := \{0, 1\}$. Let the initial distribution and latent transition dynamics at each step be uniformly random (independent of the previous state and action). For each $h \in [H]$, define the reward function $R_h : \mathcal{S}_h \times \mathcal{A}_h \to [0, 1]$ be defined by $R_h(s, a) = \frac{1}{H}\mathbb{1}[s = a]$. Define the observation distribution $\mathbb{O}_h : \mathcal{S}_h \to \Delta(\mathcal{S}_h)$ so that $\mathbb{O}_h(s \mid s) = 1 - \delta$ and $\mathbb{O}_h(1 - s \mid s) = \delta$.

It is clear that $\mathcal{P}$ is a $\delta$-perturbed Block MDP. Under the trajectory distribution $\mathbb{P}^{\pi^{\text{latent}}}$ induced by the optimal latent policy $\pi^{\text{latent}}$, it holds that $a_h = s_h$ for all $h \in [H]$ with probability 1. However, for any executable policy $\pi$, for any step $h$ and history $\tau_{1:h-1} = (s_{1:h-1}, x_{1:h-1}, a_{1:h-1})$, it holds that $\Pr^\pi[a_h = s_h \mid \tau_{1:h-1}] \le 1 - \delta$ since the prior history is independent of $s_h$, and the conditional distribution $s_h \mid x_h$ has only $1 - \delta$ mass on $x_h$. Thus,
$$\Pr^\pi[\forall h \in [H] : a_h = s_h] \le (1 - \delta)^H.$$
If $\delta \ge 1/H$ then $(1 - \delta)^H \le e^{-1} \le 1 - \Omega(1)$. Otherwise, $(1 - \delta)^H \le 1 - \Omega(\delta H)$. Thus, $\mathsf{TV}(\mathbb{P}^\pi, \mathbb{P}^{\pi^{\text{latent}}}) \ge \Omega(\min(1, \delta H))$ as claimed. $\qquad\square$

## E  Omitted Proofs for Expert Distillation

### E.1  Misspecification Bounds for Composed Policies

In this section we prove upper bounds on the misspecification of a latent policy (with respect to certain executable policies obtained by *composing* the latent with some belief state) in terms of instance-dependent error metrics. The first main result is Lemma E.3 (a restatement of Lemma 3.1), where the upper bound is in terms of decodability error (Definition 3.1) and error in approximating

the true belief state. The second main result is Lemma E.4, where the decodability error term is replaced by *action-prediction error* (Definition 6.1).

To prove Lemma E.3, it is convenient to first analyze the policy that samples a state from the true belief state induced by the current history, and then samples an action from $\pi^{\mathsf{latent}}$ accordingly. The key technical observation, below, encapsulates the intuition that resampling a near-deterministic random variable is likely to yield the same realization.

**Lemma E.1.** *Let $\mathbb{P}_{Y,Y_{\mathsf{nuis}}}$ be a joint distribution over random variables $(Y, Y_{\mathsf{nuis}})$. Let $\mathbb{P}_{Z|Y}$ be a conditional distribution. Define $\mathbb{Q}_Z$ as follows. Resample $Y' \sim \mathbb{P}_Y$ and then sample $Z \sim \mathbb{P}_{Z|Y'}$. Then*

$$\mathsf{TV}(\mathbb{P}_{Y,Y_{\mathsf{nuis}}}\mathbb{P}_{Z|Y}, \mathbb{P}_{Y,Y_{\mathsf{nuis}}}\mathbb{Q}_Z) \leq 2\mathsf{V}_\infty(\mathbb{P}_Y).$$

*Additionally,*

$$\mathsf{TV}(\mathbb{P}_{Y,Y_{\mathsf{nuis}}}\mathbb{P}_{Z|Y}, \mathbb{P}_{Y,Y_{\mathsf{nuis}}}\mathbb{Q}_Z) \leq 2\mathsf{V}_\infty(\mathbb{P}_Z).$$

**Proof.** Consider the process where we draw $(Y, Y_{\mathsf{nuis}}) \sim \mathbb{P}_{Y,Y_{\mathsf{nuis}}}$, $Z \sim \mathbb{P}_{Z|Y}$, and $Y' \sim \mathbb{P}_Y$. If $Y' = Y$ then we set $Z' = Z$; otherwise we sample $Z' \sim \mathbb{P}_{Z|Y'}$. Then $(Y, Y_{\mathsf{nuis}}, Z)$ is distributed according to $\mathbb{P}_{Y,Y_{\mathsf{nuis}}}\mathbb{P}_{Z|Y}$, and $(Y, Y_{\mathsf{nuis}}, Z')$ is distributed according to $\mathbb{P}_{Y,Y_{\mathsf{nuis}}}\mathbb{Q}_Z$. Thus, we have defined a coupling. Moreover,

$$\begin{aligned}
\Pr[(Y, Y_{\mathsf{nuis}}, Z) \neq (Y, Y_{\mathsf{nuis}}, Z')] &\leq \Pr[Y \neq Y'] \\
&\leq 2\Pr[Y \neq \arg\max_y \mathbb{P}_Y(y)] \\
&= 2\mathsf{V}_\infty(\mathbb{P}_Y)
\end{aligned}$$

as needed for the first claim. For the second claim,

$$\begin{aligned}
\Pr[(Y, Y_{\mathsf{nuis}}, Z) \neq (Y, Y_{\mathsf{nuis}}, Z')] &\leq \Pr[Z \neq Z'] \\
&= 2\mathsf{V}_\infty(\mathbb{P}_Z)
\end{aligned}$$

as needed. $\qquad\square$

**Lemma E.2.** *Let $\pi^{\mathsf{latent}} \in \Pi^{\mathsf{latent}}$ be a latent (Markovian) policy and define the executable policy $\pi$ by*

$$\pi(x_{1:h}, a_{1:h-1}) := \pi^{\mathsf{latent}} \circ \mathbf{b}_h(x_{1:h}, a_{1:h-1}).$$

*Then*

$$\mathsf{TV}(\mathbb{P}^{\pi^{\mathsf{latent}}}, \mathbb{P}^\pi) \leq 2\sum_{h=1}^H \mathbb{E}^\pi[\mathsf{V}_\infty(\mathbf{b}_h(x_{1:h}, a_{1:h-1}))].$$

**Proof.** For each $1 \leq h \leq H+1$ let $\pi \circ_h \pi^{\mathsf{latent}}$ denote the policy that follows $\pi$ for the first $h-1$ actions and subsequently follows $\pi^{\mathsf{latent}}$. Since $\pi \circ_1 \pi^{\mathsf{latent}} = \pi^{\mathsf{latent}}$ and $\pi \circ_{H+1} \pi^{\mathsf{latent}} = \pi$, it suffices to show that, for each $h \in [H]$,

$$\mathsf{TV}(\mathbb{P}^{\pi \circ_h \pi^{\mathsf{latent}}}, \mathbb{P}^{\pi \circ_{h+1} \pi^{\mathsf{latent}}}) \leq 2\mathbb{E}^\pi[\mathsf{V}_\infty(\mathbf{b}_h(x_{1:h}, a_{1:h-1}))].$$

Observe that both distributions have the same conditional distributions over $(s_{h+1:H}, x_{h+1:H}, a_{h+1:H})$ given $(s_{1:h}, x_{1:h}, a_{1:h})$. By this fact and the data processing inequality,

$$\mathsf{TV}(\mathbb{P}^{\pi \circ_h \pi^{\mathsf{latent}}}, \mathbb{P}^{\pi \circ_{h+1} \pi^{\mathsf{latent}}}) = \mathsf{TV}(\mathbb{P}^{\pi \circ_h \pi^{\mathsf{latent}}}_{X,Y,Y_{\mathsf{nuis}},Z}, \mathbb{P}^{\pi \circ_{h+1} \pi^{\mathsf{latent}}}_{X,Y,Y_{\mathsf{nuis}},Z})$$

where $X = (x_{1:h}, a_{1:h-1})$, $Y = s_h$, $Y_{\mathsf{nuis}} = s_{1:h-1}$, and $Z = a_h$. Both distributions have the same marginal over $(X, Y, Y_{\mathsf{nuis}})$. The distribution of $Y|X$ is precisely $\mathbf{b}_h(x_{1:h}, a_{1:h-1})$ by Lemma C.2 and the fact that $\pi$ is executable. In the former distribution, $a_h$ is generated by sampling from $\pi^{\mathsf{latent}}(s_h)$. In the latter distribution, $a_h$ is generated by sampling $s'_h \sim \mathbf{b}_h(x_{1:h}, a_{1:h-1})$ and then sampling from $\pi^{\mathsf{latent}}(s'_h)$. Thus, the conditions of Lemma E.1 are met (after conditioning out $X$), and we get

$$\mathsf{TV}(\mathbb{P}^{\pi \circ_h \pi^{\mathsf{latent}}}, \mathbb{P}^{\pi \circ_{h+1} \pi^{\mathsf{latent}}}) \leq 2\mathbb{E}^\pi[\mathsf{V}_\infty(\mathbf{b}_h(x_{1:h}, a_{1:h-1}))]$$

as needed. $\qquad\square$

We now prove the following restatement of Lemma 3.1.

**Lemma E.3.** *Let* $\pi^{\mathsf{latent}} \in \Pi^{\mathsf{latent}}$ *be a latent (Markovian) policy and let* $\widetilde{\mathbf{b}}_{1:H}$ *be a collection of functions* $\widetilde{\mathbf{b}}_h : \mathcal{X}^h \times \mathcal{A}^{h-1} \to \Delta(\mathcal{S}_h)$. *Define the executable policy* $\widetilde{\pi}$ *by*

$$\widetilde{\pi}(x_{1:h}, a_{1:h-1}) := \pi^{\mathsf{latent}} \circ \widetilde{\mathbf{b}}_h(x_{1:h}, a_{1:h-1}).$$

*Then*

$$\mathsf{TV}(\mathbb{P}^{\pi^{\mathsf{latent}}}, \mathbb{P}^{\widetilde{\pi}}) \le \sum_{h=1}^{H} \mathbb{E}^{\pi}\left[2\mathsf{V}_{\infty}(\mathbf{b}_h(x_{1:h}, a_{1:h-1})) + \left\|\mathbf{b}_h(x_{1:h}, a_{1:h-1}) - \widetilde{\mathbf{b}}_h(x_{1:h}, a_{1:h-1})\right\|_1\right]$$

*and*

$$\mathsf{TV}(\mathbb{P}^{\pi^{\mathsf{latent}}}, \mathbb{P}^{\widetilde{\pi}}) \le \sum_{h=1}^{H} \mathbb{E}^{\pi}\left[2\mathsf{V}_{\infty}(\mathbf{b}_h(x_{1:h}, a_{1:h-1}))\right] + \mathbb{E}^{\widetilde{\pi}}\left[\left\|\mathbf{b}_h(x_{1:h}, a_{1:h-1}) - \widetilde{\mathbf{b}}_h(x_{1:h}, a_{1:h-1})\right\|_1\right]$$

*where* $\pi(x_{1:h}, a_{1:h-1}) := \pi^{\mathsf{latent}} \circ \mathbf{b}_h(x_{1:h}, a_{1:h-1})$.

**Proof.** We can couple $\mathbb{P}^{\pi}$ and $\mathbb{P}^{\widetilde{\pi}}$ so that at any step $h$, if the trajectories have thus far been the same sequence $(x_{1:h}, a_{1:h-1})$, then the probability that they choose different actions $a_h$ is at most

$$\left\|\mathbf{b}_h(x_{1:h}, a_{1:h-1}) - \widetilde{\mathbf{b}}_h(x_{1:h}, a_{1:h-1})\right\|_1.$$

By a standard hybrid argument, it follows that

$$\mathsf{TV}(\mathbb{P}^{\pi}, \mathbb{P}^{\widetilde{\pi}}) \le \sum_{h=1}^{H} \mathbb{E}^{\pi}\left[\left\|\mathbf{b}_h(x_{1:h}, a_{1:h-1}) - \widetilde{\mathbf{b}}_h(x_{1:h}, a_{1:h-1})\right\|_1\right]$$

and also

$$\mathsf{TV}(\mathbb{P}^{\pi}, \mathbb{P}^{\widetilde{\pi}}) \le \sum_{h=1}^{H} \mathbb{E}^{\widetilde{\pi}}\left[\left\|\mathbf{b}_h(x_{1:h}, a_{1:h-1}) - \widetilde{\mathbf{b}}_h(x_{1:h}, a_{1:h-1})\right\|_1\right].$$

Combining with Lemma E.2 completes the proof. $\qquad\square$

The preceding lemma used the intuition that if the latent state is near-deterministic when conditioned on the observation/action history, then resampling it is unlikely to change it. The following lemma uses the intuition that if the *action* is near-deterministic (which is likely when the action-prediction is small) when conditioned on the action/observation history, then resampling the latent state — and subsequently resampling the action conditioned on this resampled latent state — is unlikely to change the action, though it may have changed the state.

**Lemma E.4.** *Let* $\pi^{\mathsf{latent}} \in \Pi^{\mathsf{latent}}$ *be a latent (Markovian) policy and let* $\widetilde{\mathbf{b}}_{1:H}$ *be a collection of functions* $\widetilde{\mathbf{b}}_h : \mathcal{X}^h \times \mathcal{A}^{h-1} \to \Delta(\mathcal{S}_h)$. *Define the executable policy* $\widetilde{\pi}$ *by*

$$\widetilde{\pi}(x_{1:h}, a_{1:h-1}) := \pi^{\mathsf{latent}} \circ \widetilde{\mathbf{b}}_h(x_{1:h}, a_{1:h-1}).$$

*Then*

$$\mathsf{TV}(\mathbb{P}^{\pi^{\mathsf{latent}}}, \mathbb{P}^{\widetilde{\pi}}) \le \sum_{h=1}^{H} \mathbb{E}^{\pi}\left[2\varepsilon_h^{\mathsf{act};\pi^{\mathsf{latent}}}(\pi) + \left\|\pi^{\mathsf{latent}} \circ \mathbf{b}_h(x_{1:h}, a_{1:h-1}) - \pi^{\mathsf{latent}} \circ \widetilde{\mathbf{b}}_h(x_{1:h}, a_{1:h-1})\right\|_1\right]$$

*where* $\pi(x_{1:h}, a_{1:h-1}) := \pi^{\mathsf{latent}} \circ \mathbf{b}_h(x_{1:h}, a_{1:h-1})$.

**Proof.** As with the proof of Lemma E.3, it suffices to show that

$$\mathsf{TV}(\mathbb{P}^{\pi^{\mathsf{latent}}}, \mathbb{P}^{\pi}) \le \sum_{h=1}^{H} \mathbb{E}^{\pi}\left[2\varepsilon_h^{\mathsf{act};\pi^{\mathsf{latent}}}(\pi)\right].$$

The proof is identical to that of Lemma E.3 except for using the second claim of Lemma E.1 instead of the first. $\qquad\square$

## E.2 Analysis of `Forward` for $L$-step Executable Policies

In this section we describe the (slightly modified) `Forward` imitation learning algorithm [62], applied to the problem of distilling an expert latent policy $\pi^{\mathsf{latent}}$ to an $L$-step executable policy $\widehat{\pi} \in \pi^L$. We first formally derive the expression for the policy learned in the infinite-sample limit (Lemma E.5), and then prove Theorem E.1 (the regret bound for this policy, restated as Theorem 4.1). Finally, we prove a finite-sample guarantee for the modified `Forward` algorithm (Theorem E.2), using the same ideas together with a standard finite-sample analysis for Maximum Likelihood Estimation [20].

**`Forward` with $L$-step random actions.** For a latent Markovian policy $\pi^{\mathsf{latent}} \in \Pi^{\mathsf{latent}}$ and a parameter $L \in [H]$, the (modified) `Forward` algorithm computes an $L$-step executable policy $\widehat{\pi} = \widehat{\pi}_{1:H}$ as follows. For $h = 1, \dots, H$:

1. Draw $n$ trajectories $\tau^i = (s^i_{1:H}, x^i_{1:H}, a^i_{1:H})$ from the policy $\widehat{\pi}_{1:h-L-1} \circ_{h-L} \mathrm{Unif}(\mathcal{A})$ (which follows $\widehat{\pi}$ until step $h - L - 1$ and subsequently plays uniformly random actions).

2. Compute

$$\widehat{\pi}_h := \arg\max_{\pi_h \in \Pi^L_h} \frac{1}{n} \sum_{i=1}^n \log \pi_h(\pi^{\mathsf{latent}}_h(s_h) \mid x_{\max(1,h-L+1):h}, a_{\max(1,h-L):h-1}).$$

Above, $\Pi^L_h$ is the set of $L$-step conditional distributions $\pi_h : \mathcal{X}^{h-L+1:h} \times \mathcal{A}^{h-L:h-1} \to \Delta(\mathcal{A})$. Note that the standard `Forward` algorithm is identical except that it draws data from $\widehat{\pi}_{1:h-1}$ at step $h$. The modified algorithm is simpler to analyze since the random actions do not bias the conditional distribution of the latent state at step $h$ given the partial history $(x_{h-L+1:h}, a_{h-L:h-1})$, so this distribution can be directly related to an approximate belief state with appropriate prior (Lemma C.3).

For notational convenience, let $\widetilde{\pi}$ denote the policy obtained from the above algorithm in the infinite-sample limit, i.e. at step $h$ we define

$$\widetilde{\pi}_h := \arg\max_{\pi_h \in \Pi^L_h} \mathbb{E}^{\widetilde{\pi}_{1:h-L-1} \circ_{h-L} \mathrm{Unif}(\mathcal{A})} \left[ \log \pi_h(\pi^{\mathsf{latent}}_h(s_h) \mid x_{\max(1,h-L+1):h}, a_{\max(1,h-L):h-1}) \right].$$

The following lemma gives a closed-form expression for this policy.

**Lemma E.5.** *It holds for all $h \in [H]$ that*

$$\widetilde{\pi}_h(\cdot \mid x_{h-L+1:h}, a_{h-L:h-1}) = \begin{cases} \pi^{\mathsf{latent}}_h \circ \mathbf{b}^{\mathsf{apx}}_h(x_{h-L+1:h}, a_{h-L:h-1}; d^{\widetilde{\pi}}_{h-L}) & \text{if } h > L \\ \pi^{\mathsf{latent}}_h \circ \mathbf{b}_h(x_{1:h}, a_{1:h-1}) & \text{otherwise} \end{cases}$$

*so long as $\Pi_h$ contains this conditional distribution.*

**Proof.** We have for each $h > L$ that

$$\widetilde{\pi}_h = \arg\max_{\pi_h \in \Pi_h} \mathbb{E}^{\widetilde{\pi}_{1:h-L-1} \circ_{h-L} \mathrm{Unif}(\mathcal{A})} \left[ \log \pi_h(\pi^{\mathsf{latent}}_h(s_h) \mid x_{h-L+1:h}, a_{h-L:h-1}) \right].$$

Since population-level maximum likelihood minimizes KL divergence, we get

$$\begin{aligned} \widetilde{\pi}_h(a_h \mid x_{h-L+1:h}, a_{h-L:h-1}) &= \sum_{s_h \in \mathcal{S}_h} \pi^{\mathsf{latent}}_h(a_h \mid s_h) \cdot \mathbb{P}^{\widetilde{\pi}_{1:h-L-1} \circ_{h-L} \mathrm{Unif}(\mathcal{A})}[s_h \mid x_{h-L+1:h}, a_{h-L:h-1}] \\ &= \sum_{s_h \in \mathcal{S}_h} \pi^{\mathsf{latent}}_h(a_h \mid s_h) \cdot \mathbf{b}^{\mathsf{apx}}_h(x_{h-L+1:h}, a_{h-L:h-1}; d^{\widetilde{\pi}_{1:h-L-1} \circ_{h-L} \mathrm{Unif}(\mathcal{A})}_{h-L})(s_h) \\ &= \sum_{s_h \in \mathcal{S}_h} \pi^{\mathsf{latent}}_h(a_h \mid s_h) \cdot \mathbf{b}^{\mathsf{apx}}_h(x_{h-L+1:h}, a_{h-L:h-1}; d^{\widetilde{\pi}}_{h-L})(s_h) \end{aligned}$$

where the second equality uses Lemma C.3. The application of this lemma uses the fact that $\widetilde{\pi}_{1:h-L-1} \circ_{h-L} \mathrm{Unif}(\mathcal{A})$ plays actions at steps $h - L, \dots, h - 1$ that are uniformly random. This proves the lemma in the case $h > L$. The case $h \le L$ is analogous but uses Lemma C.2 instead of Lemma C.3. $\qquad\square$

**Theorem E.1.** *Suppose that the POMDP $\mathcal{P}$ is a $\delta$-perturbed Block MDP with deterministic transitions, and fix $L \in \mathbb{N}$. Let $\pi^{\mathsf{latent}} \in \Pi^{\mathsf{latent}}$ be a latent (Markovian) policy, and let $\widetilde{\pi}$ be the policy computed by `Forward` with $L$-step random actions, in the infinite-sample limit. Then*

$$J(\pi^{\mathsf{latent}}) - J(\widetilde{\pi}) \le \mathsf{TV}(\mathbb{P}^{\pi^{\mathsf{latent}}}, \mathbb{P}^{\widetilde{\pi}}) \le O(\delta) + (C_{D.1}\delta)^{L/9} SH.$$

**Proof.** Define the executable policy $\pi$ by $\pi(x_{1:h}, a_{1:h-1}) := \pi^{\text{latent}} \circ \mathbf{b}_h(x_{1:h}, a_{1:h-1})$. By Lemma E.3 and the closed-form expression for $\widetilde{\pi}$ (Lemma E.5), we have

$$\mathsf{TV}(\mathbb{P}^{\pi^{\text{latent}}}, \mathbb{P}^{\widetilde{\pi}}) \leq \sum_{h=1}^{H} \mathbb{E}^{\pi}[2\mathsf{V}_{\infty}(\mathbf{b}_h(x_{1:h}, a_{1:h-1}))]$$
$$+ \sum_{h=L+1}^{H} \mathbb{E}^{\widetilde{\pi}}\left[\left\|\mathbf{b}_h(x_{1:h}, a_{1:h-1}) - \mathbf{b}_h^{\text{apx}}(x_{h-L+1:h}, a_{h-L:h-1}; d_{h-L}^{\widetilde{\pi}})\right\|_1\right].$$

By Proposition D.1 and the fact that $\pi$ is executable, the first term can be bounded as

$$\sum_{h=1}^{H} \mathbb{E}^{\pi}[2\mathsf{V}_{\infty}(\mathbf{b}_h(x_{1:h}, a_{1:h-1}))] \leq \sum_{h=1}^{H} \min\left(\delta, (C_{D.1}\delta)^{(h-1)/9}\right)$$
$$\leq O(\delta).$$

Next, for each $h \in \{L+1, \ldots, H\}$, we can bound

$$\mathbb{E}^{\widetilde{\pi}}\left[\left\|\mathbf{b}_h(x_{1:h}, a_{1:h-1}) - \mathbf{b}_h^{\text{apx}}(x_{h-L+1:h}, a_{h-L:h-1}; d_{h-L}^{\widetilde{\pi}})\right\|_1\right]$$
$$\leq (C_{D.1}\delta)^{L/9}\mathbb{E}^{\widetilde{\pi}}\left[\left\|\frac{\mathbf{b}_h(x_{1:h}, a_{1:h-1})}{d_{h-L}^{\widetilde{\pi}}}\right\|_{\infty}\right]$$
$$\leq (C_{D.1}\delta)^{L/9}\mathbb{E}^{\widetilde{\pi}}\left[\sum_{s \in \mathcal{S}_h} \frac{\mathbf{b}_h(x_{1:h}, a_{1:h-1})(s)}{d_{h-L}^{\widetilde{\pi}}(s)}\right]$$
$$= (C_{D.1}\delta)^{L/9} \sum_{s \in \mathcal{S}_h} \frac{1}{d_{h-L}^{\widetilde{\pi}}(s)}\mathbb{E}^{\widetilde{\pi}}[\mathbf{b}_h(x_{1:h}, a_{1:h-1})(s)]$$
$$= (C_{D.1}\delta)^{L/9} \sum_{s \in \mathcal{S}_h} \frac{1}{d_{h-L}^{\widetilde{\pi}}(s)}\mathbb{E}^{\widetilde{\pi}}\left[\mathbb{P}^{\widetilde{\pi}}[s_h = s \mid x_{1:h}, a_{1:h-1}]\right]$$
$$= (C_{D.1}\delta)^{L/9} \sum_{s \in \mathcal{S}_h} \frac{d_{h-L}^{\widetilde{\pi}}(s)}{d_{h-L}^{\widetilde{\pi}}(s)}$$
$$= (C_{D.1}\delta)^{L/9}S$$

where the first inequality uses Theorem D.1 (and the fact that $\widetilde{\pi}$ is an executable policy), and the second equality uses Lemma C.2 (and the fact that $\widetilde{\pi}$ is an executable policy). Putting everything together, we get

$$\mathsf{TV}(\mathbb{P}^{\pi^{\text{latent}}}, \mathbb{P}^{\widetilde{\pi}}) \leq O(\delta) + (C_{D.1}\delta)^{L/9}SH$$

as claimed. $\qquad\square$

### E.3 Finite-sample guarantee

**Theorem E.2.** *There is a constant $C_{E.2} > 0$ with the following property. Let $\delta, \eta, \varepsilon_{\text{opt}} > 0$ and suppose that the POMDP $\mathcal{P}$ is a $\delta$-perturbed Block MDP with deterministic transitions. If $n \geq C_{E.2}X^L A^{3L+1}H^2\varepsilon_{\text{opt}}^{-2}\log(Hn/\eta)$, the output $\widehat{\pi}$ of the modified* `Forward` *algorithm with $n$ samples per step satisfies, with probability at least $1 - \eta$,*

$$J(\pi^{\text{latent}}) - J(\widehat{\pi}) \leq \mathsf{TV}(\mathbb{P}^{\pi^{\text{latent}}}, \mathbb{P}^{\widehat{\pi}}) \leq O(\delta) + (C_{D.1}\delta)^{L/9}SH + \varepsilon_{\text{opt}}.$$

*Moreover, $\widehat{\pi}$ can be computed in time* $\text{poly}(n, H, X^L, A^L)$.

**Proof.** By a standard analysis of the log-loss for unconstrained distribution classes, $\widehat{\pi}_h(x_{\max(1,h-L+1):h}, a_{\max(1,h-L):h-1})$ is precisely the empirical distribution of $\pi_h^{\text{latent}}(s_h^i)$ over the data $i \in [n]$ with $(x_{\max(1,h-L+1):h}^i, a_{\max(1,h-L):h-1}^i) = (x_{\max(1,h-L+1):h}, a_{\max(1,h-L):h-1})$. Thus, $\widehat{\pi}$ can be computed in the stated time complexity.

To prove the claimed statistical bound, fix $h \in [H]$. Let $\mathcal{G} = \{g_{\pi_h} : \pi_h \in \Pi_h^L\}$ denote the family of distributions indexed by $\Pi_h^L$, where $g_{\pi_h}$ is the joint distribution of $\tau_h = (x_{\max(1,h-L+1):h}, a_{\max(1,h-L):h-1})$ and $\pi_h(\tau_h)$ over trajectories drawn from $\widehat{\pi}_{1:h-L-1} \circ_{h-L} \mathrm{Unif}(\mathcal{A})$. Also let $g^\star$ denote the joint distribution of $\tau_h$ and $\pi_h^{\mathsf{latent}}(s_h)$ over trajectories drawn from $\widehat{\pi}_{1:h-L-1} \circ_{h-L} \mathrm{Unif}(\mathcal{A})$. Then $g^\star = g_{\widehat{\pi}_h^\star} \in \mathcal{G}$ where

$$\widehat{\pi}_h^\star(\cdot \mid x_{h-L+1:h}, a_{h-L:h-1}) = \begin{cases} \pi_h^{\mathsf{latent}} \circ \mathbf{b}_h^{\mathsf{apx}}(x_{h-L+1:h}, a_{h-L:h-1}; d_{h-L}^{\widehat{\pi}}) & \text{if } h > L \\ \pi_h^{\mathsf{latent}} \circ \mathbf{b}_h(x_{1:h}, a_{1:h-1}) & \text{otherwise} \end{cases}$$

by the same argument as in Lemma E.5. Moreover, $g_{\widehat{\pi}_h}$ is precisely the Maximum Likelihood Estimation (MLE) estimate over distribution class $\mathcal{G}$ with $n$ samples from $g^\star$. Thus, we can compare $\widehat{\pi}_h$ and $\widehat{\pi}_h^\star$ using a standard analysis for MLE, e.g. [20, Proposition B.1]: we can bound the log-covering number of $\mathcal{G}$ (with discretization error $\varepsilon := 1/(Hn)$) by $X^L A^{L+1} \log(1/\varepsilon)$, so we get with probability at least $1 - \eta/H$ that

$$\mathsf{TV}(g^\star, g_{\widehat{\pi}_h}) = \mathbb{E}^{\widehat{\pi}_{1:h-L-1} \circ_{h-L} \mathrm{Unif}(\mathcal{A})}[\mathsf{TV}(\widehat{\pi}_h^\star(\cdot \mid \tau_h), \widehat{\pi}_h(\cdot \mid \tau_h)]$$
$$\leq \mathcal{O}\left(\sqrt{\frac{X^L A^{L+1} \log(Hn/\eta)}{n}}\right)$$

where $\tau_h := (x_{\max(1,h-L+1):h}, a_{\max(1,h-L):h-1})$. By change-of-measure, it follows that

$$\mathbb{E}^{\widehat{\pi}_{1:h-1}}[\mathsf{TV}(\widehat{\pi}_h^\star(\cdot \mid \tau_h), \widehat{\pi}_h(\cdot \mid \tau_h)] \leq \mathcal{O}\left(A^L \cdot \sqrt{\frac{X^L A^{L+1} \log(Hn/\eta)}{n}}\right) \leq \frac{\varepsilon_{\mathsf{opt}}}{H} \qquad (15)$$

where the final inequality holds by the theorem assumption that $n \geq C_{E.2} X^L A^{3L+1} H^2 \varepsilon_{\mathsf{opt}}^{-2} \log(Hn/\eta)$, so long as $C_{E.2}$ is a sufficiently large constant. Next, we observe that

$$\mathsf{TV}(\mathbb{P}^{\pi^{\mathsf{latent}}}, \mathbb{P}^{\widehat{\pi}})$$
$$\leq \sum_{h=1}^{H} \mathbb{E}^{\widehat{\pi}}[\mathsf{TV}(\pi_h^{\mathsf{latent}}(s_h), \widehat{\pi}_h(\tau_h))]$$
$$\leq \sum_{h=1}^{H} \mathbb{E}^{\widehat{\pi}}[\mathsf{TV}(\pi_h^{\mathsf{latent}}(s_h), \widehat{\pi}_h^\star(\tau_h))] + \sum_{h=1}^{H} \mathbb{E}^{\widehat{\pi}}[\mathsf{TV}(\widehat{\pi}_h^\star(\tau_h), \widehat{\pi}_h(\tau_h))]$$
$$\leq \sum_{h=1}^{H} \mathbb{E}^{\pi}[2\mathsf{V}_\infty(\mathbf{b}_h(x_{1:h}, a_{1:h-1}))] + \sum_{h=L+1}^{H} \mathbb{E}^{\widehat{\pi}}\left[\left\|\mathbf{b}_h(x_{1:h}, a_{1:h-1}) - \mathbf{b}_h^{\mathsf{apx}}(x_{h-L+1:h}, a_{h-L:h-1}; d_{h-L}^{\widehat{\pi}})\right\|_1\right]$$
$$+ \sum_{h=1}^{H} \mathbb{E}^{\widehat{\pi}}[\mathsf{TV}(\widehat{\pi}_h^\star(\tau_h), \widehat{\pi}_h(\tau_h))]$$

where the final inequality is by Lemma E.3. As in Theorem E.1, the first term can be bounded by $O(\delta)$ and the second term can be bounded by $(C_{D.1}\delta)^{L/9} S$. By Eq. (15) and a union bound over $h \in [H]$, the third term is at most $\varepsilon_{\mathsf{opt}}$ with probability at least $1 - \eta$. Substituting these bounds into the above expression completes the proof. $\square$

## F Omitted Proofs for Reinforcement Learning

In this section we prove Corollary 3.1, stated formally below.

**Corollary F.1.** There is a reinforcement learning algorithm that, for any given $\delta, \beta \in (0, 1/3)$ and $L \in \mathbb{N}$, and any $\delta$-perturbed Block MDP $\mathcal{P}$, learns a policy $\pi^{\mathsf{rl}}$ satisfying

$$J(\pi^\star) - J(\pi^{\mathsf{rl}}) \leq (C_{3.2}\delta)^{L/18} \cdot \mathrm{poly}(S, X, H)$$

with probability at least $1 - \beta$ and in time $(XA/\delta)^{\mathcal{O}(L)} \cdot \mathrm{poly}(H, S, \log(1/\beta))$.

**Proof.** Recall from Remark D.1 that any $\delta$-perturbed Block MDP $\mathcal{P}$ is $(1 - 2\delta)$-observable (Definition B.4). Also, by Theorem D.1, $\mathcal{P}$ satisfies $(\varepsilon; \phi, L)$-belief contraction for any $\phi > 0$ and $L \in \mathbb{N}$ with $\varepsilon := (C_{D.1}\delta)^{L/9} \cdot \phi^{-1}$. Thus, we can invoke Theorem B.1 with $\gamma := 1/3$ and $\varepsilon := (C_{D.1}\delta)^{L/18}\sqrt{3C^\star H^5 S^{7/2} X^2}$, where $C^\star$ is as defined in Theorem B.1. The choice of $\phi$ in Theorem B.1 indeed satisfies $\varepsilon = (C_{D.1}\delta)^{L/9} \cdot \phi^{-1}$, so $\mathcal{P}$ satisfies $(\varepsilon; \phi, L)$-belief contraction, and thus the algorithm BaSeCAMP [23] produces an executable policy $\widehat{\pi}$ that satisfies $J(\pi^\star) - J(\widehat{\pi}) \leq \varepsilon \cdot \mathrm{poly}(S, X, H)$ in time $\mathrm{poly}((AX)^L, H, S, \varepsilon^{-1}, \log(\beta^{-1}))$. Substituting in the choice of $\varepsilon$ completes the proof. $\qquad\square$

# G   A Motivating Toy Model for Smoothing

Adding to the discussion from Section 6, we informally discuss a theoretical toy model in which smoothing the expert may decrease (a metric version of) action-prediction error and thus improve final performance. Consider a horizon-1 POMDP where the state and action spaces have metrics $d_S$ and $d_A$, and the reward function $R$ is binary-valued. For each latent state $s$, let $G(s)$ be the set of "good" actions, i.e. $G(s) := \{a \in \mathcal{A} : R(s, a) = 1\}$, and let $D(s)$ be the diameter of $G(s)$. Suppose that the following natural assumptions hold:

1. The map $s \mapsto G(s)$ is $d_S \to d_A$ Lipschitz. That is, perturbing $s$ by $\varepsilon$ (with respect to metric $d_S$) only perturbs the set $G(s)$ by $O(\varepsilon)$ (with respect to metric $d_A$).

2. Under any observation, the posterior on states is "$\varepsilon$-local" with respect to $d_S$, i.e. contained in an $\varepsilon$-ball.

With no smoothing, the optimal expert may, for each $s$, play an arbitrary action in $G(s)$, so the "metric" action-prediction error (i.e. expected dispersion of actions played by the expert, conditioned on an observation) can be as large as $O(\varepsilon) + \max_s D(s)$. However, suppose the environment has motor noise. In particular, suppose the support of the noise is an $\eta$-radius ball (with respect to $d_A$) around the chosen action. Then for each $s$, the optimal policy is forced to the "interior" of $G(s)$, i.e. not within $\eta$ of the boundary, effectively equivalent to decreasing the diameter of $G(s)$ by $\eta$. Moreover, if $\eta \gtrsim \varepsilon$, then for any two states $s, s'$ in the posterior of observation $o$, the actions chosen by the optimal policy will lie in both $G(s)$ and $G(s')$, so (under mild additional structural assumptions, e.g. convexity of $G(s)$ in Euclidean space) distillation should produce an optimal policy.

Of course, if $\eta$ is too large, the "interior" of some of the $G(s)$ sets becomes empty, i.e. the optimal policy cannot play a robustly good action. As a result, it may play arbitrary actions for these states, so the action prediction error can become large again (and the policy value decreases).

# H   Supplemental Materials for Experiments

In Appendix H.1 we present details for our empirical test of whether perfect decodability holds in image-based locomotion tasks. In Appendix H.3 we present figures omitted from the main body of the paper. In Appendix H.4 we include hyperparameter choices and details about compute resources.

## H.1   Misspecification of Decodability in Practice

For each of the three tasks (walker-run, humanoid-walk, and dog-walk), we collect 2000 trajectories from the expert latent policy, and for each $L \in \{1, 2, 3, 4, 5\}$ we train a model directly mapping from $L$ observations $x_{h-L+1:h}$ to the state $s_h$, by minimizing mean-squared error. We then evaluate the validation error of the model on 100 trajectories collected from the same policy and plot the per-timestep error in Figure 5. We normalize the states with the trajectory mean and standard deviation of the combined dataset. Note that the error is composed with three parts: 1) error due to model capacity; 2) using fixed-length frame-stacks instead of the whole history; 3) inherent failure of perfect decodability. We observe that the trained model is able to achieve a small error in the later timesteps, which suggests that error 1, model capacity error, is (likely) small. However, the error is large in the initial timesteps. As error 2 does not exist for steps $h \leq L$, it follows that error 3 is non-trivial, i.e., perfect decodability fails.

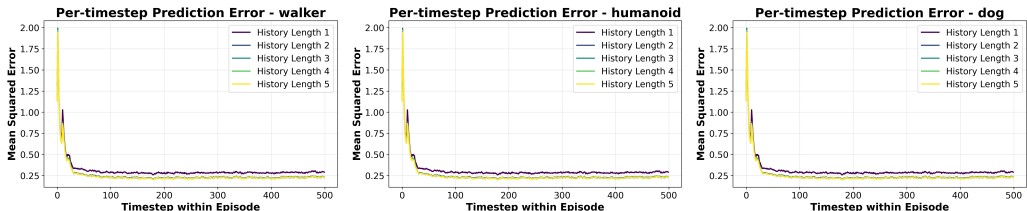

Figure 5: Per timestep validation error of the state-prediction model for different choices of frame-stack $L \in \{1, 2, 3, 4, 5\}$. The model is trained to predict the state given the most recent $L$ observations. We present the average error across 5 random seeds.

**Ablation.** To investigate whether the error at initial timesteps is due to parameter sharing, we also tried to train non-stationary models (i.e., one model for each timestep) or using weighted loss with higher weights for the initial timesteps, but neither approach significantly changed the results.

**Conclusions.** We conjecture that the higher error at early timesteps is due to a nearly uniform distribution for the initial state distribution, where states that induce occlusion may be quite likely. In later time-steps, the observations along the expert trajectory are in a more stable regime and thus may introduce less occlusion (and hence be more likely to correspond to a unique latent state). Finally, we observe that even at later time-steps there is still significant prediction error. While this could potentially be due to model capacity error, it nevertheless demonstrates impracticality of learning a perfect decoder, and it roughly corresponds with task difficulty (`humanoid-walk` and `dog-walk` are harder than `walker-run`).

**Source of the error.** One may naturally wonder if the error is caused by unpredictable state components that are irrelevant to decision-making (one example could be the absolute position of the agent, but in the environments that we test on, absolute coordinates are actually not part of the latent space). If this is the case, then the error would not negatively impact the performance of the policy distilled from the latent expert.

One piece of evidence against this hypothesis is that the action-prediction error is indeed non-trivial for our tasks of interest (c.r. Figure 9), which suggests that the error is not only due to irrelevant components. In addition, we take a more detailed look at the state prediction error and identify the components that contribute most to the error. With the same setup as in Figure 5, we compute the mean squared error for each coordinate of the state, averaged over the whole trajectories, and present the top 5 and bottom 5 coordinates in Table 1. We see that the coordinates that contribute most to the error are mostly angular velocities of limbs, which are indeed hard to predict from images. On the other hand, the coordinates that contribute least to the error are mostly joint angles or balance point coordinates, which are easier to predict from images. From first principles, all of these coordinates are crucial for the optimal policy, providing additional confirmation that the error is not only due to irrelevant components.

Table 1: (Left) Top 5 and (right) bottom 5 coordinates contributing to state prediction error.

| Coordinates | Error | Coordinates | Error |
|:---:|:---:|:---:|:---:|
| left_ankle_y angular velocity | 0.82 | left_elbow joint angle | 0.006 |
| left_hip_x angular velocity | 0.81 | balance point z | 0.020 |
| right_shoulder1 angular velocity | 0.69 | balance point y | 0.034 |
| right_shoulder2 angular velocity | 0.67 | balance point x | 0.044 |
| right_ankle_y angular velocity | 0.64 | left_knee joint angle | 0.045 |

## H.2 Imitating a Smoother Expert under Deterministic Latent Dynamics

In Section 6, we showed that under stochastic latent dynamics, imitating a smoother expert (which is trained under higher motor noise) can lead to better performance. A natural question is whether this phenomenon also exists under deterministic latent dynamics. Theoretically, the answer is no as we showed in Theorem E.1 that with enough framestack, imitating the non-smoothed expert can already be optimal under deterministic latent dynamics. That said, it remains unclear if smoothing the expert

can help improve the performance in practice. To answer this question, we conduct experiments in the same setup as in Section 6, but with motor noise $\sigma = 0$ when performing expert distillation. We vary the motor noise level used to train the latent expert over $\{0.1, 0.2, 0.3, 0.4, 0.5\}$, and we use a framestack of size 3.

We present the results in Figure 6. We see that imitating a smoother expert does not lead to better performance in this case, and the best performance is achieved by imitating the non-smoothed expert (c.r., Figure 1). This corroborates our theoretical findings.

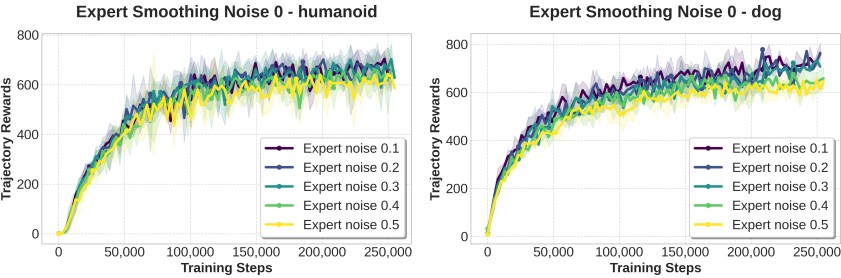

Figure 6: Performance of `DAgger` on the validation dataset for the `humanoid-walk` and `dog-walk` environments with motor noise $\sigma = 0$, as the noise level for the training environment (i.e. the environment in which the latent expert was trained) varies over $\{0.1, 0.2, 0.3, 0.4, 0.5\}$.

### H.3 Omitted Figures

#### H.3.1 Belief contraction with/without motor noise

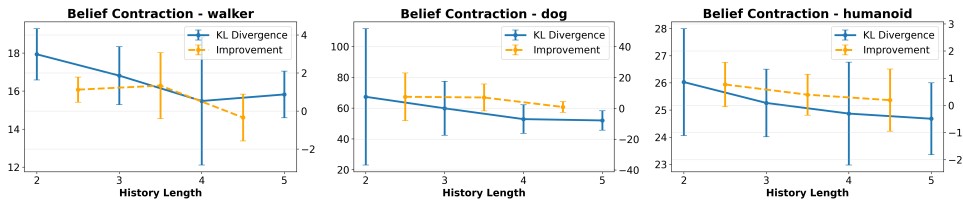

Figure 7: Belief contraction error with respect to the framestack $L = \{2, 3, 4, 5\}$ on all tasks. For each framestack $L$, we use train a Gaussian parametrized neural network to predict the belief with $L$ framestack input. We compute the KL distance to the output of an $L = 10$ network (serving as an approximation of the true belief), averaged over a validation dataset with 100 episodes of data. The orange plot denotes the decrease in KL divergence between two numbers of framestacks. We repeat the experiment for 5 times and plot the mean and standard deviation. We observe that the belief contraction error decreases (although not as fast as predicted by the theory) as the number of framestack increases.

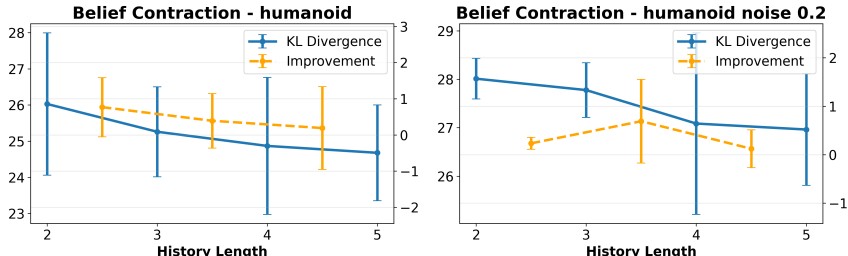

Figure 8: Belief contraction error with respect to the framestack $L = \{2, 3, 4, 5\}$ on humanoid-walk tasks, with and without motor noise. For each framestack $L$, we train a Gaussian parametrized neural network to predict the belief with $L$ framestack input. We compute the KL distance to the output of an $L = 10$ network (serving as an approximation of the true belief), averaged over a validation dataset with 100 episodes of data. The orange plot denotes the decrease in KL divergence between two numbers of framestacks. We repeat the experiment for 5 times and plot the mean and standard deviation. We observe that very similar belief contraction phenomena occur with or without the motor noise.

#### H.3.2 Action prediction error with smoothed experts

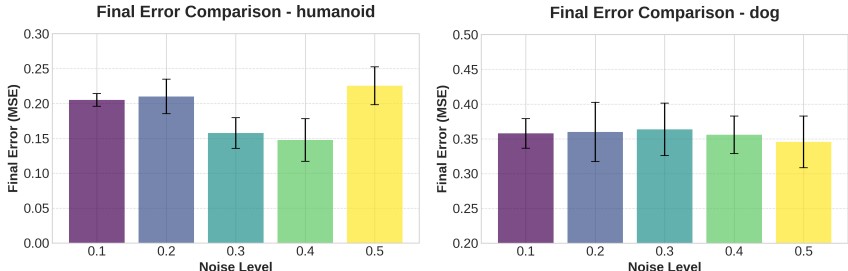

Figure 9: Comparison of the (estimated) action-prediction error of the latent policy on the validation dataset for the `humanoid-walk` and `dog-walk` environments with motor noise $\sigma = 0.2$, as the noise level for the training environment (i.e. the environment in which the latent expert was trained) varies over $\{0.1, 0.2, 0.3, 0.4, 0.5\}$. The action-prediction error was estimated using MSE as a proxy (normalized by dimension of the action space, as detailed in Section 2). We observe that imitating the latent expert that is trained on the same noise level does not yield the smallest prediction error. Moreover, policies with lower action-prediction error also broadly have higher performance (Figure 4).

## H.4 Experiment Details

**Hyperparameters for state prediction models.** The hyperparameters used and considered for the belief/state prediction models (both deterministic and Gaussian parametrized), corresponding to the experiments in Sections 3.1 and 5.2, in Table 2. For the neural network architecture, we use the same cnn block prescribed in [22], followed by a three layer neural network with ReLU activation. The architecture remains the same for the policies, and the hyperparameter hidden size refers to the hidden size of the feedforward part of the neural network.

Table 2: Hyperparameters for belief prediction models.

|  | Final value | Considered Values |
| --- | --- | --- |
| Minibatch size | 256 | 128, 256 |
| Learning rate | 1e-4 | 1e-3, 2e-4, 1e-4 |
| Optimizer | Adam | Adam |
| Number of epochs | 100 | 25, 50, 100 |
| Hidden layer size | 512 | 128, 256, 512 |

**Hyperparameters for expert distillation.** The hyperparameters of BC and `DAgger` are provided in Table 3 and Table 4 respectively. Note that the only exception is that `DAgger` is run for 6500 episodes in the motor noise 0.1 and 0.3 experiment because it converges slower than the rest of the experiments.

Table 3: Hyperparameters for BC.

|  | Final value | Considered Values |
| --- | --- | --- |
| Minibatch size | 256 | 128, 256 |
| Learning rate | 1e-4 | 1e-3, 2e-4, 1e-4 |
| Optimizer | Adam | Adam |
| Number of episodes | 2000 | 1000, 2000, 5000 |
| Number of epochs | 1000 | 100, 500, 1000 |
| Hidden layer size | 256 | 128, 256, 512 |

**Hyperparameters for RL.** The hyperparameters for RL follows the original hyperparameters prescribed in [22], and we train for 50000 episodes.

**Computation details.** All of our experiments are run with 1 L40S GPU with 8 threads of CPU.

Table 4: Hyperparameters for `DAgger`.

|  | Final value | Considered Values |
|---|---|---|
| Minibatch size | 256 | 128, 256 |
| Learning rate | 1e-4 | 1e-3, 2e-4, 1e-4 |
| Optimizer | Adam | Adam |
| Number of episodes | 5 | 2, 5, 10, 20 |
| Number of iterations | 5000 | 1000, 2000, 5000, 100000 |
| Number of gradient step per iteration | 50 | 20, 50, 100 |
| Hidden layer size | 256 | 128, 256, 512 |

