# OpenReview forum: "To Distill or Decide? Understanding the Algorithmic Trade-off in Partially Observable RL"
_NeurIPS.cc/2025/Conference — NeurIPS 2025 spotlight_

### Official Review · Reviewer_Nv62 · 2025-06-05

**Clarity:** 3
**Significance:** 3
**Originality:** 3
**Rating:** 5
**Confidence:** 3

**Summary:**

This paper investigates the theoretical properties of privileged expert distillation in partially observable RL. It begins by arguing that perfect decodability is unrealistic in practice and introduces the notions of approximate decodability and the perturbed Block MDP as a more practical model. The paper then systematically compares the theoretical guarantees and empirical performance of expert distillation versus standard RL in both deterministic and stochastic dynamics. Finally, it suggests that injecting stochasticity into the latent MDP can improve the effectiveness of distillation by yielding smoother experts.

**Questions:**

1. Is there a theoretical or empirical distinction between injecting noise into actions, observations, or latent state transitions? How do these different sources of stochasticity affect decodability and distillation?

2. In the stochastic setting, is the privileged expert trained with noise injected into the latent MDP, or is it trained on the original (noise-free) latent MDP? If the latter, doesn't that imply the expert and student are being trained on different MDPs, which could explain the degradation in performance?

3. What is the relationship between environment noise and expert smoothing noise? For instance, does higher environment noise require a higher level of expert smoothing? Could expert smoothing still help in environments with no noise?

4. Can the authors comment on whether the deterministic filter condition from [1] relates to their assumptions in Section 4? Is it possible to extend the results for deterministic dynamics to this broader condition?

5. Could the authors provide some discussions on the paradigm introduced in [2]?

[1] Cai, Y, et al. Provable partially observable reinforcement learning with privileged information.

[2] Li, Y, et al. Guided Policy Optimization under Partial Observability.

**Ethical Concerns:**

["NO or VERY MINOR ethics concerns only"]

**Final Justification:**

My concerns are well addressed, especially regarding the perfect decodability issue. Although the paper still has a limited scope compared to general POMDPs, I believe it establishes a solid foundation.

**Limitations:**

yes

**Paper Formatting Concerns:**

No concerns.

**Quality:**

3

**Strengths And Weaknesses:**

**Strengths**

- The paper addresses an important and timely question: when and why expert distillation is beneficial, especially in the context of robot learning. This has significant practical relevance.

- The focus on settings where decodability fails but observations remain informative is both realistic and more practical than prior assumptions of perfect decodability.

- It presents a comprehensive and thoughtful literature review.

- The theoretical results are well-developed, with formal proofs that support the key claims.

- Experimental results are carefully designed and align well with the theoretical predictions.

- The idea of smoothing the expert policy to improve distillation performance is well-motivated.

**Weakness**

1. Assumption about perfect decodability may be premature:

The paper's rejection of perfect decodability as unrealistic is foundational to its entire framing, and should be argued more rigorously. For instance:
  - The presence of large initial prediction error is not necessarily evidence against decodability; even in decodable settings, early predictions may be inaccurate due to limited history
  - Prediction errors could be caused by irrelevant or redundant latent variables (e.g., absolute coordinates) that are not essential for the task and naturally hard to infer without a reference frame.
  - Figure 5 lacks clarity — it's difficult to determine whether the final error is substantively large or just numerically small due to scale. A more illustrative example showing what is being mispredicted and how much could better support the claim.

2.  Lack of comparison to asymmetric RL:

Privileged expert distillation is often compared to asymmetric RL, which also utilizes privileged information during training. However, this paper only compares expert distillation to standard RL without privileged information. If latent states are available, then asymmetric RL would be a more practical and competitive baseline.

3. Limited task diversity:

While the experimental evaluation includes three DeepMind control suite tasks, all of them are locomotion-based. The lack of coverage of other types of partial observability — such as multi-agent coordination, strategy games, or visual navigation — limits the generality of the conclusions.

---

> ### Author Rebuttal · Authors · 2025-07-31
>
> We thank the reviewer for their careful attention to our work and we look forward to a constructive discussion period.
>
> We start by addressing the reviewer’s comments about our experiment which shows that perfect decodability is unrealistic:
>
> - **Initial prediction error**
>
>     > The presence of large initial prediction error is not necessarily evidence against decodability; even in decodable settings, early predictions may be inaccurate due to limited history
>
>     This is a matter of definition: we agree with the reviewer that in a realistic system, early predictions are often inaccurate, and this is precisely what we observe in Fig. 5. Unfortunately, the prior work [1] required the states at *all* steps to be perfectly decodable, in order for their algorithm with latent state information to have provable guarantees. The reviewer’s observation demonstrates why this assumption may be often unrealistic.
>
> - **Irrelevant latent variables**
>
>     > Prediction errors could be caused by irrelevant or redundant latent variables (e.g., absolute coordinates)...
>
>
>      Thanks for bringing up this point! For this rebuttal, we have performed the same experiment as in Fig. 5 but measuring action-prediction error instead, which avoids this confounder — if the state prediction error were solely due to difficult-to-predict latent variables that are irrelevant to the expert actions, we would find that the action-prediction error is very small. However, we observe below that it is in fact substantially large. The action-prediction error is normalized so that the “scale” is $O(1)$, just like the state-prediction error — see the response below.
>
>
>     |  | walker-run | humanoid-walk | dog-walk |
>     | --- | --- | --- | --- |
>     | Error at timestep 5 | 0.202 | 0.224 | 0.414 |
>     | Error at timestep 500 | 0.043 | 0.125 | 0.149 |
>
> - **Scale and source of prediction error**
>
>     > Figure 5 lacks clarity...A more illustrative example showing what is being mispredicted and how much could better support the claim.
>
>     Quantitatively, note that the states are normalized so that each coordinate has zero mean and unit variance; moreover, the MSE is normalized by 1/dimension (we mention this in Section 2 but will reiterate it in Section G.1). Therefore an MSE of 0.25 (humanoid) or 0.4 (dog) is indeed substantively large.
>
>     For this rebuttal, we have investigated which coordinates are mis-predicted the most. We use humanoid-walk as an example, and we measure the per-coordinate state prediction error averaged across 2000 samples, with frame-stack 3. We measured (a) the average error across all time-steps, (b) the first 20 steps, and (c) the last 20 steps, and we observe the coordinates with the highest and lowest errors in (a)/(b)/(c) are the same sets of coordinates. Here we report the results for (a):
>
>     **top-5 coordinates by error:**
>
>     | coordinate | error |
>     | --- | --- |
>     | left_ankle_y angular velocity | 0.82 |
>     | left_hip_x angular velocity | 0.81 |
>     | right_shoulder1 angular velocity | 0.69 |
>     | right_shoulder2 angular velocity | 0.67 |
>     | right_ankle_y angular velocity | 0.64 |
>
>     **bottom-5 coordinates by error:**
>
>     | coordinate | error |
>     | --- | --- |
>     | left_elbow joint angle | 0.006 |
>     | balance point z | 0.020 |
>     | balance point y | 0.034 |
>     | balance point x | 0.044 |
>     | left_knee joint angle | 0.045 |
>
>     We remark that the state space does not include absolute coordinates; all x,y positions are measured relative to the model’s torso. We observe that the hardest coordinates to predict are usually velocity terms that require multiple frames to decode, and suffer from environment stochasticity and especially from occlusion. Also, note that all of these coordinates are essential for decision-making, rather than redundant latent variables. We thank the reviewer for raising this interesting question and we will make sure to add the comprehensive version of this result in the final version of the paper.
>
>
> We proceed to the reviewer’s other comments and questions.
>
> > Lack of comparison to asymmetric RL.
>
> Asymmetric RL is an exciting research direction that holds significant promise for balancing computational/statistical efficiency with accuracy. On these axes, we see it as almost a continuum of methods interpolating between expert distillation and standard RL — indeed, typical asymmetric RL methods combine aspects of both paradigms. Our goal in this paper was to understand the trade-offs between the two methods at the “extremes” — which are, to our knowledge, also the two most mainstream methods in applied vision-based robotics and autonomous driving [4,5].
>
> That said, we agree that obtaining a theoretical understanding of when asymmetric RL can achieve the best-of-both-worlds is an exciting direction for future research. Our theoretical framework could be an interesting jumping-off point for investigating these methods. We will expand our discussion of asymmetric RL in the paper to include these points.
>
> > Limited task diversity
>
> We fully agree: as we mention in the paper, our conclusions are specific to the setting where observations are highly informative of the latent state, which is realistic in e.g. image-based robotics but certainly unrealistic in many other domains. That said, the most general theory for POMDPs asserts that learning and planning are statistically and computationally intractable [3]. Such a general theory is too pessimistic for any practical guidance, and thus we believe more fine-grained theory is necessary to understand real-world POMDP problems.
>
> > Is there a theoretical or empirical distinction between injecting noise into actions, observations, or latent state transitions? How do these different sources of stochasticity affect decodability and distillation?
>
> Injecting noise into actions is a specific method of injecting latent state transition noise. We chose this particular type of transition noise since it is well-motivated by real-world robotics systems. As for observation noise, this is exactly the construction of our key setting, perturbed block MDPs, where the observation can be understood as a noisy measurement of the latent state. Our theory shows that (1) with observation noise but without action noise, decodability (and hence the performance of distillation) improves as framestack increases, but (2) with observation noise and action noise, decodability is fundamentally limited by the observation noise. Without observation noise, the latent expert is the same as the POMDP expert, so distillation is trivial (regardless of action noise). Thus, there is indeed a distinction between action noise and observation noise.
>
> > In the stochastic setting, is the privileged expert trained with noise injected into the latent MDP, or is it trained on the original (noise-free) latent MDP?
>
> In the stochastic setting (Sec 5.2), we trained the expert and the student on the same MDP — i.e., with noise injected into the latent MDP. We will better clarify this point in the revised version.
>
> > What is the relationship between environment noise and expert smoothing noise? For instance, does higher environment noise require a higher level of expert smoothing? Could expert smoothing still help in environments with no noise?
>
> This is an interesting question — we do not have a complete understanding of how the environment’s inherent stochasticity affects the optimal smoothing noise. In the extreme case where the environment is deterministic, our theory suggests that expert smoothing should not really help, as expert distillation is already near-optimal (with sufficient frame-stacking). For the rebuttal, we conducted this experiment (with $L$=3, the same setting as Fig.4, except no motor noise in the latent dynamics) and it confirms the theory (expert noise denotes expert trained under latent mdp with motor noise $n$):
>
> | expert noise |  0 | 0.1 | 0.2 | 0.3 | 0.4 | 0.5 |
> | --- | --- | --- | --- | --- | --- | --- |
> | dog | 722.5 | 696.8 | 673.6 | 673.3 | 688.4 | 689.7 |
> | humanoid | 739.1 | 676.6 | 680.5 | 683.7 | 677.1 | 654.0 |
>
> > Can the authors comment on whether the deterministic filter condition from [1] relates to their assumptions in Section 4? Is it possible to extend the results for deterministic dynamics to this broader condition?
>
> Good question, we will clarify this point in the paper. The deterministic filter condition is the same as perfect $H$-step decodability as defined in our paper. However, it is not actually broader than the setting of Proposition 4.1, because $\delta$-perturbed Block MDPs generically have positive decodability error, even with deterministic transitions. It is unclear if the setting can be relaxed further, but some assumptions on the emission distribution are definitely necessary [3].
>
> > discussions on the paradigm introduced in [2]
>
> We appreciate the reviewer for pointing out this related work. We were not aware of this paper as it showed up after the submission deadline. That said, we believe [2] is highly relevant as it also considers POMDPs with privileged latent information. The paper proposes a practical teacher-student co-training algorithm that achieves competitive performance on several benchmarks. It focuses on settings where active information gathering is crucial, which is complementary to the focus of our work but an exciting direction for future theoretical investigation. We will make sure to discuss it in our final version.
>
> [1] Cai, Y, et al. Provable partially observable reinforcement learning with privileged information.
>
> [2] Li, Y, et al. Guided Policy Optimization under Partial Observability.
>
> [3] Jin, C, et al. Sample-efficient reinforcement learning of undercomplete pomdps.
>
> [4] Ibarz, J, et al. How to train your robot with deep reinforcement learning: lessons we have learned.
>
> [5] Tang, C, et al. Deep reinforcement learning for robotics: A survey of real-world successes.

---

> > ### Comment · Reviewer_Nv62 · 2025-08-01
> >
> > Thank you for your response. My concerns are well addressed, especially regarding the perfect decodability issue. Although the paper still has a limited scope compared to general POMDPs, I believe it establishes a solid foundation. I have raised my score to 5.

---

> > > ### Author Response · Authors · 2025-08-01
> > >
> > > We thank the reviewer for their timely response and we are glad to know that our rebuttal addresses the reviewer's concerns. We will make sure to incorporate the rebuttal into our final version of the paper.

---

### Official Review · Reviewer_Aw6u · 2025-06-23

**Clarity:** 3
**Significance:** 3
**Originality:** 4
**Rating:** 4
**Confidence:** 3

**Summary:**

This paper investigates the trade-off between expert distillation and standard RL in POMDPs, finding that the stochasticity of the latent dynamics governs their relative performance. The authors provide a theoretical framework based on approximate decodability and belief contraction to explain this trade-off. They also show empirically that distilling a smoother expert, trained with noise, can be a more effective strategy than distilling the technically optimal one.

**Questions:**

Please see weaknesses section.

**Ethical Concerns:**

["NO or VERY MINOR ethics concerns only"]

**Final Justification:**

I will maintain Borderline Accept.

**Limitations:**

yes.

**Quality:**

3

**Strengths And Weaknesses:**

Strengths
- The paper provides a strong theoretical framework for the distillation vs. RL trade-off by introducing the 'perturbed Block MDP' model and using it to contrast the core conditions of 'approximate decodability' and 'belief contraction'.
- The paper effectively connects theory to practice, not only by validating its theoretical predictions with experiments (Fig. 3) but also by uncovering a significant and practical insight: distilling a 'smoother' expert can be more effective than distilling the optimal one (Fig. 4).

Weaknesses
- The provided theoretical bounds, which appear to be worst-case guarantees, do not always align perfectly with the empirical outcomes.
- The finding that smoother experts can be better teachers is one of the most exciting directions in the paper, and the authors provide a compelling proof-of-concept by injecting motor noise. It would be valuable to see it explored further. For instance, a more systematic study on how to tune the 'optimal' amount of noise for a given task, or a deeper theoretical dive into how this smoothing technique impacts the action-prediction error bound, could significantly build upon this novel contribution.

---

> ### Author Rebuttal · Authors · 2025-07-31
>
> We thank the reviewer for their careful attention to our work and we look forward to a constructive discussion period.
>
> > The provided theoretical bounds, which appear to be worst-case guarantees, do not always align perfectly with the empirical outcomes.
>
> We agree completely: there are several gaps between the empirical results and the worst-case theoretical guarantees (e.g., Sec. 5.2, belief contraction error as a metric for sub-optimality of RL), and we see these gaps as exciting directions for future work — places where a more complex theory could be more tightly predictive. POMDPs are a rich (and still largely mysterious) model class, and the goal of our theory was not to develop some complete quantitative characterization, but rather to achieve a principled qualitative understanding of factors that influence popular algorithms’ performance in practice. We believe the empirical results demonstrate that our theory does have predictive power, despite being derived via worst-case guarantees.
>
> > The finding that smoother experts can be better teachers is one of the most exciting directions in the paper, and the authors provide a compelling proof-of-concept by injecting motor noise. It would be valuable to see it explored further. For instance, a more systematic study on how to tune the 'optimal' amount of noise for a given task, or a deeper theoretical dive into how this smoothing technique impacts the action-prediction error bound, could significantly build upon this novel contribution.
>
> We are glad that the reviewer also found this result exciting. While this result was indeed a proof-of-concept, we can share some additional thoughts on noise tuning and the theoretical basis for smoothing:
>
> - For tuning the noise, one approach (suggested by our theory and roughly validated by the experiment in Fig. 8) is to select the expert with the smallest action prediction error. Although ideally the error would be estimated under learner visitation distribution, we believe that using the expert visitation distribution could be a good surrogate. A more direct approach is to simply pick the highest noise level for which the expert performance is within a reasonable threshold (which might require certain domain knowledge).
> - Our basic theory for how motor noise improves action-prediction error is the following. Consider a horizon-1 decision-making task where the state and action spaces have metrics $d_S$ and $d_A$. For each latent state $s$, let $G(s)$ be the set of ''good'' actions, i.e. actions that achieve maximal reward. Suppose that (1) the sets $G(s)$ are $d_S \to d_A$ Lipschitz, i.e. perturbing $s$ by $\epsilon$ w.r.t. $d_S$ only perturbs $G(s)$ by $O(\epsilon)$ w.r.t. $d_A$, and (2) the posterior under any observation is ''local'' w.r.t. $d_S$, i.e. contained in an $\epsilon$-ball. With no smoothing, the optimal expert may, for each $s$, play an arbitrary action in $G(s)$, so the action-prediction error can be as large as $O(\epsilon)$ + 2 * max diameter of $G(s)$. However, if the environment has motor noise -- say that the support of the noise is an $r$-radius ball (w.r.t. $d_A$) around the chosen action. Then for each $s$, the optimal policy is forced to the ``interior'' of $G(s)$, i.e. not within $r$ of the boundary, effectively equivalent to decreasing the diameter of $G(s)$ by $r$.
>
> This leads to two predictions. First, the action-prediction error initially decreases as $r$ increases. Second, when $r$ gets too large, the ``interior" of some of the $G(s)$ sets becomes empty, i.e. the optimal policy cannot play a robustly good action. As a result, it may play arbitrary actions for these states, so the action prediction error becomes larger again (and the policy value decreases). This prediction is validated in the humanoid task (Fig 8a) but unclear for the dog task (Fig 8b) -- where the changes in action-prediction error are within margin-of-error (and the downstream performance gains from smoothing are also smaller).
>
> We appreciate the questions and we will make sure to include these discussions in the final version of the paper.

---

> > ### Author Response · Authors · 2025-08-05
> >
> > Dear Reviewer Aw6u,
> >
> > As the discussion period is coming towards an end, we would greatly appreciate hearing whether our response addressed your concerns, or whether you have further questions for us.
> >
> > Thank you again for your time!

---

> > ### Comment · Reviewer_Aw6u · 2025-08-07
> >
> > Thank you for your response. I will maintain my positive score.

---

> > > ### Author Response · Authors · 2025-08-08
> > >
> > > Thank you for your timely reply and recognition of our work!

---

### Official Review · Reviewer_Yzkr · 2025-07-03

**Clarity:** 3
**Significance:** 2
**Originality:** 2
**Rating:** 4
**Confidence:** 4

**Summary:**

The paper studies RL vs imitation learning in POMDPs. It firstly verifies that perfect L-decodable POMDP may not hold using experiments and then considered a setting call $\delta$-perturbed Block MDP. Then the paper proposed provable algorithms based on the approximate $L$-decodability induced by the $\delta$-perturbed Block MDP assumption. Finally, the paper also discussed how to improve expert distillation by smoothness.

**Questions:**

NA

**Ethical Concerns:**

["NO or VERY MINOR ethics concerns only"]

**Final Justification:**

The rebuttal has clarified my questions and I thus updated my evaluation.

**Limitations:**

Yes

**Quality:**

2

**Strengths And Weaknesses:**

Strength:

1. The paper combines many theoretical claims with experimental justifications, where provide valuable insights.
2. The argument that latent expert might not be the best target and smoothness might help is novel.

Weakness:

1. I find the paper oftentimes confuses the concept of decodable and $L$-decodable. Technically speaking, I am not aware of any references that use the nomenclature of ``decodable POMDP''. Although the condition by [6] should be equivalent to decodable by $H$ steps. However, it seems to me what the paper has relaxed is the perfect $L$-decodable condition with small $L$ (Efroni et al 2022) instead of the condition in [6] as claimed in the intro. Therefore, to substantiate a meaningful relaxation over [6], the paper should either: (a) establish provable guarantees in the regime of large $L$, or (b) clearly discuss this distinction in the main text to avoid conceptual confusion.

2. Secondly, it seems the technical assumption that really enables approximate $L$-decodability is perturbed block MDP + deterministic transition. In terms of theoretical contributions, this is arguably quite strong to have structural assumptions on emission and transition at the same time.

3. Meanwhile, in the experimental validation, the paper is only using $L\in [5]$ to verify the decodability. This again tries to only verifies whether $L$-decodability holds with small $L$. Moreover, I understand in practice ppl often only use observation history. However, action history is also important in theory and in terms of verifying the theoretical conditions faithfully.

4. As claimed, one important goal of the paper is to understand expert distillation vs RL. However, the message is that with deterministic transition, expert distillation match RL bounds. With stochastic transition, expert distillation is worse. Therefore, it seems the theory itself does not reveal the benefits of distillation.

---

> ### Author Rebuttal · Authors · 2025-07-31
>
> We thank the reviewer for their careful attention to our work and we look forward to a constructive discussion period.
>
> Since the reviewer has several comments about decodable vs $L$-step decodable / large $L$ vs small $L$, we address these first.
>
> - **Nomenclature:** Thanks for pointing out that ``decodable POMDP'' may not be standard terminology. When we say “decodable” with no preface, we indeed mean “H-step decodable”; we will clarify this in the paper.
> - **Experimentally verifying failure of $H$-step perfect decodability:** As the reviewer points out, we only explicitly verify failure of $L$-step decodability for $1 \leq L \leq 5$. However, from this experiment (Fig. 5) we can make two additional deductions: **(1)** For each step $1 \leq h \leq L$, the most recent $L$ observations comprise the entire history, and the decodability error at such steps $h$ is still large. The method of [6] requires perfect decodability at all steps, so their assumptions are indeed not met. **(2)** In all three tasks, the decodability error appears to essentially converge by $L = 5$. Since there is no principled reason why much older observations should be much more informative, we see this as strong evidence that the steps $h > L$ are also not $H$-step decodable.
> - **Comparison with (Cai et al., 2024):** The reviewer points out that approximate $L$-step decodability does not relax H-step perfect decodability, and suggests that:
>
>     > …the paper should either: (a) establish provable guarantees in the regime of large L, or (b) clearly discuss this distinction in the main text to avoid conceptual confusion.
>
>     Thanks for bringing this up; we will add clarification. In our introduction, we did not intend to claim that we have “relaxed” the assumption of (Cai et al., 2024), but merely to argue that perfect $H$-step decodability can be unrealistic in the settings that we care about. This demonstrates that the result of (Cai et al., 2024), though very interesting, is not directly applicable in our tasks of interest.
>
>     To be clear, our guarantee for expert distillation is incomparable with that of (Cai et al., 2024) — for several reasons, including what the reviewer mentioned. Another reason is that their guarantees are for a variant of expert distillation (compared to what is commonly implemented), which involves learning an autoregressive state decoder. This lets them demonstrate a provable computational benefit of latent state information. Our theory is complementary to theirs: we elucidate the factors that govern the performance of expert distillation as it is commonly implemented — i.e., with some constant frame-stack $L$. We will make this more clear in the final version.
>
>
> We now turn to the reviewer’s other comments.
>
> > Secondly, it seems the technical assumption that really enables approximate L-decodability is perturbed block MDP + deterministic transition. In terms of theoretical contributions, this is arguably quite strong to have structural assumptions on emission and transition at the same time.
>
> We agree that “perturbed block MDP + deterministic transition” is a strong assumption, but this is precisely the point of the theory. One of the main messages of our theoretical results is that even in the perturbed Block MDP setting (a natural theoretical testbed that imposes structure on the emissions based on the properties in the real applications), strong guarantees for expert distillation *cannot* be achieved without some assumption limiting the stochasticity of the transitions. We emphasize that the goal of our theory is not to identify the weakest possible assumptions under which POMDPs are tractable (though this question is very interesting and well-studied), but rather to understand the tradeoffs between popular (existing) algorithms, in settings where they are commonly applied (e.g., image-based robotics and autonomous driving). These settings motivate our theoretical abstractions, and the theory provides a principled explanation for the phenomena that we observe empirically.
>
> > Moreover, I understand in practice ppl often only use observation history. However, action history is also important in theory and in terms of verifying the theoretical conditions faithfully.
>
> We agree with the reviewer that the current experiment is to appeal to the common practice. For the rebuttal, we conduct experiments where the history is augmented with actions — for both DAgger and RL, and both deterministic and stochastic latent transitions. We find that adding actions to the history does not significantly change the main empirical findings, though it seems to (inconsistently) make RL worse. In the new results, we repeat each experiment with 3 random seeds and report the average. Below, we compare with the existing results (observation-only history):
>
> |  | DAgger | DAgger with action history **(new)** | RL | RL with action history **(new)** |
> | --- | --- | --- | --- | --- |
> | humanoid noise free | 739.1 | 710.8 | 572.4 | 568.6 |
> | humanoid noise=0.2 | 473.9 | 491.2 | 583.6 | 509.0 |
> | dog noise free | 722.5 | 739.3 | 762.6 | 719.5 |
> | dog noise=0.2 | 568.5 | 559.4 | 722.7 | 728.2 |
>
> Understanding when action history is unnecessary (or even detrimental) from a theoretical perspective is an interesting question.
>
> > As claimed, one important goal of the paper is to understand expert distillation vs RL. However, the message is that with deterministic transition, expert distillation match RL bounds. With stochastic transition, expert distillation is worse. Therefore, it seems the theory itself does not reveal the benefits of distillation.
>
> The primary “benefits of distillation” are already empirically well-understood: as a supervised learning method, expert distillation enjoys faster and more stable convergence than RL [2], i.e. it is computationally simpler. Our empirical results corroborate this common knowledge (Fig. 1). However, the main focus of our work — both the theory and empirics — is to shed light on a  much less-understood axis of the trade-off between distillation and RL: the price that we pay in *accuracy* for using (fast) expert distillation instead of (slow) RL.
>
> It is true that our theoretical results never show expert distillation to achieve superior accuracy than RL. This is corroborated by the empirical results, when computation is taken out of the picture (i.e. both algorithms are run to convergence). Moreover, it is to be expected: modulo issues of optimization (and capacity of the policy network), RL will always approach optimality with enough samples and computation, whereas expert distillation is fundamentally limited by misspecification of the expert policy (regardless of the capacity of the student network).
>
> [1] Eberhard, Muehlebach, and Vernade. "Partially Observable Reinforcement Learning with Memory Traces." *ICML 2025*.
>
> [2] Levine. “Imitation learning vs. offline reinforcement learning.” Lecture @ UC Berkeley RAIL. 2022.

---

> > ### Author Response · Authors · 2025-08-05
> >
> > Dear Reviewer Yzkr,
> >
> > As the discussion period is coming towards an end, we would greatly appreciate hearing whether our response addressed your concerns, or whether you have further questions for us.
> >
> > Thank you again for your time!

---

> > > ### Comment · Reviewer_Yzkr · 2025-08-08
> > >
> > > I think the reviewer for clarifications, especially that the distillation considered in the paper also operates on the finite-memory policy class. I have updated the evaluation accordingly.
> > >
> > > However, I do have a follow-up questions. The author said "distillation is computationally simpler". Is it an empirical conclusion from the existing empirical literature or the paper has also shed light on this either using theory or experiments?

---

> > > > ### Author Response · Authors · 2025-08-08
> > > >
> > > > Thanks for the response! Yes, the fact that distillation is computationally simpler than RL is observed in the existing empirical literature (see [2] for a summary). Our experiments corroborate it as well -- notice in Figure 1 that RL converges slower in terms of wall-clock time and has larger confidence intervals.
> > > >
> > > > [2] Levine. “Imitation learning vs. offline reinforcement learning.” Lecture @ UC Berkeley RAIL. 2022.

---

### Official Review · Reviewer_GVpx · 2025-07-03

**Clarity:** 4
**Significance:** 4
**Originality:** 3
**Rating:** 5
**Confidence:** 3

**Summary:**

The paper investigates the trade-off between privileged expert distillation—where a latent policy is trained using privileged information and then distilled into an executable policy via online behavior cloning—and reinforcement learning (RL) without privileged information, which uses frame-stacking, in partially observable Markov decision processes (POMDPs).

After formalizing the sources of error in POMDPs in terms of decodability and belief contraction, the authors introduce a theoretical model called the perturbed block MDP. This model extends the block MDP framework by incorporating some irreducible error in the emission distribution. Using this instructive model, the authors demonstrate that expert distillation can be both computationally and sample efficient in deterministic environments, where the belief state becomes concentrated over transitions. However, they also show that in the worst case, expert distillation may lead to irreducible sub-optimality.

Additionally, the authors present a series of controlled experiments on locomotion tasks (including humanoid, dog, and walker) to empirically illustrate their theoretical findings

**Questions:**

- in section 5.1, what do you mean by uniformly mixing dynamic? does it mean that we can teleport to any latent state from any latent state  at each step? does this mean that the belief state is uniform over the state space?
- in the case of uniformly mixing dynamic, do you think we ever need to do frame-stacking, since previous observations don;t inform us about the next state ?
- in section 5.2, I am curious what happens if we let expert distillation to run for more time and samples. does it recover the RL performance asymptotically ?  I am wondering if it is about having more samples or there are real fundamental gap between the two method in these environments.

**Ethical Concerns:**

["NO or VERY MINOR ethics concerns only"]

**Final Justification:**

the authors addresses my questions. I maintain my accept score

**Limitations:**

yes

**Paper Formatting Concerns:**

no issue

**Quality:**

4

**Strengths And Weaknesses:**

- The paper is well-motivated and clearly structured.
- The topic of expert distillation versus reinforcement learning is highly relevant and of significant practical interest.
- Although the paper is mathematically and notationally dense, the authors consistently provide intuitive explanations and high-level interpretations.
- The authors openly discuss the limitations of their theoretical model and acknowledge the presence of other confounding factors not accounted for.
I-  particularly appreciate the seamless integration of theorems with empirical validation, which helps to concretize the theoretical findings.
- Due to time constraints and limited background in POMDP theory, I was unable to thoroughly review the proofs in the appendix and therefore cannot fully assess the correctness of the theorems.

---

> ### Author Rebuttal · Authors · 2025-07-31
>
> We thank the reviewer for their careful attention to our work and we look forward to a constructive discussion period.
>
> > in section 5.1, what do you mean by uniformly mixing dynamic? does it mean that we can teleport to any latent state from any latent state at each step? does this mean that the belief state is uniform over the state space?
>
> First part of the question: Yes, by “uniformly mixing dynamics”, we mean that at each timestep $h$, the transition distribution from each latent state is uniform, i.e. we can “teleport”. This is of course a rather extreme setting, but the experiments (with more realistic sources of stochasticity) validate the basic theoretical intuition that we obtain from this setting.
>
> Second part of the question: No, recall that the belief state is the posterior distribution of $s_h$ after observing $(x_{1:h},a_{1:h-1})$. Even under uniformly mixing dynamics, the observation $x_h$ may be highly informative of $s_h$. For example, under a $\delta$-perturbed Block MDP with uniformly mixing dynamics, any true belief state $\mathbf{b_h}(x_{1:h},a_{1:h-1})$ should concentrate at least $(1-S\delta)$ mass onto some state, where $S$ is the size of the latent state. We will make sure to explain this point more clearly in the final version.
>
> > in the case of uniformly mixing dynamic, do you think we ever need to do frame-stacking, since previous observations don;t inform us about the next state ?
>
> Your understanding is correct: in this case, frame-stacking does not help expert distillation. This is essentially the message of Proposition 5.1: even as the frame-stack $L$ increases, the decodability error is irreducible, resulting in the stated suboptimality lower bound.
>
> Note that in this case frame-stacking also does not “help” for RL, but this is because POMDPs with uniformly mixing dynamics are already easy for RL: the history does not matter, so we can simply solve the $H$-step decision-making problem as $H$ independent 1-step problems, which are tractable by standard RL methods (both in theory and in practice).
>
> > in section 5.2, I am curious what happens if we let expert distillation to run for more time and samples. does it recover the RL performance asymptotically ? I am wondering if it is about having more samples or there are real fundamental gap between the two method in these environments.
>
> Our theory suggests that there is a real fundamental gap between expert distillation and RL under the stochastic dynamics, and our experiment is trying to demonstrate this point empirically. For this rebuttal, we have rerun the DAgger algorithm for noise levels {0.2, 0.3} and frame-stack $L$=3, in the setting of Fig. 3 (note that for noise level 0.1, Fig. 3 demonstrates that distillation is competitive with RL, so running DAgger longer would only solidify that conclusion). This time, we use 1.5x the number of samples. We repeat the new experiment with 3 random seeds and report the average. While the new averages are somewhat higher, they are now definitely at convergence; moreover, the original empirical conclusion still holds: there is a fundamental gap between distillation and RL in noisier environments. We use the percentage to denote the progress of the training.
>
> |  | DAgger (66%) | DAgger (83%) | DAgger (100%) | RL |
> | --- | --- | --- | --- | --- |
> | humanoid noise 0.2 | 434.2 | 478.1 | 473.9 | 583.6 |
> | humanoid noise 0.3 | 430.8 | 453.3 | 455.1 | 585.6 |
>
> To further demonstrate the generalizability of the finding, we repeat the experiment on another task, dog-walk, and we observe that the gap between expert distillation and RL is even larger.
>
> |  | DAgger (66%) | DAgger (83%) | DAgger (100%) | RL |
> | --- | --- | --- | --- | --- |
> | dog noise 0.2 | 540.0 | 570.3 | 568.5 | 722.7 |
> | dog noise 0.3 | 432.4 | 449.8 | 452.3 | 703.4 |
>
> We believe these new results will further consolidate our empirical findings.

---

### Decision · Program_Chairs · 2025-09-17

**Decision:**

Accept (spotlight)

**Comment:**

This paper studies the trade-off between privileged expert distillation (training with access to latent states) and standard RL without privileged information in partially observable environments. It introduces the perturbed block MDP as a theoretical model to compare approximate decodability (favoring distillation) vs. belief contraction (favoring RL). Experiments on locomotion tasks show that distillation is efficient and competitive under deterministic dynamics, but performance degrades under stochastic dynamics where RL would bennefit from frame-stacking. The authors further propose smoothing latent experts (adding motor noise during training) as a lightweight way to improve distillation robustness.

This is a very interesting paper that poses a very clear question and provides clear insights about algorithmic decisions that come up frequently in robotics and sequential decision-making with partial observability in general. The paper has theoretical novelty in terms of introducing approximate decodability. The perturbed block MDP model provides a clear framework for analyzing trade-offs between distillation and RL in POMDPs. The paper also combines formal analysis with controlled experiments on challenging locomotion tasks.

In terms of weaknesses, one of the most notable is that the analysis does not cover tasks requiring exploration purely for state disambiguation, where distillation may fundamentally fail. Additionally, while the paper offers guidelines, the decision over distilling vs. RL remains qualitative.

Reviewers were satisfied with the technical contributions of the paper. Many (minor) issues and clarifications were addressed during the rebuttal, while some limitations, such as comparison or analysis of asymmetric RL, remain and could not be accomplished during the course of the rebuttal. That said, there is unanimous support among reviewers that the paper is a strong contribution, and I agree with them. I recommend that the paper be accepted.